# Gradient Flow Provably Learns Robust Classifiers for Orthonormal GMMs

Hancheng Min [1 2]   René Vidal [1 2 3]

## Abstract

Deep learning-based classifiers are known to be vulnerable to adversarial attacks. Existing methods for defending against such attacks require adding a defense mechanism or modifying the learning procedure (e.g., by adding adversarial examples). This paper shows that for certain data distributions one can learn a provably robust classifier using standard learning methods and without adding a defense mechanism. More specifically, this paper addresses the problem of finding a robust classifier for a binary classification problem in which the data comes from an isotropic mixture of Gaussians with orthonormal cluster centers. First, we characterize the largest $\ell_2$-attack any classifier can defend against while maintaining high accuracy, and show the existence of optimal robust classifiers achieving this maximum $\ell_2$-robustness. Next, we show that given data from the orthonormal Gaussian mixture model, gradient flow on a two-layer network with a polynomial ReLU activation and without adversarial examples provably finds an optimal robust classifier.

## 1. Introduction

The vulnerability of neural networks to *adversarial attacks* (Szegedy et al., 2014), i.e., perturbations to their input that are typically human-imperceptible, has led to numerous efforts in building defenses against these attacks (Shafahi et al., 2019; Papernot et al., 2016; Wong et al., 2019; Guo et al., 2018; Cohen et al., 2019; Levine & Feizi, 2020; Yang et al., 2020; Sulam et al., 2020; Kinfu & Vidal, 2022). These defenses have been counteracted by new adaptive attacks (Athalye et al., 2018; Carlini et al., 2019; Croce & Hein, 2020), leading to new defenses and so on.

Large Language Models are also susceptible to adversarial attacks (Chao et al., 2023; Shah et al., 2023), leading to undesired or harmful outputs, and the competition between adversaries and defenders continues (Robey et al., 2023; Ji et al., 2024). While such a competition allows us to design more robust networks, it will not end unless many fundamental questions about adversarial robustness are answered.

One question is, *what is the maximum adversarial perturbation a neural network can tolerate?* Many works on certified robustness (Cohen et al., 2019; Fazlyab et al., 2020; Zhang et al., 2018) aim to find a certified radius such that a neural network can provably maintain a high prediction accuracy for adversarial attacks within that radius. However, their reported certified radii are often too small compared to what can be achieved by practical defenses (Tramèr et al., 2018; Guo et al., 2018; Gowal et al., 2020; Wu et al., 2020). Yet, practical defenses come at the cost of computing adversarial examples, or sophisticated model designs, mostly without theoretical guarantees, except for the case of linear classifiers (Zou et al., 2021). This also gives rise to an intriguing question: *Is it possible to find a robust network by standard training methods without adversarial examples?*

Note that this question might not always be well-defined. For example, Dobriban et al. (2023); Javanmard et al. (2020) prove that, for certain tasks, trade-offs between clean accuracy and adversarial robustness are unavoidable, hence the notion of robust networks requires additional specifications on how much accuracy one would sacrifice for robustness. However, for tasks without such trade-offs, a *robust classifier* can be accurate on clean data while maintaining a substantial level of robustness against adversarial attacks. The existence of such a robust classifier is closely related to data geometry. For instance, Pal et al. (2023; 2024) show that if the *data is localized*, i.e., if the distribution of the data given the class concentrates in a set of small volume, then a robust classifier is guaranteed to exist. Moreover, they show that a $2r$ separation (w.r.t. to some distance metric) between the sets that contain each class-conditioned probability mass is sufficient for the existence of a robust classifier against attacks of radius $r$ in the same distance metric. These results motivate us to explore the following question: *Can standard training methods, without adversarial examples, provably find a classifier for localized data that achieves maximum robustness while maintaining good clean accuracy?*

[1]Center for Innovation in Data Engineering and Science (IDEAS) [2]Department of Electrical and Systems Engineering [3]Department of Radiology, University of Pennsylvania, Philadelphia, U.S.A.. Correspondence to: Hancheng Min <hanchmin@seas.upenn.edu>.

*Proceedings of the $42^{nd}$ International Conference on Machine Learning*, Vancouver, Canada. PMLR 267, 2025. Copyright 2025 by the author(s).

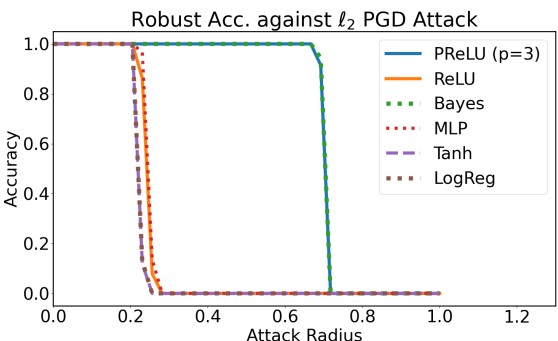

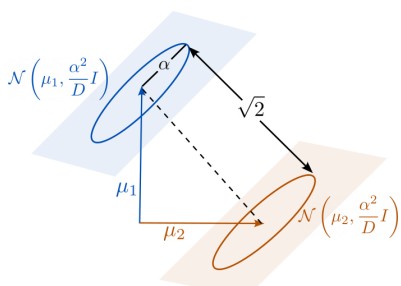

*Figure 1.* Illustration of two clusters in high-dimensions, each concentrated on a $(D-1)$-dimensional affine subspace such that the subspaces are separated by a Euclidean distance of $\sqrt{2}$.

*Figure 2.* Given mixture of Gaussian data with 12 positive clusters and 8 negative clusters ($D = 2000$), gradient descent (SGD, small initialization) on (bias-free, width-200) two-layer ReLU network (ReLU) fails to find a robust classifier. This issue persists after 1) increasing depth to 4 (MLP); 2) (blindly) switching to another activation (Tanh); or 3) using a linear classifier (LogReg). However, by choosing a suitable activation (pReLU, $p=3$), GD can find a nearly optimal robust classifier. Here, the plot emphasizes the importance of choosing an appropriate function class (activation). Additional experiments in Appendix B.2 highlight the importance of choosing proper network parameterization and initialization.

In this paper, we show that for certain localized data distributions, one can characterize the maximum robustness any classifier can achieve based on how class-conditional probability masses are separated. We also show one can make suitable architectural designs that exploit the data geometry such that a nearly optimal robust classifier can be provably learned by standard training, such as gradient flow (GF).

In what follows, we explain each one of these contributions in more detail. Before that, we introduce our data model.

**Orthonormal Gaussian Mixture Model**. Consider a balanced mixture of $K$ Gaussians in $\mathbb{R}^D$, split into two classes:

$$\mathcal{N}(\boldsymbol{\mu}_1, \alpha^2 \boldsymbol{I}/D), \cdots, \mathcal{N}(\boldsymbol{\mu}_{K_1}, \alpha^2 \boldsymbol{I}/D), \quad \text{(Positive Class)}$$
$$\mathcal{N}(\boldsymbol{\mu}_{K_1+1}, \alpha^2 \boldsymbol{I}/D), \cdots, \mathcal{N}(\boldsymbol{\mu}_K, \alpha^2 \boldsymbol{I}/D), \quad \text{(Negative Class)}$$

where the *cluster centers* $\boldsymbol{\mu}_1, \cdots, \boldsymbol{\mu}_K \in \mathbb{R}^D$ are othonormal, $\alpha^2$ denotes the *intra-cluster variance*, and the data dimension $D$ is sufficiently large. One can show that this mixture of Gaussian distribution satisfies data localization and separation properties similar to those studied in Pal et al. (2023), thus a robust classifier is guaranteed to exist.

**Maximum $\ell_2$-robustness**. As illustrated in Figure 1 for the case of two clusters (one from the positive class and one from the negative class), the class-conditioned probability masses concentrate around two $(D-1)$-dimensional affine subspaces[1] separated by a Euclidean distance of almost $\sqrt{2}$.

---

[1]To see this, start with the fact that the distribution $\mathcal{N}(\mu, \frac{\alpha^2}{D}\boldsymbol{I})$ concentrates around a sphere of radius $\alpha$, then use the result stating that one can cover most masses of this high-dimensional sphere by any set $S$ that is a Minkowski sum of a $\mathcal{O}(1/\sqrt{D})$-radius ball and a $(D-1)$-dimensional affine subspace that contains the Gaussian mean $\mu$ (one should think $S$ being the inflated version of the affine subspaces shown in Figure 1 with thickness $\mathcal{O}(1/\sqrt{D})$).

Based on this observation, our first set of results are:

**Theorem** (Proposition 1 & 2, informal). *No classifier can defend against an adversarial attack of $\ell_2$ radius $\frac{\sqrt{2}}{2}$. However, one can construct a nearly optimal robust classifier that can defend against attacks of radius arbitrarily close to $\frac{\sqrt{2}}{2}$ when $D$ is sufficiently large.*

Our results show that data localization and separation are important properties in understanding the maximum achievable robustness for a classifier. Moreover, we will show that the classifier we construct is the Bayes optimal classifier w.r.t. the 0-1 loss, which operates as a nearest-cluster rule: classifiers that exploit the multi-cluster data structure are naturally and optimally robust.

**Learning optimal robust networks**. So far everything seems to be intuitive and straightforward given the fairly simple distributional assumption. However, issues arise when one does not know the data distribution a priori and seeks a classifier by training a neural network on sampled data via gradient descent (GD). As Figure 2 suggests, a trained ReLU network fails to find a classifier with the same level of robustness as the Bayes classifier (which indeed can defend against attack of radius $\sim \frac{\sqrt{2}}{2}$, as our results suggest). This matter is first discussed by Frei et al. (2023), where they show that any two-layer ReLU network trained by GD under data samples from orthonormal Gaussian mixture is non-robust against adversarial attacks of $\ell_2$-radius $\Theta\left(\frac{1}{\sqrt{K}}\right)$, where $K$ is the total number of clusters. Later, Min & Vidal (2024) show that this issue is caused by the fact that a ReLU network fails to learn, internally with its weight parameters, the multi-cluster structure of the data distribution, even if the sampled data points are revealing such a structure, and a convergence analysis of GD in Li et al. (2025) theoreti-

cally supports this argument. Therefore, while the structural property of the data distribution allows one to construct an optimal robust classifier, gradient descent algorithms on neural networks may struggle to learn these key properties, leading to non-robust classifiers.

To address this issue, Min & Vidal (2024) propose to change the activation. More specifically, replacing the ReLU activation with a polynomial ReLU activation (pReLU) with polynomial degree $p$ as a hyperparameter (defined later in (5)). They empirically show that when $p$ is large, the pReLU network can internally learn the data structure, leading to a more robust classifier. However, a rigorous analysis of convergence is not provided. Our second set of results is to develop a full convergence analysis for gradient flow on a two-layer pReLU network and show that:

**Theorem** (Theorem 1 & Corollary 1, informal). *When the intra-cluster variance $\alpha^2$ is sufficiently small, gradient flow on pReLU networks (5) with $p > 2$ converges to a nearly optimal robust classifier.*

Our result is based on prior works on gradient descent/flow with small initialization on two-layer ReLU networks (Maennel et al., 2018; Phuong & Lampert, 2021; Boursier et al., 2022; Kumar & Haupt, 2024; Chistikov et al., 2023; Wang & Ma, 2023; Min et al., 2024) and we extend their convergence analyses to pReLU networks. We show how the implicit bias (Vardi, 2023) of the gradient flow dynamics critically depends on a careful choice of activation function, allowing the network to learn accurately the underlying data structure, which, as we have discussed, is essential for finding a robust classifier.

**Notation**. We denote the inner product between vectors $\boldsymbol{x}$ and $\boldsymbol{y}$ by $\langle \boldsymbol{x}, \boldsymbol{y} \rangle = \boldsymbol{x}^\top \boldsymbol{y}$, and the cosine of the angle between them as $\cos(\boldsymbol{x}, \boldsymbol{y}) = \langle \frac{\boldsymbol{x}}{\|\boldsymbol{x}\|}, \frac{\boldsymbol{y}}{\|\boldsymbol{y}\|} \rangle$. For an $n \times m$ matrix $\boldsymbol{A}$, we let $\|\boldsymbol{A}\|$ and $\|\boldsymbol{A}\|_F$ denote the spectral and Frobenius norm of $\boldsymbol{A}$, respectively. We define $\mathbb{1}_A$ as the indicator for a statement $A$: $\mathbb{1}_A = 1$ if $A$ is true and $\mathbb{1}_A = 0$ otherwise, and define $[\cdot]_+ := \max\{\cdot, 0\}$. We also let $\mathcal{N}(\boldsymbol{\mu}, \boldsymbol{\Sigma}^2)$ denote the normal distribution with mean $\boldsymbol{\mu}$ and covariance matrix $\boldsymbol{\Sigma}^2$, and $\mathrm{Unif}(S)$ denote the uniform distribution over a set $S$. Lastly, we let $[N]$ denote the integer set $\{1, \cdots, N\}$ and let $\mathbb{S}^{D-1}$ be the unit-sphere in $\mathbb{R}^D$.

## 2. Optimal Robust Classifiers for Orthonormal Gaussian Mixture

We start by studying the optimal robust classifiers for orthonormal Gaussian Mixture.

**Orthonormal Gaussian Mixture Model**. We study a balanced mixture of $K$ Gaussians, with $K_1$ of them belonging to the positive (+1) class and $K_2 := K - K_1$ of them belonging to the negative (−1) class. Formally, consider a tu-

ple of random variables $(X, Y, Z)$ on $\mathbb{R}^D \times \{+1, -1\} \times [K]$ representing *observed data*, *observed class label*, and *latent cluster membership*, respectively, defined as follow:

$$Z \sim \mathrm{Unif}(\{1, \cdots, K\}),$$
$$X|Z \sim \mathcal{N}\left(\boldsymbol{\mu}_Z, \alpha^2 \boldsymbol{I}/D\right), \ Y|Z = \mathbb{1}_{Z \leq K_1} - \mathbb{1}_{Z > K_1}, \ (1)$$

where the $\boldsymbol{\mu}_1, \cdots, \boldsymbol{\mu}_K$, called *cluster centers*, are a set of orthonormal vectors in $\mathbb{R}^D$, i.e. $\langle \boldsymbol{\mu}_k, \boldsymbol{\mu}_l \rangle = \mathbb{1}_{l=k}$. We denote the marginal distribution of the $(X, Y)$ pair by $\mathcal{D}_{X,Y}$, and use $(\boldsymbol{x}, y)$ to denote a sample from $\mathcal{D}_{X,Y}$.

$\ell_2$**-robust classifier for** $\mathcal{D}_{X,Y}$. Our interest is to find a classifier that not only accurately predicts the label $y$ given an observed data $\boldsymbol{x}$, but does so in a way that is robust to some additive adversarial attacks on $\boldsymbol{x}$. Specifically, we seek a classifier $f: \mathbb{R}^D \to \mathbb{R}$ such that given a sample $(\boldsymbol{x}, y)$ from $\mathcal{D}_{X,Y}$, with high probability, we have $f(\boldsymbol{x})y > 0$, which suggests that $\mathrm{sign}(f(\boldsymbol{x}))$ correctly predicts the label $y$; Moreover, we require that for some $r > 0$, $f$ is *robust to additive adversarial attacks of $\ell_2$-norm radius $r$*, i.e., $\min_{\|\boldsymbol{d}\| \leq 1} f(\boldsymbol{x} + r\boldsymbol{d})y > 0$ with high probability over sample $(\boldsymbol{x}, y)$. This suggests that $\mathrm{sign}(f(\boldsymbol{x} + r\boldsymbol{d}))$ still makes a correct prediction on $y$ even though $\boldsymbol{x}$ has been corrupted by some attack $r\boldsymbol{d}$. Ideally, we want a classifier that is robust to attack of radius $r$, with as large $r$ as possible.

**Maximum achievable $\ell_2$-robustness**. Inevitably, any classifier fails to be robust if the adversary has too much power, i.e., the attack radius $r$ exceeds some value. Indeed, for the data distribution $\mathcal{D}_{X,Y}$ of our interest, no classifier can defend against attacks of radius $\frac{\sqrt{2}}{2}$, as shown below:

**Proposition 1.** *Let $f : \mathbb{R}^D \to \mathbb{R}$ be any Lebesgue measurable function such that the random variable $\min_{\|\boldsymbol{d}\| \leq 1}\left[f\left(\boldsymbol{x} + \frac{\sqrt{2}}{2}\boldsymbol{d}\right)y\right]$ is also measurable. Given a sample $(\boldsymbol{x}, y) \sim \mathcal{D}_{X,Y}$, we have*

$$\mathbb{P}\left(\min_{\|\boldsymbol{d}\| \leq 1}\left[f\left(\boldsymbol{x} + \frac{\sqrt{2}}{2}\boldsymbol{d}\right)y\right] \leq 0\right) \geq \frac{\min\{K_1, K_2\}}{K}. \ (2)$$

We refer the readers to Appendix C.1 for the proof. We explain Proposition 1 from a geometric perspective, expanding upon the discussion in the introduction: Consider data from two clusters $\mathcal{N}\left(\boldsymbol{\mu}_1, \frac{\alpha^2}{D}\boldsymbol{I}\right)$ and $\mathcal{N}\left(\boldsymbol{\mu}_2, \frac{\alpha^2}{D}\boldsymbol{I}\right)$ corresponding to different classes. As shown in Figure 1, when ambient dimension $D$ is large, we expect that each cluster concentrates around a $D - 1$ affine subspace that is orthogonal to the vector $\boldsymbol{\mu}_1 - \boldsymbol{\mu}_2$. Most importantly, the distance between these two affine subspaces is $\sqrt{2}$, suggesting that given any decision boundary that separates two affine subspaces, an adversary can perturb a substantial portion of the probability mass of these clusters to cross the boundary with an attack radius $\frac{\sqrt{2}}{2}$. The same argument holds for $K$-clusters, where any pair of clusters is separated by a Euclidean distance

$\sqrt{2}$. We also note that extending Proposition 1 to attacks in another metric amounts to measuring this separation in that metric. Our second result shows the Bayes optimal classifier w.r.t. the 0-1 loss is also nearly optimally robust:

**Proposition 2.** *The Bayes optimal classifier w.r.t. the 0-1 loss is* $\mathrm{sign}\left(f^*(\boldsymbol{x})\right)$*, where*

$$f^*(\boldsymbol{x}) = \sum_{k=1}^{K_1} \exp\left(\frac{D\langle \boldsymbol{x}, \boldsymbol{\mu}_k\rangle}{\alpha^2}\right) - \sum_{k=K_1+1}^{K} \exp\left(\frac{D\langle \boldsymbol{x}, \boldsymbol{\mu}_k\rangle}{\alpha^2}\right).$$

*Moreover, given a sample* $(\boldsymbol{x}, y) \sim \mathcal{D}_{X,Y}$*, we have, for any* $\frac{2\sqrt{2}\alpha^2 \log K}{D} \le \nu \le \sqrt{2}$*,*

$$\mathbb{P}\left(\min_{\|\boldsymbol{d}\|\le 1}\left[f^*\left(\boldsymbol{x} + \frac{\sqrt{2}-\nu}{2}\boldsymbol{d}\right)y\right] > 0\right)$$
$$\ge 1 - 2K\exp\left(-\frac{D\nu^2}{64\alpha^2}\right). \quad (3)$$

We refer the readers to Appendix C.2 for the proof. If we pick $\nu = \Theta\left(\left(\frac{\alpha^2}{D}\right)^{\frac{1}{4}}\right)$ in Proposition 2, then the result shows that $f^*$ is robust against attacks of radius $\frac{\sqrt{2}}{2} - \Theta\left(\left(\frac{\alpha^2}{D}\right)^{\frac{1}{4}}\right)$ with probability at least $1 - \mathcal{O}\left(K\exp\left(-\left(\frac{D}{\alpha^2}\right)^{\frac{1}{2}}\right)\right)$ over new sample from $\mathcal{D}_{X,Y}$. Therefore, $f^*$ is nearly optimal robust when $\frac{\alpha^2}{D} = o(1)$, i.e., the ambient dimension is large or the intra-class variance is small.

**Interpreting $f^*$ as a nearest-cluster rule**. One can show that (please see Appendix C.2 for the derivation)

$$\mathrm{sign}\left(f^*(\boldsymbol{x})\right) = \mathrm{sign}\Big(\max_{1\le k\le K_1}\langle\boldsymbol{x},\boldsymbol{\mu}_k\rangle - \max_{K_1+1\le k\le K}\langle\boldsymbol{x},\boldsymbol{\mu}_k\rangle$$
$$+ \mathcal{O}\big(\frac{\alpha^2}{D}\log K\big)\Big). \quad (4)$$

When the error $\mathcal{O}\left(\log K\frac{\alpha^2}{D}\right)$ is small, the Bayes classifier $f^*(\boldsymbol{x})$ finds the closest cluster center to $\boldsymbol{x}$ and outputs the label to that cluster, which is a nearest-cluster rule. Therefore, by respecting the multi-cluster structure of $\mathcal{D}_{X,Y}$, $f^*$ achieves the maximum $\ell_2$-robustness.

So far we have shown that a nearly optimal robust classifier for $\mathcal{D}_{X,Y}$ can be easily constructed as a nearest-cluster rule. However, as we discussed in the introduction, gradient descent algorithms with sampled data often fail to find a classifier with the same level of robustness. Next, we address the problem of finding a nearly optimal robust classifier by gradient flow dynamics.

## 3. Optimal Robust Classifiers Obtained via Gradient Flow

In this section, we aim to find a nearly optimal $\ell_2$-robust classifier for $\mathcal{D}_{X,Y}$ by vanilla gradient descent without adversarial training. We start by stating the problem of training

two-layer networks with gradient flow. Then we show that with a pReLU activation, gradient flow provably finds a classifier that is nearly optimal $\ell_2$-robust.

### 3.1. Preliminaries: training on two-layer networks for orthonormal Gaussian mixture

**pReLU network**. We consider a two-layer pReLU network (Min & Vidal, 2024), which is defined as follows:

$$f^{(p)}(\boldsymbol{x};\boldsymbol{\theta}) = \sum_{i=1}^{h} v_j \frac{\sigma^p(\langle\boldsymbol{x},\boldsymbol{w}_j\rangle)}{\|\boldsymbol{w}_j\|^{p-1}} \quad \left(\boldsymbol{\theta} := \{\boldsymbol{w}_j, v_j\}_{j=1}^{h}\right). \quad (5)$$

Note that $f^{(p)}$ can be viewed as a generalized version of the ReLU network. When $p = 1$, $f^{(1)}$ is exactly a two-layer ReLU network. When $p > 1$ and that $\boldsymbol{x}$ is approximately unit-norm, the output of the hidden activation is approximately equal to the one of the ReLU network multiplied by $\cos^{p-1}(\boldsymbol{x}, \boldsymbol{w}_j)$ (Min & Vidal, 2024), which discourages large angle separation between data $\boldsymbol{x}$ and *neuron* $\boldsymbol{w}_j$.

**$\ell_2$-loss function and balanced dataset**. Given a dataset $\{\boldsymbol{x}_i, y_i\}_{i=1}^{n}$, define the loss function as $\mathcal{L}(\boldsymbol{\theta}; \{\boldsymbol{x}_i, y_i\}_{i=1}^{n}) = \sum_{i=1}^{n} \ell(y_i, \hat{y}_i)$, where $\hat{y}_i = f^{(p)}(\boldsymbol{x}_i; \boldsymbol{\theta})$. For classification problems, the typical choice for $\ell$ is the exponential loss $\exp(-y\hat{y})$, or the logistic loss $\log(1 + \exp(-y\hat{y}))$. Most of our theoretical analysis works for these choices for $\ell$. However, using classification losses poses additional challenges in analyzing the late phase of the training (details explained in later sections). Therefore, our theorem considers the $\ell_2$-loss: $\ell(y, \hat{y}) = \frac{1}{2}\|y - \hat{y}\|^2$, and the extension to classification losses is discussed in Appendix A.

As for the dataset, since $\mathcal{D}_{X,Y}$ samples data with equal probability from each cluster, there are approximately equal number of samples from each cluster when we sample a large number of data. Therefore, instead of considering a dataset directly sampled from $\mathcal{D}_{X,Y}$, we consider the following *balanced dataset* $\hat{\mathcal{D}} = \{\boldsymbol{x}_i, y_i\}_{i=1}^{KN}$, where

$$\boldsymbol{x}_i \sim \mathcal{N}\left(\boldsymbol{\mu}_k, \alpha^2\boldsymbol{I}/D\right), y_i = \mathbb{1}_{k\le K_1} - \mathbb{1}_{k>K_1},$$
$$(k-1)N + 1 \le i \le kN, \ 1 \le k \le K. \quad (6)$$

We call this dataset balanced because $\hat{\mathcal{D}}$ has exactly $N$ samples from each cluster $\mathcal{N}(\boldsymbol{\mu}_k, \alpha^2\boldsymbol{I}/D)$. This assumption allows us to omit the additive perturbations in our analysis introduced by imbalanced per-cluster sample size.

**Gradient flow with small and balanced initialization**. Given the network parametrization $\boldsymbol{\theta}$ and the loss function $\mathcal{L}$ constructed from a balanced dataset $\hat{\mathcal{D}}$, we consider training the network by the following *gradient flow* dynamics[2]:

$$\dot{\boldsymbol{\theta}} = -\nabla_{\boldsymbol{\theta}}\mathcal{L}(\boldsymbol{\theta}; \hat{\mathcal{D}}), \qquad \boldsymbol{\theta}(0) = \boldsymbol{\theta}_0, \quad (7)$$

---

[2]Note that we focus on pReLU network with $p > 2$. Therefore, the network $f^{(p)}$ differentiable everywhere w.r.t. $\boldsymbol{\theta}$, and so is $\mathcal{L}$.

We assume the initialization $\boldsymbol{\theta}(0)$ is $\epsilon$-*small* and *balanced*, formally defined as the following.

**Assumption 1** ($\epsilon$-small and balanced initialization)**.** *The initialization $\boldsymbol{\theta}(0) = \{\boldsymbol{w}_j(0), v_j(0)\}_{j=1}^h$ satisfies the following: there exists an **initialization shape** $\{\boldsymbol{w}_{j0}, v_{j0}\}_{j=1}^h$ with $W_{\min} \le \|\boldsymbol{w}_{j0}\| \le W_{\max}, \forall j$, for some $W_{\min}, W_{\max} > 0$ and an **initialization scale** $\epsilon > 0$ such that*

$$\boldsymbol{w}_j(0) = \epsilon \boldsymbol{w}_{j0}, \ v_j(0) = \epsilon v_{j0}, \ \|\boldsymbol{w}_{j0}\| = |v_{j0}|, \forall j. \quad (8)$$

Under a balanced initialization, we have $\|\boldsymbol{w}_j(0)\| = |v_j(0)|, \forall j$, and this balancedness holds throughout the GF trajectory (See Appendix E.1): $\|\boldsymbol{w}_j(t)\| = |v_j(t)|, \forall j$. The balancedness between $w_j$ and $v_j$ allows us to focus on the dynamics of $w_j$, which has been a common assumption in prior work of this type (Maennel et al., 2018; Boursier et al., 2022; Chistikov et al., 2023; Min et al., 2024). Readers may view this assumption as made out of convenience, but we think it is essential for a tractable analysis, and at the same time, the theoretical results out of this assumption match the empirical results when no balancedness is enforced (Min et al., 2024). Moreover, this assumption allows for an elegant interpretation of the dynamics of $w_j$ (Maennel et al., 2018; Boursier & Flammarion, 2024) as searching for some directions that maximize its alignment with the data.

Given a balanced initialization, one can show that $\text{sign}(v_j(t)) = \text{sign}(v_j(0)), \forall j, \forall t \ge 0$ (Boursier et al., 2022). Roughly speaking, this means that $\text{sign}(v_j(0))$ determines the dynamical behavior of neuron $\boldsymbol{w}_j$ under gradient flow: neurons with $\text{sign}(v_j(0)) = +1$ tend to align its direction with one of the positive cluster centers, $\boldsymbol{\mu}_k, \ k = 1, \cdots, K_1$, and those with $\text{sign}(v_j(0)) = -1$ tend to align with one of the negative cluster centers. For this reason, we define the following neuron index sets: $\mathcal{N}_+ := \{j \in [h] : \text{sign}(v_j(0)) = +1\}$ and $\mathcal{N}_- := \{j \in [h] : \text{sign}(v_j(0)) = -1\}$.

## 3.2. Main results: pReLU ($p > 2$) provably finds (near)-optimal robust classifiers

**pReLU classifier and the conjecture**. Min & Vidal (2024) study the adversarial robustness of the following pReLU classifier (a particular case of $f^{(p)}(\boldsymbol{x}; \boldsymbol{\theta})$ for a choice of $\boldsymbol{\theta}$):

$$F^{(p)}(\boldsymbol{x}) = \sum_{k=1}^{K_1} \sigma^p(\langle \boldsymbol{x}, \boldsymbol{\mu}_k \rangle) - \sum_{k=K_1+1}^{K} \sigma^p(\langle \boldsymbol{x}, \boldsymbol{\mu}_k \rangle), \quad (9)$$

and show that $F^{(p)}$ with $p > 2$ is robust to adversarial attacks of $\ell_2$ radius arbitrarily close to $\frac{\sqrt{2}}{2}$ when $\frac{D}{\alpha^2}$ is large.[3] They conjecture that when $p$ is large and the intra-cluster variance $\alpha^2$ is small, the gradient flow on pReLU network

---

[3]To provide our view on why $F^{(p)}$ is robust, we show in Appendix D that it behaves like a nearest-cluster rule for large $p$.

$f^{(p)}(\cdot; \boldsymbol{\theta})$ with small initialization finds a classifier that is close to $F^{(p)}$ up to a constant scaling factor. Then they argue that such proximity to $F^{(p)}$ implies that the trained network has the same level of robustness as $F^{(p)}$. Our main results in Section 3.2.2 fully prove this conjecture with $p > 2$.

**Remark 1.** *Min & Vidal (2024) also conjecture that GF on a ReLU network ($p = 1$) finds a classifier that is not robust against adversarial attacks of $\ell_2$-radius $\Theta(\frac{1}{\sqrt{K}})$, which is later proved by Li et al. (2025). As extensively discussed in both works, the main reason why a trained ReLU network is not robust is because of its inability to learn the multi-cluster structure of $\mathcal{D}_{X,Y}$ via GD/GF.*

**Closeness to $F^{(p)}$ implies robustness**. We first show that given any classifier $f$ that is positively homogeneous of degree 1 w.r.t. $\boldsymbol{x}$ and is close to $F^p$ in terms of some distance measure, it is nearly optimal robust when the intra-class variance is small (We refer to Appendix D for the proof.).

**Proposition 3.** *Let $p > 2$. Given a classifier $f$ that satisfies $f(\gamma \boldsymbol{x}) = \gamma f(\boldsymbol{x}), \forall \boldsymbol{x} \in \mathbb{R}^D, \forall \gamma > 0$ and $\text{dist}(f, F^{(p)}) = \inf_{c>0} \sup_{\boldsymbol{x} \in \mathbb{S}^{D-1}} |cf(\boldsymbol{x}) - F^{(p)}(\boldsymbol{x})| \le \nu$ for some $0 < \nu \le \left(\frac{\sqrt{2}}{8}\right)^p$. Then for a sample $(\boldsymbol{x}, y) \sim \mathcal{D}_{X,Y}$, we have*

$$\mathbb{P}\left(\min_{\|\boldsymbol{d}\| \le 1}\left[f\left(\boldsymbol{x} + \frac{\sqrt{2} - 8\nu^{\frac{1}{p}}}{2}\boldsymbol{d}\right)y\right] > 0\right)$$
$$\ge 1 - 2K \exp\left(-\frac{D\nu^{\frac{2}{p}}}{2K^2\alpha^2}\right) - 4 \exp\left(-\frac{3}{8\alpha^2}\right). \quad (10)$$

Given this result, it remains to show that gradient flow finds a network $f^{(p)}(\cdot; \boldsymbol{\theta})$ (which is positively homogeneous of degree 1) that is close to $F^{(p)}$ in the distance measure defined above. We will first discuss an additional assumption required on the initialization, then state our main result.

### 3.2.1. Non-degenerate initialization shape

To properly define a non-degenerate initialization shape, we need to define a *radial Voronoi tessellation* of $\mathbb{R}^{D-1} \setminus \{0\}$ given a tuple of unit-norm vectors $\{\boldsymbol{\mu}_k\}_{k \in \mathcal{K}}$.

**Definition 1.** *Given a set of unit-norm vectors $\{\boldsymbol{\mu}_k\}_{k \in \mathcal{K}}$, define the following*

$$\mathcal{R}_k := \big\{\boldsymbol{w} \in \mathbb{R}^{D-1} \setminus \{0\} \ | \ [\cos(\boldsymbol{\mu}_k, \boldsymbol{w})]_+ > [\cos(\boldsymbol{\mu}_l, \boldsymbol{w})]_+, \forall l \ne k\big\}, k \in \mathcal{K},$$
**(Voronoi regions)**
$$\mathcal{R}^\circ := \big\{\boldsymbol{w} \in \mathbb{R}^{D-1} \setminus \{0\} \ | \ [\cos(\boldsymbol{\mu}_k, \boldsymbol{w})]_+ = 0, \forall k \in \mathcal{K}\big\}.$$
**(Void region)**

From this definition, it is clear that $\mathcal{R}_k, k \in \mathcal{K}$ and $\mathcal{R}^\circ$ are disjoint subsets of $\mathbb{R}^{D-1} \setminus \{0\}$. We are ready to define a non-degenerate initialization shape:

**Definition 2** (Non-degenerate initialization shape)**.** *Given a set of unit-norm vectors $\{\boldsymbol{\mu}_k\}_{k\in\mathcal{K}}$, let $\{\mathcal{R}_k\}_{k\in\mathcal{K}}$ and $\mathcal{R}^\circ$ be their Voronoi and void regions, as per Definition 1. A set of initialization shape $\{\boldsymbol{w}_{j0}\}_{j\in\mathcal{N}}$ is **non-degenerate** w.r.t. $\{\boldsymbol{\mu}_k\}_{k\in\mathcal{K}}$ if there exist disjoint subsets $\{\mathcal{N}_k\}_{k\in\mathcal{K}}, \mathcal{N}^\circ$ of $\mathcal{N}$, such that $\mathcal{N} = (\bigcup_{k\in\mathcal{K}} \mathcal{N}_k) \cup \mathcal{N}^\circ$ and*

- *(Neurons must be within one of the regions) $\boldsymbol{w}_{j0} \in \mathcal{R}_k, \forall k \in \mathcal{N}_k$; $\boldsymbol{w}_{j0} \in \mathcal{R}^\circ, \forall k \in \mathcal{N}^\circ$;*

- *(Non-void regions must contain at least one neuron) $\mathcal{N}_k \neq \emptyset, \forall k \in \mathcal{K}.$*

*Moreover, we let $d(\boldsymbol{w}, S) = 1 - \sup_{\boldsymbol{s}\in S, \boldsymbol{s}\neq 0} \cos(\boldsymbol{w}, \boldsymbol{s})$ and define **non-degeneracy gap**:*

$$\Delta := \min\Big\{ \min_{j \in \bigcup_{k\in\mathcal{K}} \mathcal{N}_k} d\Big(\boldsymbol{w}_{j0}, \partial(\bigcup_{k\in\mathcal{K}} \mathcal{R}_k)\Big),$$
$$\min_{j\in\mathcal{N}^\circ} d\Big(\boldsymbol{w}_{j0}, \partial\mathcal{R}^\circ\Big)\Big\}. \quad (11)$$

Whenever a vector $\boldsymbol{w}$ falls into one of the $\mathcal{R}_k$, it means that: 1) the angle between $\boldsymbol{w}$ and the corresponding $\boldsymbol{\mu}_k$ is less than $\frac{\pi}{2}$; and 2) compared to all other $\boldsymbol{\mu}$s, $\boldsymbol{\mu}_k$ is the closest (in angle) to $\boldsymbol{w}$. This suggests that neurons initialized within some $\mathcal{R}_k$ converge to the corresponding $\boldsymbol{\mu}_k$ under GF, and those initialized within $\mathcal{R}^\circ$ stay in $\mathcal{R}^\circ$. This is indeed the case, as we will see in Section 3.2.3.

The special case when a neuron is exactly initialized on the boundary of these Voronoi regions $\partial(\bigcup_{k\in\mathcal{K}} \mathcal{R}_k)$ cannot be analyzed since if a neuron has equal angular distance to two $\boldsymbol{\mu}$ vectors, there is no way to determine which $\boldsymbol{\mu}$ vector it converges to under GF with sampled data around these $\boldsymbol{\mu}$ vectors. Similarly, if a neuron is initialized at the boundary between some $\mathcal{R}_k$ and $\mathcal{R}^\circ$, then we can not determine whether it converges to $\boldsymbol{\mu}_k$, or it falls into the interior of $\mathcal{R}^\circ$ and stays there after that. Therefore we require an initialization shape with a positive non-degeneracy gap. Moreover, every $\mathcal{R}_k$ must contain one neuron, ensuring the corresponding $\boldsymbol{\mu}_k$ gets learned. This leads to our assumption:

**Assumption 2.** *(The initialization has a non-degeneracy gap of at least $\Delta$) $\exists \Delta > 0$ such that $\{\boldsymbol{w}_{j0}\}_{j\in\mathcal{N}_+}$ is non-degenerate w.r.t. $\{\boldsymbol{\mu}_k\}_{1\leq k\leq K_1}$ with a non-degeneracy gap of at least $\Delta$, and $\{\boldsymbol{w}_{j0}\}_{j\in\mathcal{N}_-}$ is non-degenerate w.r.t. $\{\boldsymbol{\mu}_k\}_{K_1\leq k\leq K}$ with a non-degeneracy gap of at least $\Delta$.*

As one can see, this condition is stated per class: Positive (Negative) neurons must be initialized to be non-degenerate w.r.t. cluster centers from the positive (negative) class, and we have disjoint subsets $\{\mathcal{N}_k\}_{1\leq k\leq K_1}, \mathcal{N}_+^\circ$ of $\mathcal{N}_+$ (disjoint subsets $\{\mathcal{N}_k\}_{K_1+1\leq k\leq K}, \mathcal{N}_-^\circ$ of $\mathcal{N}_-$). As suggested in our previous discussion, we show that (See Section 3.2.3) under GF, all neurons in $\mathcal{N}_k$ converge in angle to $\boldsymbol{\mu}_k$, which is an essential part of our theoretical results. We also let $\mathcal{N}_c := \mathcal{N}_+^\circ \cup \mathcal{N}_-^\circ$ contain indices of neurons that are initialized within the void region.

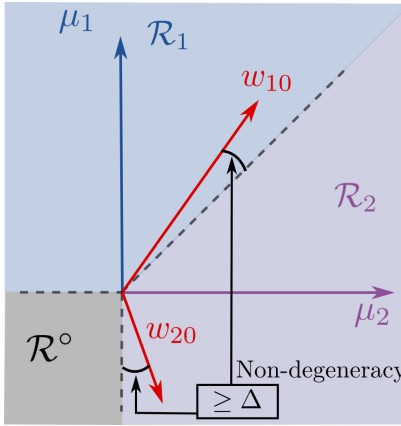

*Figure 3.* Illustration of a non-degenerate initialization shape $\{\boldsymbol{w}_{10}, \boldsymbol{w}_{20}\}$ w.r.t. two orthonormal vectors $\{\boldsymbol{\mu}_1, \boldsymbol{\mu}_2\}$.

### 3.2.2. CONVERGENCE OF PRELU ($p > 2$) ON ORTHONORMAL GAUSSIAN MIXTURE

Now we are ready to state our main theorem:

**Theorem 1** (pReLU converges to optimal robust classifier for orthonormal Gaussian mixture)**.** *Let $p > 2$. Given $0 \leq \delta \leq 1$ and a sufficiently small $\alpha_0^2$, assume the intra-cluster variance, the data dimension, and per-cluster sample size satisfy that $\alpha^2 \leq \alpha_0^2$, $D \geq \tilde{\Omega}(\alpha_0^{-2})$ and $\tilde{\Omega}(\alpha_0^{-2}) \leq N \leq \tilde{o}(\exp(\alpha_0^{-2}))$, respectively. Suppose that the initialization $\boldsymbol{\theta}(0)$ is balanced, $\epsilon$-small (Assumption 1) with $\epsilon = \tilde{\Theta}\left(\alpha_0^{8K}\right)$, and satisfies Assumption 2 with a non-degeneracy gap $\Delta = \Theta(1)$. Then with probability at least $1 - \delta$, the GF dynamics (7) with a sampled balanced dataset $\hat{\mathcal{D}} = \{\boldsymbol{x}_i, y_i\}_{i=1}^{KN}$, starting from $\boldsymbol{\theta}(0)$, has its solution $\boldsymbol{\theta}(t)$ satisfying that: for some $t^* = \tilde{\mathcal{O}}\left(\log\frac{1}{\alpha_0}\right)$ and $T^* = \tilde{\Theta}\left(\log\frac{1}{\alpha_0}\right) + \tilde{\Omega}\left(\alpha_0^{-\min\{p-2,2\}}\right)$ with $[t^*, T^*] \neq \emptyset$, we have $\mathcal{L}(\boldsymbol{\theta}(t)) = \tilde{\mathcal{O}}(\alpha_0^4), \forall t \in [t^*, T^*]$, and*

$$\sup_{t\in[t^*,T^*]} \sup_{\boldsymbol{x}\in\mathbb{S}^{D-1}} \left| f^{(p)}(\boldsymbol{x}; \boldsymbol{\theta}(t)) - F^{(p)}(\boldsymbol{x}) \right| \leq \tilde{\mathcal{O}}(\alpha_0^2). \quad (12)$$

$\tilde{\Omega}, \tilde{o}, \tilde{\mathcal{O}}$ hide logarithmic factor $\log\frac{K}{\delta}$ and constant factors that depend on $p$ (in the worst case, $2^p$). Theorem 1 formally proves the conjecture in Min & Vidal (2024), showing that GF on pReLU networks finds a robust classifier for orthonormal Gaussian mixture. Notably, the smaller the intra-cluster variance $\alpha_0$ is, the closer the network $f^{(p)}(\cdot; \boldsymbol{\theta})$ is to $F^{(p)}$, which is observed numerically in Min & Vidal (2024).

We organize the subsequent discussions as follows: First, we state several remarks on understanding our main result and comparing it with prior work; Then we conclude with technical discussions on its proof sketch in Section 3.2.3.

**Nearly optimal robust classifier via GF**. The major implication of Theorem 1 is that one can find a nearly optimal $\ell_2$-robust classifier by GF without adversarial examples. When the intra-cluster variance $\alpha^2$ is small, along the GF trajectory there exists a $f^{(p)}(\cdot; \boldsymbol{\theta}(t))$ that is $\tilde{\mathcal{O}}(\alpha_0^2)$ close to a nearly optimal $\ell_2$-robust classifier $F^{(p)}$, and such proximity to $F^{(p)}$ implies the same level of $\ell_2$-robustness, as shown in Proposition 3. We can immediately conclude that $f^{(p)}$ is also nearly optimal $\ell_2$-robust:

**Corollary 1** (Nearly optimal $\ell_2$-robustness). *Given any* $f^{(p)}(\cdot; \boldsymbol{\theta}(t))$ *obtained at* $t \in [t^*, T^*]$ *from Theorem 1, it can defend against adversarial attacks of radius* $\frac{\sqrt{2}}{2} - \tilde{\mathcal{O}}\big(\alpha_0^{\frac{2}{p}}\big)$ *with probability* $1 - \tilde{\mathcal{O}}(\exp(-\alpha_0^{-2}))$ *over a new sample* $(\boldsymbol{x}, y) \sim \mathcal{D}_{X,Y}$, *thus nearly optimal* $\ell_2$-*robust for* $\mathcal{D}_{X,Y}$.

**Comparison with prior work: robust classifier for orthonormal Gaussian mixture**. The problem of finding a robust classifier for data drawn from orthogonal cluster centers was initiated by Frei et al. (2023), who show theoretically that any classifier obtained by gradient descent on a two-layer ReLU network is susceptible to an adversarial attack of $\ell_2$-radius $\mathcal{O}\big(\frac{1}{\sqrt{K}}\big)$, even though one can easily construct a ReLU network that is robust to attacks of radius $\Theta(1)$[4]. Then Min & Vidal (2024) explain this non-robustness issue with ReLU networks from a neural alignment perspective (Maennel et al., 2018; Boursier & Flammarion, 2024). They propose to replace the ReLU network by the pReLU, $f^{(p)}(\cdot; \boldsymbol{\theta})$, conjecture that training the pReLU with samples from $\mathcal{D}_{X,Y}$ leads to a classifier that is close to $F^{(p)}$ when the intra-cluster variance $\alpha^2$ is small, and provide empirical validation for their conjecture. Our work takes one step further to theoretically prove the convergence to $F^{(p)}$ under GF, and also show that the achieved $\ell_2$-robustness is nearly optimal. Also, a small initialization is critical for finding a robust classifier: since our data is approximately low-dimensional, adversarial examples exist if the initialization scale is large (Melamed et al., 2024), which is verified by our numerical results in Appendix B.2.

**Comparison with prior work: GF with small initialization**. Over the past year, gradient descent/flow with small initialization has been studied for both linear networks (Gunasekar et al., 2017; 2018; Arora et al., 2019a; Woodworth et al., 2020; Gidel et al., 2019; Jacot et al., 2021; Stöger & Soltanolkotabi, 2021; Jin et al., 2023; Xu et al., 2025) and nonlinear networks (Maennel et al., 2018; Phuong & Lampert, 2021; Boursier et al., 2022; Kumar & Haupt, 2024; Chistikov et al., 2023; Wang & Ma, 2023; Min et al., 2024; Tsoy & Konstantinov, 2024; Zhu et al., 2025), to understand the implicit bias of gradient descent algorithms towards structurally simple networks, as opposed to those in the

large initialization regime (Jacot et al., 2018; Chizat et al., 2019; Lee et al.; Woodworth et al., 2020; Luo et al., 2021; Min et al., 2021; Du et al., 2019; Arora et al., 2019b). Our analysis follows the line of work on two-layer ReLU networks, as we will explain in Section 3.2.3 in detail, and improves upon it by considering a more complicated dataset. Specifically, the GF on two-layer ReLU networks has been studied for orthogonally separable data (Phuong & Lampert, 2021; Min et al., 2024; Chistikov et al., 2023), that is, data with the same (different) label has positive (negative) correlation, for mutually orthogonal data (Boursier et al., 2022), and for positively correlated data (but only with two data points) (Wang & Ma, 2023). Our data assumption is closest to mutually orthogonal data (Boursier et al., 2022), but considers a non-zero intra-cluster variance. In addition, we consider a pReLU activation function, whose inductive bias is significantly different than that of ReLU (We elaborate this point in Section 3.2.3).

**Limitations of our current result**. One of the weaknesses of Theorem 1 is that random initialization does not satisfy Assumption 2 (Nonetheless, we experimentally verify in Appendix B.1 that GF/GD still converges to the robust classifier from random initialization): Since the non-degeneracy gap is defined by inner products between neurons and cluster centers, one can show that under random initialization, the non-degeneracy gap is $\Delta = \mathcal{O}\big(\frac{1}{\sqrt{D}}\big)$ with high probability, while the theorem requires a $\Theta(1)$ gap. We have discussed this issue when we define non-degenerate initialization in Section 3.2.1: When neurons are initialized close to the boundary between one Voronoi region $\mathcal{R}_k$ and another $\mathcal{R}_l$ (which is the case under random initialization), whether they align with $\boldsymbol{\mu}_k$ or with $\boldsymbol{\mu}_l$ will depend on the points actually sampled in the data set. In this regard, at the very initial stage of GF there is a "burn-in" phase during which neurons "choose" their target clusters depending on the samples. Once the neurons have moved away from the boundary of these Voronoi regions with gap $\Delta = \Theta(1)$, we characterize the GF dynamics by Theorem 1. In Appendix A, we elaborate on why this "burn-in" phase is challenging to analyze and discuss several additional technical limitations.

**Remark 2.** *The goal of this paper is to understand theoretically the problem of learning a robust classifier for orthonormal Gaussian mixture via GF. Thus, we focus on proving rigorous theorems and discussing the associated technical challenges. For the practical implications of this problem, we refer the reader to* Min & Vidal (2024); Li et al. (2025), *where they empirically validate that in some transfer learning settings, the training problem is related to learning robust classifiers for orthonormal Gaussian mixture. Formally establishing these connections is an interesting future research direction.*

---

[4]Frei et al. (2023) consider data from $\mathcal{N}(\sqrt{D}\boldsymbol{\mu}_k, \alpha^2 \boldsymbol{I})$, thus their results should be rescaled by $\frac{1}{\sqrt{D}}$ when applied to $\mathcal{D}_{X,Y}$.

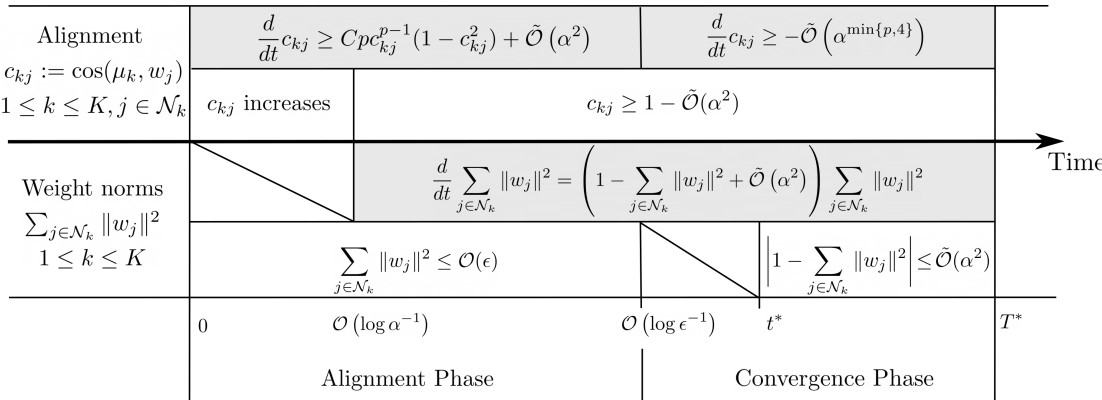

*Figure 4.* Important quantities (alignment and weight norms) and their dynamics long GF trajectory

### 3.2.3. PROOF SKETCH

For simplicity, we consider the case $\alpha^2 = \alpha_0^2$ and use $\alpha^2$ throughout this section. The discussion is conditioned on a good event (has at least $1 - \delta$ probability) when samples are concentrated around their respective cluster centers.

**Overall proof**. Our proof in spirit is close to that of Boursier et al. (2022), with a two-phase analysis of the GF focusing on different quantities. Specifically, at the initial phase, called *alignment phase*, one studied the dynamics of the neuron direction $\frac{w_j}{\|w_j\|}$ through cosine angles between $w_j$ and cluster center $\mu_k$, where one shows that, for all $k$ and $j \in \mathcal{N}_k$, we have that $c_{kj} := \cos(\mu_k, w_j)$ monotonically increases until it reaches $1 - \tilde{\mathcal{O}}(\alpha^2)$. That is, as we mentioned earlier, neurons initialized within $\mathcal{R}_k$ converge in angle to the corresponding $\mu_k$. Then in the second *convergence phase*, we show that all $c_{kj}$ can probably stay above $1 - \tilde{\mathcal{O}}(\alpha^2)$ until $T^*$, and in the meantime, the norm of the neurons (measured by $\sum_{j \in \mathcal{N}_k} \|w_j\|^2$ for each $k$) monotonically grows until it reaches $1 \pm \tilde{\mathcal{O}}(\alpha^2)$ before $t^*$. Moreover, the norm of the neurons initialized in the void region stays small: $\sum_{j \in \mathcal{N}_c} \|w_j(t)\| = \tilde{o}(\alpha^2)$. These three conditions $c_{kj} \geq 1 - \tilde{\mathcal{O}}(\alpha^2)$, $\left|1 - \sum_{j \in \mathcal{N}_k} \|w_j\|^2\right| \leq \tilde{\mathcal{O}}(\alpha^2)$ and $\sum_{j \in \mathcal{N}_c} \|w_j\| = \tilde{o}(\alpha^2)$ together imply the desired bound between $f^{(p)}$ and $F^{(p)}$. See Figure 4 for an illustration.

**Alignment phase**. (See Appendix F) The alignment phase is the time interval between $0$ and some $\tilde{\mathcal{O}}\left(\log \frac{1}{\epsilon}\right)$ time, where $\epsilon$ is the initialization scale. During the alignment phase, the norms of the weights stays $\tilde{\mathcal{O}}(\epsilon)$ small, which follows a similar proof in Boursier et al. (2022); Min et al. (2024). The small norm of the weights, together with the positive non-degeneracy gap assumption, allows for the following characterization of the alignment for $1 \leq k \leq K, j \in \mathcal{N}_k$:

$$\frac{d}{dt}c_{kj} \geq Cpc_{kj}^{p-1}(1 - c_{kj}^2)$$
$$+ \tilde{\mathcal{O}}\left(\frac{\alpha}{\sqrt{N}} + \frac{\alpha}{\sqrt{D}}\right) + \tilde{\mathcal{O}}\left(\alpha^2 + \frac{\alpha}{\sqrt{D}}\right) + \tilde{\mathcal{O}}(\epsilon), \quad (13)$$

for some constant $C > 0$. Consider the case when $\alpha = 0$, and $\epsilon \to 0$, for $j \in \mathcal{N}_k$, the dynamics $\frac{d}{dt}c_{kj} \geq Cpc_{kj}^{p-1}(1 - c_{kj}^2)$ characterize the nominal effect of cluster centers $\mu_k, 1 \leq k \leq K$ on neuron direction $\frac{w_j}{\|w_j\|}$: each cluster centers is either attracting or repelling $\frac{w_j}{\|w_j\|}$, depending on whether their label matches the sign of $v_j$, and the aggregate effect is pushing $\frac{w_j}{\|w_j\|}$ towards $\mu_k$, the closest cluster center to $w_j$ in angle at initialization. We call the $k$-th cluster the target cluster for $w_j$.

The rest of the terms are considered perturbations due to noisy samples around cluster centers and a non-zero initialization scale: The first $\tilde{\mathcal{O}}(\alpha/\sqrt{N} + \alpha/\sqrt{D})$ term is due to the noisy samples from (the target) $k$-th cluster. Since we have a $\Delta = \Theta(1)$ non-degeneracy gap, $w_j$ has a positive inner product with every sampled data within the $k$-th cluster, then one can utilize concentration results to bound the effect of noise. The second $\tilde{\mathcal{O}}(\alpha^2 + \alpha/\sqrt{D})$ term is due to the noisy samples from other non-target clusters. Unfortunately, we have no control over how many of them have positive inner products with $w_j$, thus a worse bound $\tilde{\mathcal{O}}(\alpha^2)$ is derived. Lastly, $\tilde{\mathcal{O}}(\epsilon)$ is due to an $\epsilon$-small weight norm. With $N = \tilde{\Omega}(\alpha^{-2})$ samples, $D = \tilde{\Omega}(\alpha^{-2})$ dimension, and small $\epsilon$, the dominant terms become $\tilde{\mathcal{O}}(\alpha^2)$, allowing us to prove:

**Proposition 4** (Alignment in pReLU network). *Given the same assumptions as in Theorem 1 and consider the same GF solution $\theta(t), t \geq 0$. There exist some $t_1 = \mathcal{O}\left(\log \frac{1}{\alpha}\right)$ and $t_2 = \mathcal{O}\left(\log \frac{1}{\epsilon}\right)$ such that $\forall k$ and $\forall j \in \mathcal{N}_k$, $\cos(\mu_k, w_j(t)) \geq 1 - \tilde{\mathcal{O}}(\alpha^2), \forall t \in [t_1, t_2]$.*

We explicitly state the result during the alignment phase in Proposition 4 to highlight the difference between its described alignment for pReLU network ($p > 2$) to that of Boursier et al. (2022) for ReLU networks, where neurons are aligned with class average $\mu_+ = \sum_{1 \leq k \leq K_1} \mu_k$ and $\mu_- = \sum_{K_1+1 \leq k \leq K} \mu_k$ instead of cluster centers.

**Convergence phase**. (See Appendix G) During the convergence phase, the weight norm grows and exceeds $\epsilon$-level, as

suggested by the following dynamics:

$$\frac{d}{dt}\sum_{j\in\mathcal{N}_k}\|\boldsymbol{w}_j\|^2 = \left(1 - \sum_{j\in\mathcal{N}_k}\|\boldsymbol{w}_j\|^2\right)\sum_{j\in\mathcal{N}_k}\|\boldsymbol{w}_j\|^2$$
$$+\left(\tilde{\mathcal{O}}\left(\frac{\alpha}{\sqrt{N}}\right)+\tilde{\mathcal{O}}\left(\alpha^2\right)+\tilde{\mathcal{O}}\left(\alpha^p\right)\right)\sum_{j\in\mathcal{N}_k}\|\boldsymbol{w}_j\|^2, \quad (14)$$

which holds when $c_{kj} \geq 1 - \tilde{\mathcal{O}}(\alpha^2), \forall k, j \in \mathcal{N}_k$. $\frac{d}{dt}\sum_{j\in\mathcal{N}_k}\|\boldsymbol{w}_j\|^2 = \left(1 - \sum_{j\in\mathcal{N}_k}\|\boldsymbol{w}_j\|^2\right)\sum_{j\in\mathcal{N}_k}\|\boldsymbol{w}_j\|^2$ (nominal dynamics) describes the weight growth if $c_{kj} = 1, \forall k, j \in \mathcal{N}_k$ and $\alpha = 0$. Following nominal dynamics, $\sum_{j\in\mathcal{N}_k}\|\boldsymbol{w}_j\|^2$ converges to 1 for every $k$.

The rest of the terms are considered perturbations due to noisy samples around cluster centers and the fact that alignment $c_{kj}$ are only close to 1. The first $\tilde{\mathcal{O}}(\alpha/\sqrt{N})$ is due to the noisy sample from the target $k$-th cluster, the second $\tilde{\mathcal{O}}(\alpha^2)$ term is from imperfect alignment $c_{kj} \geq 1 - \tilde{\mathcal{O}}(\alpha^2)$, and the last $\tilde{\mathcal{O}}(\alpha^p)$ term is from the noisy sample from the non-target clusters (Notice that now $\boldsymbol{w}_j$s are almost orthogonal to non-target clusters, thus the effect of non-target clusters is smaller). With $N = \tilde{\Omega}(\alpha^{-2})$ samples, the dominant terms become $\tilde{\mathcal{O}}(\alpha^2)$, allowing us to show that $\sum_{j\in\mathcal{N}_k}\|\boldsymbol{w}_j\|^2$ converges to $1 \pm \tilde{\mathcal{O}}(\alpha^2)$ within $t^*$ time.

The only missing piece is that this argument requires $c_{kj} \geq 1 - \tilde{\mathcal{O}}(\alpha^2), \forall k, j \in \mathcal{N}_k$ but one no longer has (13) after $\tilde{\Theta}\left(\log\frac{1}{\epsilon}\right)$ when weight norm starts to grow to $\tilde{\Theta}(1)$-level. Nonetheless, once the alignment $c_{kj}$ is $1 - \tilde{\mathcal{O}}(\alpha^2)$, it is hard to drop below this level as it relies on the attraction from non-target clusters but they are now near orthogonal to the neurons. Indeed, during the convergence phase, we can show that $\frac{d}{dt}c_{kj} \geq -\tilde{\mathcal{O}}(\alpha^{\min\{p,4\}})$, by which we show $c_{kj}$ can stay at $1 - \tilde{\mathcal{O}}(\alpha^2)$ level until $T^*$ time. Since $T^* \geq t^*$ for small $\alpha$, our analysis of the weight norm growth is valid.

## 4. Conclusion

In this paper, we investigated the problem of learning robust classifiers for orthonormal GMMs. While constructing the optimal robust classifiers is straightforward if the distribution is known, the challenge arises, however, when a neural network is trained by gradient-based optimization algorithms with sampled data. We showed how a careful choice of activation function provably addresses the non-robustness issue of the network trained via gradient flow, highlighting how the implicit bias of training algorithms can critically guide the network to learn accurately the data structure, thereby achieving maximum robustness against adversarial attacks. Future research includes addressing the limitations in the current convergence analyses, generalizing the data assumptions to more realistic ones, for example, the union of subspaces.

## Acknowledgments

The authors acknowledge the support of the NSF under grants 2031985 and 2212457, the Simons Foundation under grant 814201, and the ONR MURI Program under grant 503405-78051. H. Min acknowledges the support of the DDDI and the IDEAS at UPenn under the AI x Science Fellowship.

## Impact Statement

In this theoretical paper, there is no potential societal consequence which we feel must be specifically highlighted here.

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

## A. Technical limitations of current results

We discuss here several technical limitations of our current results and potential avenues to address them. These limitations are listed in an order that the most challenging ones are stated first.

**Requirement on the initialization**. The initialization requires a non-degeneracy gap $\Delta = \Theta(1)$, this assumption aims to alleviate the challenges to analyze the activation patterns $\{\mathbb{1}_{\langle \boldsymbol{x}_i, \boldsymbol{\mu}_j \rangle \geq 0}\}$ along the GF trajectory. The activation patterns critically determine the gradient due to the existence of ReLU activations in the network. Generally, as a set of binary states in the GF dynamics, their evolution is hard to track unless the underlying training data has some special property, for example, (Boursier et al., 2022) assumes mutually orthogonal data, in which case the activation patterns remain constant throughout GF, and (Phuong & Lampert, 2021; Min et al., 2024) assume orthogonally separable data, in which case the activation can only flip in one direction ("$0 \rightarrow 1$" or "$1 \rightarrow 0$") for each neuron. Due to relatively more complicated data assumptions and the polynomial ReLU activations, there is no good technique to track the activation as of the current analysis. Therefore, we pose the non-degeneracy gap $\Delta = \Theta(1)$ assumption on the initialization, under which each neuron activates all the data points from its target cluster and none of the others, and the activation patterns are fixed thereafter. Future research on careful analysis of the activation pattern evolution can relax this initialization assumption.

As we have discussed after Theorem 1, this non-degeneracy gap $\Delta = \Theta(1)$ generally cannot be achieved by random initialization: the cosines between neurons and cluster centers are $\mathcal{O}\left(\frac{1}{\sqrt{D}}\right)$ with high probability. Given that $D = \tilde{\Omega}(\alpha^{-2})$, the actually non-degeneracy gap of a random initialization is $\tilde{\mathcal{O}}(\alpha)$. We have discussed this issue when we define non-degenerate initialization in Section 3.2.1: When neurons are initialized close to the boundary between a Voronoi region $\mathcal{R}_k$ and another region $\mathcal{R}_l$, whether they align with $\boldsymbol{\mu}_k$ or with $\boldsymbol{\mu}_l$ depends on the actually sampled points in the dataset. In this regard, when weights are randomly initialized, there is a "burn-in" phase during which neurons "choose" their target clusters depending on the samples, then once they get away from the boundary of these Voronoi regions with $\Delta = \Theta(1)$ gap, we can characterize the GF dynamics afterward by Theorem 1.

**Upper bound on $N$**. Regarding our requirement $\tilde{\Omega}(\alpha_0^{-2}) \leq N \leq \tilde{o}(\exp(\alpha_0^{-2}))$, we have discussed the lower bound $N \geq \tilde{\Omega}(\alpha_0^{-2})$ in Section 3.2.3. In fact, one can remove this lower bound and get a final bound $\tilde{\mathcal{O}}\left(\frac{\alpha_0}{\sqrt{N}}\right)$ in Theorem 1. The upper bound $N \leq \tilde{o}(\exp(\alpha_0^{-2}))$ may seem puzzling. This issue originates from ReLU nonlinearity: a data point must activate a neuron by having a positive inner product. Our analysis requires that a neuron $\boldsymbol{w}_j$ is activated by every data point from its target cluster, which is translated into two conditions: 1) $\Theta(1)$ non-degeneracy gap; and 2) $\sqrt{\log N} \alpha_0 = \tilde{o}(1)$. Here $\sqrt{\log N} \alpha_0$ is essentially the radius of a $\ell_2$-ball centered at a cluster center that can contain all the sampled points from that cluster with high probability. Without these conditions, there will be outliers in sampled points, which must be handled with extra analysis. We believe this is possible because those outliers will be rare and thus may have a negligible effect on the dynamics.

**Extension to classification losses**. Our results for the alignment phase directly apply to classification losses: The choice of the loss $\ell(y, \hat{y})$ only affects the alignment dynamics through $\nabla_{\hat{y}} \ell(y, \hat{y})|_{\hat{y}=0}$, and this quantity is same (may up to a constant scaling) regardless of whether $\ell$ is exponential, logistic, or $\ell_2$. However, the analysis of convergence phase critically depends on $\ell$: Recall that in Section 3.2.3 we show that the nominal weight norm dynamics are $\dot{z} = (1 - z)z, z = \sum_{j \in \mathcal{N}_k} \|\boldsymbol{w}_j\|^2$ for $\ell_2$ loss. For exponential loss, the nominal dynamics become $\dot{z} = \exp(-z)z$, whose closed-form solution is not available. A better characterization of the solution to the nominal dynamics of the type $\dot{z} = \exp(-z)z$ in future research naturally leads to an extension of Theorem 1 to classification losses.

**Analysis until finite time $T^*$**. Our focus is on the distance between $f^{(p)}(\cdot; \boldsymbol{\theta}(t))$ and $F^{(p)}$, thus we restrict to the time interval $[0, T^*]$ when we have explicit control of all relevant quantities (alignment, weight norms, etc.). To show convergence towards a minimizer of the loss after $T^*$, we believe applying the results in Chatterjee (2022) suffices, following the approach in Boursier et al. (2022).

# B. Additional Experiments on Learning Robust Classifiers for Data from Orthonormal GMMs

## B.1. Experiments on the role of non-degenerate initialization shape

In this section, we conduct experiments to show the difference between random initialization and non-degenerate initialization, with the main purpose being to verify the statements in our discussions **Limitation of our current result** and **Requirement on the initialization** that training with random initialization, while not satisfying our non-degeneracy assumption, still find the desired robust network.

**Experiment settings**. We consider two types of initialization shapes (The initialization scale is $\epsilon = 10^{-7}$):

- *Random initialization*: the initialization follows Assumption 1, with entries of initialization shape $\boldsymbol{w}_{j0}, j = 1, \cdots, h$ all i.i.d. samples from standard Gaussian. As we discussed, this does not satisfy our Assumption 2 in high-dimensional scenarios.

- *Non-degenerate initialization*: We nudge the above random initialization shape toward cluster centers to increase its non-degeneracy gap. Specifically, for every $j$, we let $\boldsymbol{w}_{j0} \leftarrow \boldsymbol{w}_{j0} + \delta \cdot (\boldsymbol{\mu} - \boldsymbol{w}_{j0})$, where $\boldsymbol{\mu}$ is randomly uniformly selected from one of the cluster centers $\boldsymbol{\mu}_k, k = 1, \cdots, K$. The resulting new initialization shape has a non-degeneracy gap of roughly $\delta$, thus satisfying Assumption 2. We also adjust the $|v_j|$ accordingly so the initialization satisfies Assumption 1.

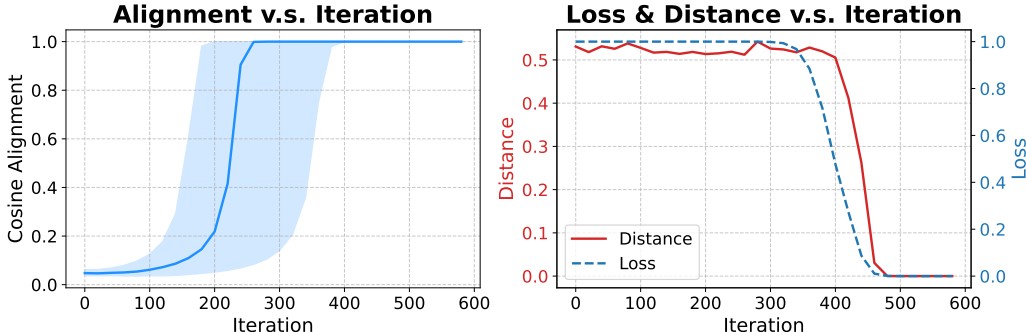

*Figure 5.* GD on two-layer pReLU ($p = 3$) with random, small initialization

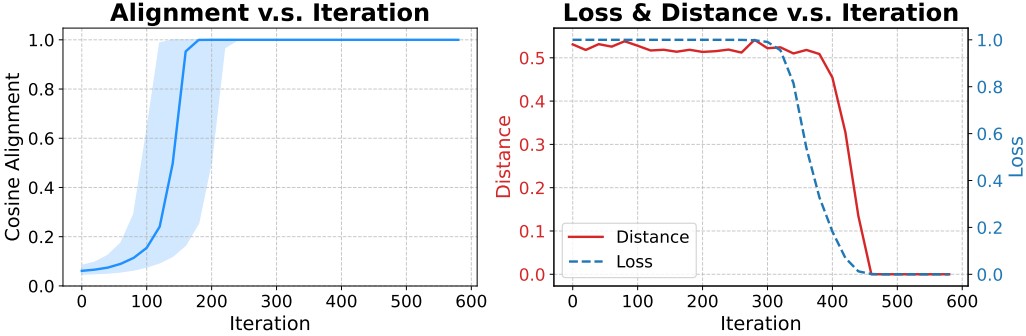

*Figure 6.* GD on two-layer pReLU ($p = 3$) with non-degenerate ($\delta = 0.05$), small initialization

After the initialization is determined, we run GD with step size $0.2$ on a synthetic GMM dataset of size $n = 5000$ with $D = 1000, K_1 = 5, K_2 = 5, \alpha = 0.1$, and keep track of the following:

- *Alignment*: The alignment measures we are interested in are $\max_k \cos(\boldsymbol{\mu}_k, \boldsymbol{w}_j), j = 1, \cdots, h$, and our Theorem 1 and its proof suggest that they converge to close to 1 after sufficient training time. For clarity, we report their median (lines), the 1st and 3rd quantiles (shaded regions).

- *Loss*: The mean square loss.

- *Distantance to $F^{(p)}$*: The quantity $\sup_{\boldsymbol{x} \in S^{D-1}} |f^{(p)}(\boldsymbol{x}; \boldsymbol{\theta}) - F^{(p)}(\boldsymbol{x})|$. We estimate this quantity by randomly sampling large batches of $x$ and running projected gradient ascent on this distance, similar to (Min & Vidal, 2024).

**Experimental results**. We plot the results in Figures 5 and 6. First of all, Figure 6 shows the convergence of GD under non-degenerate initialization (in both the loss and the distance to $F^{(p)}$). Then the same convergence happens under random initialization. However, the alignment between neurons and cluster centers is slower for random initialization than with the nudged initialization with $\delta = 0.05$ non-degeneracy gap. This agrees with our discussions in **Limitation of our current result** and **Requirement on the initialization**: having some non-degeneracy gap skips a "burn-in" phase for the neurons' directional dynamics.

### B.2. Experiments on the role of parametrization and initialization scale

In this section, we provide additional experiments to that in Figure 2, highlighting the importance of parametrization of function space, and the hyperparameters of training algorithm in determining whether one can succeed in obtaining robust classifier for data from orthonormal Gaussian mixture.

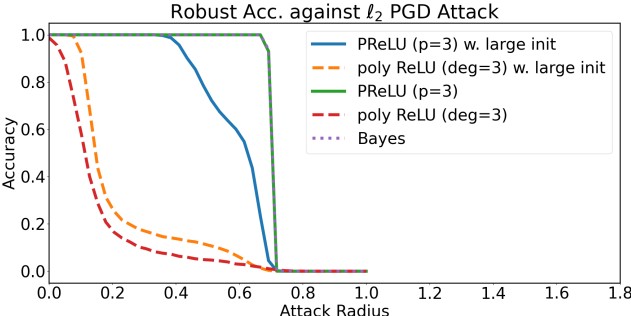

*Figure 7.* Given balanced orthonormal Gaussian mixture data with 12 positive clusters and 8 negative clusters ($D = 2000$), gradient descent (SGD, small initialization) on (bias-free, width-200) two-layer network with regular polynomial ReLU activation of degree 3 fails to find a robust classifier. Moreover, if one increases the variance of the random initialization, both regular polynomial ReLU network and pReLU network can not find a robust classifier. All networks here are trained for a sufficient amount of epochs until they achieve perfect training accuracy on a synthesis orthonormal Gaussian mixture dataset of size 20000.

**Regular polynomial ReLU networks**. In this experiment, we consider both the regular polynomial ReLU networks to pReLU networks. In particular, recall that the regular polynomial ReLU networks are defined as:

$$g(\boldsymbol{x}; \tilde{\boldsymbol{\theta}}) = \sum_{j=1}^{h} v_j \sigma^p(\langle \boldsymbol{x}, \boldsymbol{w}_j \rangle), \qquad (\tilde{\boldsymbol{\theta}} := \{\boldsymbol{w}_j, v_j\}_{j=1}^{h}).$$

(Two-layer Networks with Polynomial ReLU activation with degree $p$)

We note its difference with pReLU networks: regular polynomial ReLU networks do not have a weight normalization at the first layer. Nonetheless, when $p$ is fixed, it is easy to verify that the function/hypothesis spaces induced by pReLU networks and regular polynomial ReLU networks are the same: any function $f^{(p)}(\boldsymbol{x}; \boldsymbol{\theta})$ for some $\boldsymbol{\theta} = \{\boldsymbol{w}_j, v_j\}_{j=1}^{h}$ is equivalent to $g(\boldsymbol{x}; \tilde{\boldsymbol{\theta}})$ with $\tilde{\boldsymbol{\theta}} = \{\boldsymbol{w}_j, \frac{v_j}{\|\boldsymbol{w}_j\|^{p-1}}\}_{j=1}^{h}$[5].

**Regular polynomial ReLU networks v.s. pReLU**. Although the induced function/hypothesis spaces are the same, GD on regular polynomial ReLU networks and pReLU finds classifiers with different levels of robustness. As one can see in Figure 7, with a small initialization (all weight entries are randomly initialized as $\mathcal{N}(0, 1 \times 10^{-4})$), SGD on a pReLU network successfully finds a classifier that is as robust as the Bayes classifier. However, SGD on a regular polynomial ReLU network fails to find a robust classifier. This suggests that the way the function/hypothesis spaces are parametrized is also important in determining the robustness of the networks trained by GD, as different parametrization induces different implicit biases of GD in selecting the loss minimizer in the function space.

**Effect of initialization scale**. Finally, when one uses a large initialization scale, where all weight entries are randomly initialized as $\mathcal{N}(0, 0.25)$, even the GD on a pReLU network fails to find a robust classifier. This is not surprising as the initialization scale also controls the implicit bias of GD (Moroshko et al., 2020), and many works (Maennel et al., 2018; Stöger & Soltanolkotabi, 2021; Li et al., 2018; 2021) have theoretically shown the advantage of using a small initialization scale in GD.

---

[5]The neurons with $\|\boldsymbol{w}_j\| = 0$ should be eliminated from the parameters for this argument to hold.

## C. Optimal Robust Classifier for orthonormal Gaussian mixture

In this section, we discuss the optimal robust classifier for orthonormal Gaussian mixture. We first show that any measurable classifier can not defend against an adversarial attack of $\ell_2$ radius $\frac{\sqrt{2}}{2}$, leading to a robust error of at least $\frac{\min\{K_1, K_2\}}{K}$. Then we consider the Bayes optimal classifier $f^*(\boldsymbol{x}) = \arg\max_y \mathbb{P}\left(Y = y | \boldsymbol{x}\right)$ and show that it is also optimally robust: it can defend against any adversarial attack of $\ell_2$ radius $\frac{\sqrt{2}}{2} - o(1)$, as the dimension of the data $D$ increases.

### C.1. Maximum robustness against $\ell_2$ adversarial attacks

We need the following lemma (we provide proof after proving Proposition 1)

**Lemma 1.** *For any $n \times m$ matrix, let $a$ be the number of rows that contain at least one non-positive entry and $b$ be the number of columns that contain at least one non-negative entry. Then $a + b \geq \min\{n, m\}$.*

With Lemma 1, we are ready to prove Proposition 1.

**Proposition 1** (Restated). *Let $f : \mathbb{R}^D \to \mathbb{R}$ be any Lebesgue measurable function such that the random variable $\min_{\|\boldsymbol{d}\| \leq 1}\left[f\left(\boldsymbol{x} + \frac{\sqrt{2}}{2}\boldsymbol{d}\right) y\right]$ is also measurable. Given a sample $(\boldsymbol{x}, y) \sim \mathcal{D}_{X,Y}$, we have*

$$\mathbb{P}\left(\min_{\|\boldsymbol{d}\| \leq 1}\left[f\left(\boldsymbol{x} + \frac{\sqrt{2}}{2}\boldsymbol{d}\right) y\right] \leq 0\right) \geq \frac{\min\{K_1, K_2\}}{K} \,. \tag{C.1}$$

*Proof.* We start with the following:

$$\mathbb{P}\left(\min_{\|\boldsymbol{d}\| \leq 1}\left[f\left(\boldsymbol{x} + \frac{\sqrt{2}}{2}\boldsymbol{d}\right) y\right] \leq 0\right) = \sum_{k=1}^{K} \mathbb{P}\left(\min_{\|\boldsymbol{d}\| \leq 1}\left[f\left(\boldsymbol{x} + \frac{\sqrt{2}}{2}\boldsymbol{d}\right) y\right] \leq 0 \mid z = k\right) \mathbb{P}\left(z = k\right) \tag{C.2}$$

For $k \leq K_1$,

$$\mathbb{P}\left(\min_{\|\boldsymbol{d}\| \leq 1}\left[f\left(\boldsymbol{x} + \frac{\sqrt{2}}{2}\boldsymbol{d}\right) y\right] \leq 0 \mid z = k\right) = \mathbb{P}_{\boldsymbol{\varepsilon}}\left(\min_{\|\boldsymbol{d}\| \leq 1}\left[f\left(\boldsymbol{\mu}_k + \boldsymbol{\varepsilon} + \frac{\sqrt{2}}{2}\boldsymbol{d}\right)\right] \leq 0\right)$$

$$\geq \mathbb{P}_{\boldsymbol{\varepsilon}}\left(\min_{K_1 + 1 \leq l \leq K}\left[f\left(\frac{\boldsymbol{\mu}_k + \boldsymbol{\mu}_l}{2} + \boldsymbol{\varepsilon}\right)\right] \leq 0\right) \,.$$

The measurability of $f$ ensures this lower bound exists. Similarly, we have for $K_1 + 1 \leq k \leq K$

$$\mathbb{P}\left(\min_{\|\boldsymbol{d}\| \leq 1}\left[f\left(\boldsymbol{x} + \frac{\sqrt{2}}{2}\boldsymbol{d}\right) y\right] \leq 0 \mid z = k\right) = \mathbb{P}_{\boldsymbol{\varepsilon}}\left(\min_{\|\boldsymbol{d}\| \leq 1}\left[-f\left(\boldsymbol{\mu}_k + \boldsymbol{\varepsilon} + \frac{\sqrt{2}}{2}\boldsymbol{d}\right)\right] \leq 0\right)$$

$$\geq \mathbb{P}_{\boldsymbol{\varepsilon}}\left(\min_{1 \leq l \leq K_1}\left[-f\left(\frac{\boldsymbol{\mu}_k + \boldsymbol{\mu}_l}{2} + \boldsymbol{\varepsilon}\right)\right] \leq 0\right)$$

$$= \mathbb{P}_{\boldsymbol{\varepsilon}}\left(\max_{1 \leq l \leq K_1}\left[f\left(\frac{\boldsymbol{\mu}_k + \boldsymbol{\mu}_l}{2} + \boldsymbol{\varepsilon}\right)\right] \geq 0\right) \,.$$

Therefore,

$$\mathbb{P}\left(\min_{\|\boldsymbol{d}\| \leq 1}\left[f\left(\boldsymbol{x} + \frac{\sqrt{2}}{2}\boldsymbol{d}\right) y\right] \leq 0\right)$$

$$= \sum_{k=1}^{K} \mathbb{P}\left(\min_{\|\boldsymbol{d}\| \leq 1}\left[f\left(\boldsymbol{x} + \frac{\sqrt{2}}{2}\boldsymbol{d}\right) y\right] \leq 0 \mid z = k\right) \mathbb{P}\left(z = k\right)$$

$$= \frac{1}{K}\left[\sum_{1 \leq k \leq K_1} \mathbb{P}\left(\min_{\|\boldsymbol{d}\| \leq 1}\left[f\left(\boldsymbol{x} + \frac{\sqrt{2}}{2}\boldsymbol{d}\right) y\right] \leq 0 \mid z = k\right)\right.$$

$$+ \sum_{K_1+1\leq k\leq K} \mathbb{P}\left(\min_{\|\boldsymbol{d}\|\leq 1}\left[f\left(\boldsymbol{x}+\frac{\sqrt{2}}{2}\boldsymbol{d}\right)y\right]\leq 0 \mid z=k\right)\right]$$

$$\geq \frac{1}{K}\left[\sum_{1\leq k\leq K_1}\mathbb{P}_{\boldsymbol{\varepsilon}}\left(\min_{K_1+1\leq l\leq K}\left[f\left(\frac{\boldsymbol{\mu}_k+\boldsymbol{\mu}_l}{2}+\boldsymbol{\varepsilon}\right)\right]\leq 0\right)\right.$$

$$+ \sum_{K_1+1\leq k\leq K}\mathbb{P}_{\boldsymbol{\varepsilon}}\left(\max_{1\leq l\leq K_1}\left[f\left(\frac{\boldsymbol{\mu}_k+\boldsymbol{\mu}_l}{2}+\boldsymbol{\varepsilon}\right)\right]\geq 0\right)\right]$$

$$= \frac{1}{K}\left[\sum_{1\leq k\leq K_1}\int \mathbb{1}\left(\min_{K_1+1\leq l\leq K}\left[f\left(\frac{\boldsymbol{\mu}_k+\boldsymbol{\mu}_l}{2}+\boldsymbol{\varepsilon}\right)\right]\leq 0\right)p(\boldsymbol{\varepsilon})\right.$$

$$+ \sum_{K_1+1\leq k\leq K}\int \mathbb{1}\left(\max_{1\leq l\leq K_1}\left[f\left(\frac{\boldsymbol{\mu}_k+\boldsymbol{\mu}_l}{2}+\boldsymbol{\varepsilon}\right)\right]\geq 0\right)p(\boldsymbol{\varepsilon})\right]$$

$$= \frac{1}{K}\int\left[\sum_{1\leq k\leq K_1}\mathbb{1}\left(\min_{K_1+1\leq l\leq K}\left[f\left(\frac{\boldsymbol{\mu}_k+\boldsymbol{\mu}_l}{2}+\boldsymbol{\varepsilon}\right)\right]\leq 0\right)\right. \tag{C.3}$$

$$\left.+ \sum_{K_1+1\leq k\leq K}\mathbb{1}\left(\max_{1\leq l\leq K_1}\left[f\left(\frac{\boldsymbol{\mu}_k+\boldsymbol{\mu}_l}{2}+\boldsymbol{\varepsilon}\right)\right]\geq 0\right)\right]p(\boldsymbol{\varepsilon}), \tag{C.4}$$

and if we define the $K_1\times K_2$ matrix

$$M_f(\boldsymbol{\varepsilon}) := \left[f\left(\frac{\boldsymbol{\mu}_k+\boldsymbol{\mu}_l}{2}+\boldsymbol{\varepsilon}\right)\right]_{1\leq k\leq K_1,\ K_1+1\leq l\leq K} \tag{C.5}$$

and examine carefully enough, we notice that $\sum_{1\leq k\leq K}\mathbb{1}\left(\min_{K_1+1\leq l\leq K}\left[f\left(\frac{\boldsymbol{\mu}_k+\boldsymbol{\mu}_l}{2}+\boldsymbol{\varepsilon}\right)\right]\leq 0\right)$ is the number of rows of $M_f(\boldsymbol{\varepsilon})$ that contains at least one non-positive entry and $\sum_{K_1+1\leq k\leq K}\mathbb{1}\left(\max_{1\leq l\leq K_1}\left[f\left(\frac{\boldsymbol{\mu}_k+\boldsymbol{\mu}_l}{2}+\boldsymbol{\varepsilon}\right)\right]\geq 0\right)$ is the number of columns of $M_f(\boldsymbol{\varepsilon})$ that contains at least one non-negative entry. By Lemma 1, we have

$$\mathbb{P}\left(\min_{\|\boldsymbol{d}\|\leq 1}\left[f\left(\boldsymbol{x}+\frac{\sqrt{2}}{2}\boldsymbol{d}\right)y\right]\leq 0\right)\geq (\text{C.4})\geq \frac{1}{K}\int \min\{K_1,K_2\}p(\boldsymbol{\varepsilon}).$$

Therefore

$$\mathbb{P}\left(\min_{\|\boldsymbol{d}\|\leq 1}\left[f\left(\boldsymbol{x}+\frac{\sqrt{2}}{2}\boldsymbol{d}\right)y\right]\leq 0\right)\geq \frac{\min\{K_1,K_2\}}{K}. \tag{C.6}$$

$\square$

*Proof of Lemma 1.* We denote $C^*(n,m)$ the minimum value of $a+b$ over all possible choice of $n\times m$ matrices. It suffices to show $C^*(n,m)\geq \min\{n,m\}$ (The equality is obtained by an all-positive matrix when $n\leq m$ and an all-negative matrix otherwise), and we prove it by induction.

For $n=1, m=1, C^*(n,m)=1$. This is trivial. We need to show that if $C^*(n,m)=\min\{n,m\}$ holds for some $n$ and $m$, then

- $C^*(n,m+1)=\min\{n,m+1\}$;

- and $C^*(n+1,m)=\min\{n+1,m\}$.

We shall prove these two cases:

**Case 1.** $C^*(n,m)\geq \min\{n,m\}\Rightarrow C^*(n,m+1)\geq \min\{n,m+1\}$

Given an $n \times m$ matrix $\boldsymbol{M}$ and an agumented matrix $\boldsymbol{M}' = \begin{bmatrix} \boldsymbol{M} & \boldsymbol{v} \end{bmatrix}$, we let $a, b$ and $a', b'$ be the row/column counts of our interest for $\boldsymbol{M}$ and $\boldsymbol{M}'$ respectively. Without loss of generality, we suppose the first $a$ rows of $\boldsymbol{M}$ all contain at least one non-positive entry (and the rest do not, by definition of $a$). We know that $a + b \geq \min\{n, m\}$, and

$$a' = a + \sum_{i=a+1}^{n} \mathbb{1}(\boldsymbol{v}_i \leq 0), \qquad b' = b + \mathbb{1}(\max_i \boldsymbol{v}_i \geq 0), \tag{C.7}$$

which is

$$a' + b' = a + b + \sum_{i=a+1}^{n} \mathbb{1}(\boldsymbol{v}_i \leq 0) + \mathbb{1}(\max_i \boldsymbol{v}_i \geq 0). \tag{C.8}$$

There are two scenarios:

1. When $a = n$, we have $\sum_{i=a+1}^{n} \mathbb{1}(\boldsymbol{v}_i \leq 0) + \mathbb{1}(\max_i \boldsymbol{v}_i \geq 0) \geq 0$

2. When $a < n$, we have $\sum_{i=a+1}^{n} \mathbb{1}(\boldsymbol{v}_i \leq 0) + \mathbb{1}(\max_i \boldsymbol{v}_i \geq 0) \geq 1$.

Therefore, we find that

$$a' + b' \geq \min\{n + b, a + b + 1\} \geq \min\{n, \min\{n, m\} + 1\} = \min\{n, n + 1, m + 1\} = \min\{n, m + 1\}. \tag{C.9}$$

This shows $C^*(n, m + 1) \geq \min\{n, m + 1\}$.

**Case 2**. $C^*(n + 1, m) \geq \min\{n + 1, m\} \Rightarrow C^*(n + 1, m) \geq \min\{n + 1, m\}$

Given an $n \times m$ matrix $\boldsymbol{M}$ and an agumented matrix $\boldsymbol{M}' = \begin{bmatrix} \boldsymbol{M} \\ \boldsymbol{v} \end{bmatrix}$, we let $a, b$ and $a', b'$ be the row/column counts of our interest for $\boldsymbol{M}$ and $\boldsymbol{M}'$ respectively. Without loss of generality, we suppose the first $b$ columns of $\boldsymbol{M}$ all contain at least one non-negative entry (and the rest do not, by definition of $b$). We know that $a + b \geq \min\{n, m\}$, and

$$a' = a + \mathbb{1}(\min_i \boldsymbol{v}_i \leq 0), \qquad b' = b + \sum_{i=b+1}^{m} \mathbb{1}(\boldsymbol{v}_i \geq 0), \tag{C.10}$$

which is

$$a' + b' = a + b + \sum_{i=b+1}^{m} \mathbb{1}(\boldsymbol{v}_i \geq 0) + \mathbb{1}(\min_i \boldsymbol{v}_i \leq 0). \tag{C.11}$$

There are two scenarios:

1. When $b = m$, we have $\sum_{i=b+1}^{m} \mathbb{1}(\boldsymbol{v}_i \geq 0) + \mathbb{1}(\min_i \boldsymbol{v}_i \leq 0) \geq 0$

2. When $b < m$, we have $\sum_{i=b+1}^{m} \mathbb{1}(\boldsymbol{v}_i \geq 0) + \mathbb{1}(\min_i \boldsymbol{v}_i \leq 0) \geq 1$.

Therefore, we find that

$$a' + b' \geq \min\{a + m, a + b + 1\} \geq \min\{m, \min\{n, m\} + 1\} = \min\{m, n + 1, m + 1\} = \min\{n + 1, m\}. \tag{C.12}$$

This shows $C^*(n + 1, m) \geq \min\{n + 1, m\}$. $\qquad\square$

### C.2. Bayes Optimal Classifier w.r.t. 0-1 loss

We first show that the Bayes optimal classifier is approximately a nearest-cluster rule. We have the following derivation:

$$\text{sign}\left(f^*(\boldsymbol{x})\right) = \text{sign}\left(\sum_{k=1}^{K_1} \exp\left(\frac{D\langle \boldsymbol{x}, \boldsymbol{\mu}_k\rangle}{\alpha^2}\right) - \sum_{k=K_1+1}^{K} \exp\left(\frac{D\langle \boldsymbol{x}, \boldsymbol{\mu}_k\rangle}{\alpha^2}\right)\right)$$

$$= \text{sign}\left(\frac{\alpha^2}{D}\log\left(\sum_{k=1}^{K_1} \exp\left(\frac{D\langle \boldsymbol{x}, \boldsymbol{\mu}_k\rangle}{\alpha^2}\right)\right) - \frac{\alpha^2}{D}\log\left(\sum_{k=K_1+1}^{K} \exp\left(\frac{D\langle \boldsymbol{x}, \boldsymbol{\mu}_k\rangle}{\alpha^2}\right)\right)\right)$$

$$= \text{sign}\Big( \max_{1 \leq k \leq K_1} \langle \boldsymbol{x}, \mu_k \rangle - \max_{K_1+1 \leq k \leq K} \langle \boldsymbol{x}, \mu_k \rangle + \mathcal{O}\big( \log K \frac{\alpha^2}{D} \big) \Big) ,$$

where the second inequality is due to the fact that $\frac{\alpha^2}{D} \log(\cdot)$ function is a non-decreasing function, and the third inequality is because $\text{LogSumExp}(\{z_1, \cdots, z_K\})$ function with a temperature $\frac{D}{\alpha^2}$ uniformly approximate $\max_k z_k$ with an error $\mathcal{O}\big( \log K \frac{\alpha^2}{D} \big)$. As we discussed in the main paper, this is a nearest-cluster rule when $\alpha^2/D$ is small.

We next show that the Bayes optimal classifier is robust. Our proof will use Hoeffding's inequality for high-dimensional Gaussian vectors

**Lemma 2** (Hoeffding inequality). *For any unit vector $\boldsymbol{\mu} \in \mathbb{S}^{D-1}$, we have*

$$\mathbb{P}_{\boldsymbol{\varepsilon} \sim \mathcal{N}\left(0, \frac{\alpha^2}{D}\boldsymbol{I}\right)} \left( | \langle \boldsymbol{\mu}, \boldsymbol{\varepsilon} \rangle | > t \right) \leq 2 \exp\left( -\frac{Dt^2}{2\alpha^2} \right) . \tag{C.13}$$

And the concentration result of the norm of high-dimensional Gaussian vectors

**Lemma 3.** *We have*

$$\mathbb{P}_{\boldsymbol{\varepsilon} \sim \mathcal{N}\left(0, \frac{\alpha^2}{D}\boldsymbol{I}\right)} \left( \|\boldsymbol{\varepsilon}\| > t \right) \leq 4 \exp\left( -\frac{t^2}{8\alpha^2} \right) , \tag{C.14}$$

**Proposition 2** (Restated). *The Bayes optimal classifier w.r.t. the 0-1 loss is* $\text{sign}\left( f^*(\boldsymbol{x}) \right)$, *where*

$$f^*(\boldsymbol{x}) = \sum_{k=1}^{K_1} \exp\left( \frac{D \langle \boldsymbol{x}, \boldsymbol{\mu}_k \rangle}{\alpha^2} \right) - \sum_{k=K_1+1}^{K} \exp\left( \frac{D \langle \boldsymbol{x}, \boldsymbol{\mu}_k \rangle}{\alpha^2} \right) .$$

*Moreover, given a sample* $(\boldsymbol{x}, y) \sim \mathcal{D}_{X,Y}$, *we have, for any* $\frac{2\sqrt{2}\alpha^2 \log K}{D} \leq \nu \leq \sqrt{2}$,

$$\mathbb{P}\left( \min_{\|\boldsymbol{d}\| \leq 1} \left[ f^*\left( \boldsymbol{x} + \frac{\sqrt{2} - \nu}{2} \boldsymbol{d} \right) y \right] > 0 \right)$$

$$\geq 1 - 2K \exp\left( -\frac{D\nu^2}{64\alpha^2} \right) . \tag{C.15}$$

*Proof.* **Bayes optimal classifier for** $\mathcal{D}_{X,Y}$ The Bayes optimal classifier w.r.t. 0-1 loss is given by

$$f^*(\boldsymbol{x}) = \arg\max_y \mathbb{P}\left( Y = y \mid X = \boldsymbol{x} \right)$$

$$= \arg\max_y \sum_{k=1}^{K} \mathbb{P}\left( Y = y \mid Z = k, X = \boldsymbol{x} \right) \mathbb{P}\left( Z = k \mid X = \boldsymbol{x} \right)$$

$$= \begin{cases} 1, & \text{if } \sum_{k=1}^{K_1} \mathbb{P}\left( Z = k \mid X = \boldsymbol{x} \right) > \sum_{k=K_1+1}^{K} \mathbb{P}\left( Z = k \mid X = \boldsymbol{x} \right) \\ -1, & \text{o.w.} \end{cases}$$

$$= \text{sign}\left( \sum_{k=1}^{K_1} \mathbb{P}\left( Z = k \mid X = \boldsymbol{x} \right) - \sum_{k=K_1+1}^{K} \mathbb{P}\left( Z = k \mid X = \boldsymbol{x} \right) \right) . \tag{C.16}$$

Bayes rule and a few derivations give:

$$\mathbb{P}\left( Z = k \mid X = \boldsymbol{x} \right) = \frac{\mathbb{P}\left( X = \boldsymbol{x} \mid Z = k \right) \mathbb{P}\left( Z = k \right)}{\sum_{l=1}^{K} \mathbb{P}\left( X = \boldsymbol{x} \mid Z = l \right) \mathbb{P}\left( Z = l \right)}$$

$$= \frac{\exp\left( -\frac{D\|\boldsymbol{x} - \boldsymbol{\mu}_k\|^2}{2\alpha^2} \right)}{\sum_{l=1}^{K} \exp\left( -\frac{D\|\boldsymbol{x} - \boldsymbol{\mu}_l\|^2}{2\alpha^2} \right)}$$

$$= \frac{\exp\left( -\frac{D(\|\boldsymbol{x}\|^2 - 2\langle \boldsymbol{x}, \boldsymbol{\mu}_k \rangle + \|\boldsymbol{\mu}_k\|^2))}{2\alpha^2} \right)}{\sum_{l=1}^{K} \exp\left( -\frac{D(\|\boldsymbol{x}\|^2 - 2\langle \boldsymbol{x}, \boldsymbol{\mu}_l \rangle + \|\boldsymbol{\mu}_l\|^2)}{2\alpha^2} \right)} = \frac{\exp\left( \frac{D\langle \boldsymbol{x}, \boldsymbol{\mu}_k \rangle}{\alpha^2} \right)}{\sum_{l=1}^{K} \exp\left( \frac{D\langle \boldsymbol{x}, \boldsymbol{\mu}_l \rangle}{\alpha^2} \right)} . \tag{C.17}$$

Combining (C.16) and (C.17), we have

$$f^*(\boldsymbol{x}) = \text{sign} \left( \sum_{k=1}^{K_1} \exp \left( \frac{D \langle \boldsymbol{x}, \boldsymbol{\mu}_k \rangle}{\alpha^2} \right) - \sum_{k=K_1+1}^{K} \exp \left( \frac{D \langle \boldsymbol{x}, \boldsymbol{\mu}_k \rangle}{\alpha^2} \right) \right). \tag{C.18}$$

**Robustness of $f^*$.** We now proceed to show that $f^*$ is robust near-optimally. Since

$$\mathbb{P} \left( \min_{\|\boldsymbol{d}\| \le 1} \left[ f^* \left( \boldsymbol{x} + \frac{\sqrt{2} - \nu}{2} \boldsymbol{d} \right) y \right] \le 0 \right)$$

$$= \sum_{k=1}^{K} \mathbb{P} \left( \min_{\|\boldsymbol{d}\| \le 1} \left[ f^* \left( \boldsymbol{x} + \frac{\sqrt{2} - \nu}{2} \boldsymbol{d} \right) y \right] \le 0 \,\middle|\, z = k \right) \mathbb{P}(z = k),$$

It suffices to show that $\forall 1 \le k \le K$

$$\mathbb{P} \left( \min_{\|\boldsymbol{d}\| \le 1} \left[ f^* \left( \boldsymbol{x} + \frac{\sqrt{2} - \nu}{2} \boldsymbol{d} \right) y \right] \le 0 \,\middle|\, z = k \right) \le K \exp \left( -\frac{CD\nu^2}{16\alpha^2} \right). \tag{C.19}$$

When $k \le K_1$, we have

$$\mathbb{P} \left( \min_{\|\boldsymbol{d}\| \le 1} \left[ f^* \left( \boldsymbol{x} + \frac{\sqrt{2} - \nu}{2} \boldsymbol{d} \right) y \right] \le 0 \,\middle|\, z = k \right)$$

$$= \mathbb{P}_{\boldsymbol{\varepsilon}} \left( \min_{\|\boldsymbol{d}\| \le 1} \left[ f^* \left( \boldsymbol{x} + \frac{\sqrt{2} - \nu}{2} \boldsymbol{d} \right) \right] \le 0 \right)$$

$$= \mathbb{P}_{\boldsymbol{\varepsilon}} \left( \min_{\|\boldsymbol{d}\| \le 1} \left[ \exp \left( \frac{D}{\alpha^2} \left( 1 + \langle \boldsymbol{\mu}_k, \boldsymbol{\varepsilon} \rangle + \frac{\sqrt{2} - \nu}{2} \langle \boldsymbol{d}, \boldsymbol{\mu}_k \rangle \right) \right) \right. \right.$$

$$+ \sum_{l \ne k, 1 \le l \le K_1} \exp \left( \frac{D}{\alpha^2} \left( \langle \boldsymbol{\mu}_l, \boldsymbol{\varepsilon} \rangle + \frac{\sqrt{2} - \nu}{2} \langle \boldsymbol{d}, \boldsymbol{\mu}_l \rangle \right) \right)$$

$$\left. \left. - \sum_{K_1+1 \le l \le K} \exp \left( \frac{D}{\alpha^2} \left( \langle \boldsymbol{\mu}_l, \boldsymbol{\varepsilon} \rangle + \frac{\sqrt{2} - \nu}{2} \langle \boldsymbol{d}, \boldsymbol{\mu}_l \rangle \right) \right) \right] \le 0 \right)$$

$$\le \mathbb{P}_{\boldsymbol{\varepsilon}} \left( \min_{\|\boldsymbol{d}\| \le 1} \left[ \exp \left( \frac{D}{\alpha^2} \left( 1 + \langle \boldsymbol{\mu}_k, \boldsymbol{\varepsilon} \rangle + \frac{\sqrt{2} - \nu}{2} \langle \boldsymbol{d}, \boldsymbol{\mu}_k \rangle \right) \right) \right. \right.$$

$$\left. \left. - \sum_{K_1+1 \le l \le K} \exp \left( \frac{D}{\alpha^2} \left( \langle \boldsymbol{\mu}_l, \boldsymbol{\varepsilon} \rangle + \frac{\sqrt{2} - \nu}{2} \langle \boldsymbol{d}, \boldsymbol{\mu}_l \rangle \right) \right) \right] \le 0 \right)$$

$$\le \mathbb{P}_{\boldsymbol{\varepsilon}} \left( \min_{\|\boldsymbol{d}\| \le 1} \left[ \exp \left( \frac{D}{\alpha^2} \left( 1 + \langle \boldsymbol{\mu}_k, \boldsymbol{\varepsilon} \rangle + \frac{\sqrt{2} - \nu}{2} \langle \boldsymbol{d}, \boldsymbol{\mu}_k \rangle \right) \right) \right. \right.$$

$$\left. \left. - \sum_{K_1+1 \le l \le K} \exp \left( \frac{D}{\alpha^2} \left( |\langle \boldsymbol{\mu}_l, \boldsymbol{\varepsilon} \rangle| + \frac{\sqrt{2} - \nu}{2} |\langle \boldsymbol{d}, \boldsymbol{\mu}_l \rangle| \right) \right) \right] \le 0 \right)$$

$$\le \mathbb{P}_{\boldsymbol{\varepsilon}} \left( \min_{\|\boldsymbol{d}\| \le 1} \left[ \exp \left( \frac{D}{\alpha^2} \left( 1 + \langle \boldsymbol{\mu}_k, \boldsymbol{\varepsilon} \rangle + \frac{\sqrt{2} - \nu}{2} \langle \boldsymbol{d}, \boldsymbol{\mu}_k \rangle \right) \right) \right. \right.$$

$$\left. \left. - K_2 \exp \left( \frac{D}{\alpha^2} \left( \max_{K_1+1 \le l \le K} |\langle \boldsymbol{\mu}_l, \boldsymbol{\varepsilon} \rangle| + \frac{\sqrt{2} - \nu}{2} \max_{K_1+1 \le l \le K} |\langle \boldsymbol{d}, \boldsymbol{\mu}_l \rangle| \right) \right) \right] \le 0 \right)$$

$$\le \mathbb{P}_{\boldsymbol{\varepsilon}} \left( \min_{\|\boldsymbol{d}\| \le 1} \left[ 1 + \langle \boldsymbol{\mu}_k, \boldsymbol{\varepsilon} \rangle + \frac{\sqrt{2} - \nu}{2} \langle \boldsymbol{d}, \boldsymbol{\mu}_k \rangle \right. \right.$$

$$- \frac{\alpha^2}{D} \log K_2 - \max_{K_1+1 \leq l \leq K} |\langle \boldsymbol{\mu}_l, \boldsymbol{\varepsilon} \rangle| - \frac{\sqrt{2}-\nu}{2} \max_{K_1+1 \leq l \leq K} |\langle \boldsymbol{d}, \boldsymbol{\mu}_l \rangle| \Bigg] \leq 0 \Bigg)$$

$$\leq \mathbb{P}_{\boldsymbol{\varepsilon}} \Bigg( \min_{\|\boldsymbol{d}\| \leq 1} \Bigg[ 1 + \frac{\sqrt{2}-\nu}{2} \langle \boldsymbol{d}, \boldsymbol{\mu}_k \rangle - \frac{\sqrt{2}-\nu}{2} \max_{K_1+1 \leq l \leq K} |\langle \boldsymbol{d}, \boldsymbol{\mu}_l \rangle| \Bigg]$$

$$- \frac{\alpha^2}{D} \log K_2 - |\langle \boldsymbol{\mu}_k, \boldsymbol{\varepsilon} \rangle| - \max_{K_1+1 \leq l \leq K} |\langle \boldsymbol{\mu}_l, \boldsymbol{\varepsilon} \rangle| \leq 0 \Bigg) , \tag{C.20}$$

Since

$$\min_{\|\boldsymbol{d}\| \leq 1} \Bigg[ 1 + \frac{\sqrt{2}-\nu}{2} \langle \boldsymbol{d}, \boldsymbol{\mu}_k \rangle - \frac{\sqrt{2}-\nu}{2} \max_{K_1+1 \leq l \leq K} |\langle \boldsymbol{d}, \boldsymbol{\mu}_l \rangle| \Bigg]$$

$$\geq \min_{\|\boldsymbol{d}\| \leq 1} \Bigg[ 1 + \frac{\sqrt{2}-\nu}{2} \langle \boldsymbol{d}, \boldsymbol{\mu}_k \rangle - \frac{\sqrt{2}-\nu}{2} \sqrt{\sum_{K_1+1 \leq l \leq K} |\langle \boldsymbol{d}, \boldsymbol{\mu}_l \rangle|^2} \Bigg]$$

$$\geq \min_{\|\boldsymbol{d}\| \leq 1} \Bigg[ 1 - \frac{\sqrt{2}-\nu}{2} \sqrt{2} \sqrt{|\langle \boldsymbol{d}, \boldsymbol{\mu}_k \rangle|^2 + \sum_{K_1+1 \leq l \leq K} |\langle \boldsymbol{d}, \boldsymbol{\mu}_l \rangle|^2} \Bigg] \geq \min_{\|\boldsymbol{d}\| \leq 1} \Bigg[ 1 - \frac{\sqrt{2}-\nu}{\sqrt{2}} \|\boldsymbol{d}\| \Bigg] = \frac{\nu}{\sqrt{2}} ,$$

we finally have

$$(\text{C.20}) \leq \mathbb{P}_{\boldsymbol{\varepsilon}} \Bigg( \frac{\nu}{\sqrt{2}} - \frac{\alpha^2}{D} \log K_2 - |\langle \boldsymbol{\mu}_k, \boldsymbol{\varepsilon} \rangle| - \max_{K_1+1 \leq l \leq K} |\langle \boldsymbol{\mu}_l, \boldsymbol{\varepsilon} \rangle| \leq 0 \Bigg)$$

$$\leq \mathbb{P}_{\boldsymbol{\varepsilon}} \Bigg( \frac{\nu}{2\sqrt{2}} - 2 \max_{1 \leq l \leq K} |\langle \boldsymbol{\mu}_l, \boldsymbol{\varepsilon} \rangle| \leq 0 \Bigg)$$

$$\leq K \mathbb{P}_{\boldsymbol{\varepsilon}} \Bigg( |\langle \boldsymbol{\mu}_1, \boldsymbol{\varepsilon} \rangle| \geq \frac{\nu}{4\sqrt{2}} \Bigg) \leq 2K \exp \Bigg( -\frac{D\nu^2}{64\alpha^2} \Bigg) . \tag{C.21}$$

The proof for the case $k \geq K_1 + 1$ is almost identical. $\qquad\square$

# D. PReLU converges to optimal $\ell_2$-robust classifier, Part One: Convergence implies robustness

We first show that $F^p$ behaves like a nearest-cluster rule, which is critical for it to be robust. Notice that

$$
\begin{aligned}
\text{sign}\left(F^p(\boldsymbol{x})\right) &= \text{sign}\left(\sum_{k=1}^{K_1}\sigma^p(\langle\boldsymbol{x},\boldsymbol{\mu}_k\rangle) - \sum_{k=K_1+1}^{K}\sigma^p(\langle\boldsymbol{x},\boldsymbol{\mu}_k\rangle)\right) \\
&= \text{sign}\left(\left(\sum_{k=1}^{K_1}\sigma^p(\langle\boldsymbol{x},\boldsymbol{\mu}_k\rangle)\right)^{\frac{1}{p}} - \left(\sum_{k=K_1+1}^{K}\sigma^p(\langle\boldsymbol{x},\boldsymbol{\mu}_k\rangle)\right)^{\frac{1}{p}}\right) \\
&= \text{sign}\left(\left\|\begin{bmatrix}\sigma(\langle\boldsymbol{x},\boldsymbol{\mu}_1\rangle)\\\sigma(\langle\boldsymbol{x},\boldsymbol{\mu}_2\rangle)\\\vdots\\\sigma(\langle\boldsymbol{x},\boldsymbol{\mu}_{K_1}\rangle)\end{bmatrix}\right\|_p - \left\|\begin{bmatrix}\sigma(\langle\boldsymbol{x},\boldsymbol{\mu}_{K_1+1}\rangle)\\\sigma(\langle\boldsymbol{x},\boldsymbol{\mu}_{K_1+2}\rangle)\\\vdots\\\sigma(\langle\boldsymbol{x},\boldsymbol{\mu}_K\rangle)\end{bmatrix}\right\|_p\right).
\end{aligned}
$$

When $p$ gets larger, the $\|\cdot\|_p$ is closer to the $\|\cdot\|_\infty$. Therefore, $F^p$ behaves more like a nearest-cluster rule for large $p$.

We next prove Proposition 3 here.

**Proposition 3** (Restated). *Let $p > 2$. Given a classifier $f$ that satisfies $f(\gamma\boldsymbol{x}) = \gamma f(\boldsymbol{x})$, $\forall\boldsymbol{x}\in\mathbb{R}^D$, $\forall\gamma > 0$ and $\text{dist}(f, F^{(p)}) = \inf_{c>0}\sup_{\boldsymbol{x}\in\mathbb{S}^{D-1}}|cf(\boldsymbol{x}) - F^{(p)}(\boldsymbol{x})| \le \nu$ for some $0 < \nu \le \left(\frac{\sqrt{2}}{8}\right)^p$. Then for a sample $(\boldsymbol{x}, y)\sim\mathcal{D}_{X,Y}$, we have*

$$
\mathbb{P}\left(\min_{\|\boldsymbol{d}\|\le 1}\left[f\left(\boldsymbol{x} + \frac{\sqrt{2} - 8\nu^{\frac{1}{p}}}{2}\boldsymbol{d}\right)y\right] > 0\right)
$$
$$
\ge 1 - 2K\exp\left(-\frac{D\nu^{\frac{2}{p}}}{2K^2\alpha^2}\right) - 4\exp\left(-\frac{3}{8\alpha^2}\right). \tag{D.1}
$$

*Proof.* First of all, since $f(\gamma\boldsymbol{x}) = \gamma f(\boldsymbol{x})$, $\forall\boldsymbol{x}\in\mathbb{R}^D$, $\forall\gamma > 0$ and the same holds for $F^{(p)}(\cdot)$, we suppose the infimum is attained at $c^* \ge 0$, then

$$
\sup_{\boldsymbol{x}\in\mathbb{R}^D}|c^*f(\boldsymbol{x}) - F^{(p)}(\boldsymbol{x})| = \sup_{\boldsymbol{x}\in\mathbb{R}^D}\left|c^*f\left(\frac{\boldsymbol{x}}{\|\boldsymbol{x}\|}\right) - F^{(p)}\left(\frac{\boldsymbol{x}}{\|\boldsymbol{x}\|}\right)\right|\|\boldsymbol{x}\| \le \|\boldsymbol{x}\|\nu, \tag{D.2}
$$

where the last inequality uses $\text{dist}(f, F^{(p)}) \le \nu$. With (D.2), we have

$$
\mathbb{P}\left(\min_{\|\boldsymbol{d}\|\le 1}\left[f\left(\boldsymbol{x} + \frac{\sqrt{2} - 8\nu^{\frac{1}{p}}}{2}\boldsymbol{d}\right)y\right] \le 0\right)
$$
$$
\le\mathbb{P}\left(\min_{\|\boldsymbol{d}\|\le 1}\left[c^*f\left(\boldsymbol{x} + \frac{\sqrt{2} - 8\nu^{\frac{1}{p}}}{2}\boldsymbol{d}\right)y\right] \le 0\right)
$$
$$
=\mathbb{P}\left(\min_{\|\boldsymbol{d}\|\le 1}\left[c^*f\left(\boldsymbol{x} + \frac{\sqrt{2} - 8\nu^{\frac{1}{p}}}{2}\boldsymbol{d}\right)y - F^{(p)}\left(\boldsymbol{x} + \frac{\sqrt{2} - 8\nu^{\frac{1}{p}}}{2}\boldsymbol{d}\right)y + F^{(p)}\left(\boldsymbol{x} + \frac{\sqrt{2} - 8\nu^{\frac{1}{p}}}{2}\boldsymbol{d}\right)y\right] \le 0\right)
$$
$$
\le\mathbb{P}\left(\min_{\|\boldsymbol{d}\|\le 1}\left[F^{(p)}\left(\boldsymbol{x} + \frac{\sqrt{2} - 8\nu^{\frac{1}{p}}}{2}\boldsymbol{d}\right)y\right] - \max_{\|\boldsymbol{d}\|\le 1}\left|c^*f\left(\boldsymbol{x} + \frac{\sqrt{2} - 8\nu^{\frac{1}{p}}}{2}\boldsymbol{d}\right)y - F^{(p)}\left(\boldsymbol{x} + \frac{\sqrt{2} - 8\nu^{\frac{1}{p}}}{2}\boldsymbol{d}\right)y\right| \le 0\right)
$$
$$
\le\mathbb{P}\left(\min_{\|\boldsymbol{d}\|\le 1}\left[F^{(p)}\left(\boldsymbol{x} + \frac{\sqrt{2} - 8\nu^{\frac{1}{p}}}{2}\boldsymbol{d}\right)y\right] - \max_{\|\boldsymbol{z}\|^2\le 9}\left|c^*f(\boldsymbol{z})y - F^{(p)}(\boldsymbol{z})y\right| \le 0, \|\boldsymbol{x}\|^2 \le \frac{17}{2}\right) + \mathbb{P}\left(\|\boldsymbol{x}\|^2 > \frac{17}{2}\right)
$$
$$
\le\mathbb{P}\left(\min_{\|\boldsymbol{d}\|\le 1}\left[F^{(p)}\left(\boldsymbol{x} + \frac{\sqrt{2} - 8\nu^{\frac{1}{p}}}{2}\boldsymbol{d}\right)y\right] - \max_{\|\boldsymbol{z}\|^2\le 9}\left|c^*f(\boldsymbol{z})y - F^{(p)}(\boldsymbol{z})y\right| \le 0\right) + \mathbb{P}\left(\|\boldsymbol{x}\|^2 > \frac{17}{2}\right)
$$
$$
\le\mathbb{P}\left(\min_{\|\boldsymbol{d}\|\le 1}\left[F^{(p)}\left(\boldsymbol{x} + \frac{\sqrt{2} - 8\nu^{\frac{1}{p}}}{2}\boldsymbol{d}\right)y\right] \le 3\nu\right) + \mathbb{P}\left(\|\boldsymbol{x}\|^2 > \frac{17}{2}\right).
$$

The second term $\mathbb{P}\left(\|x\|^2 > \frac{17}{2}\right)$ is easy to bound, our focus is to show

$$\mathbb{P}\left(\min_{\|d\|\leq 1}\left[F^{(p)}\left(x + \frac{\sqrt{2}-8\nu^{\frac{1}{p}}}{2}d\right)y\right] \leq 3\nu\right) \geq 2(K+1)\exp\left(-\frac{CD\nu^2}{K^2\alpha^2}\right), \tag{D.3}$$

which resembles the result in Min & Vidal (2024, Theorem 1), but one can not directly obtain (D.3) from this existing result. Nonetheless, we can partially follow Min & Vidal (2024, Theorem 1)'s proof and obtain (D.3) (with non-trivial new derivations), as shown below:

Since

$$\mathbb{P}\left(\min_{\|d\|\leq 1}\left[F^{(p)}\left(x + \frac{\sqrt{2}-8\nu^{\frac{1}{p}}}{2}d\right)y\right] \leq 3\nu\right)$$

$$= \sum_{k=1}^{K}\mathbb{P}\left(\min_{\|d\|\leq 1}\left[F^{(p)}\left(x + \frac{\sqrt{2}-8\nu^{\frac{1}{p}}}{2}d\right)y\right] \leq 3\nu \,\middle|\, z = k\right)\mathbb{P}(z = k),$$

It suffices to show that $\forall 1 \leq k \leq K$

$$\mathbb{P}\left(\min_{\|d\|\leq 1}\left[F^{(p)}\left(x + \frac{\sqrt{2}-8\nu^{\frac{1}{p}}}{2}d\right)y\right] \leq 3\nu \,\middle|\, z = k\right) \leq 2(K_2+2)\exp\left(-\frac{CD\delta^2}{2(K_2+1)^2\alpha^2}\right). \tag{D.4}$$

When $k \leq K_1$, we have

$$\mathbb{P}\left(\min_{\|d\|\leq 1}\left[F^{(p)}\left(x + \frac{\sqrt{2}-8\nu^{\frac{1}{p}}}{2}d\right)y\right] \leq 3\nu \,\middle|\, z = k\right)$$

$$= \mathbb{P}_{\varepsilon}\left(\min_{\|d\|\leq 1}\left[F^{(p)}\left(x + \frac{\sqrt{2}-8\nu^{\frac{1}{p}}}{2}d\right)y\right] \leq 3\nu\right)$$

$$= \mathbb{P}_{\varepsilon}\left(\min_{\|d\|\leq 1}\left[\sigma^p\left(1 + \langle\boldsymbol{\mu}_k,\varepsilon\rangle + \frac{\sqrt{2}-8\nu^{\frac{1}{p}}}{2}\langle d,\boldsymbol{\mu}_k\rangle\right)\right.\right.$$

$$+ \sum_{l\neq k, 1\leq l\leq K_1}\sigma^p\left(\langle\boldsymbol{\mu}_l,\varepsilon\rangle + \frac{\sqrt{2}-8\nu^{\frac{1}{p}}}{2}\langle d,\boldsymbol{\mu}_l\rangle\right)$$

$$\left.\left. - \sum_{K_1+1\leq l\leq K}\sigma^p\left(\langle\boldsymbol{\mu}_l,\varepsilon\rangle + \frac{\sqrt{2}-8\nu^{\frac{1}{p}}}{2}\langle d,\boldsymbol{\mu}_l\rangle\right)\right] \leq 3\nu\right)$$

$$\leq \mathbb{P}_{\varepsilon}\left(\min_{\|d\|\leq 1}\left[\sigma^p\left(1 + \langle\boldsymbol{\mu}_k,\varepsilon\rangle + \frac{\sqrt{2}-8\nu^{\frac{1}{p}}}{2}\langle d,\boldsymbol{\mu}_k\rangle\right)\right.\right.$$

$$\left.\left. - \sum_{K_1+1\leq l\leq K}\sigma^p\left(\langle\boldsymbol{\mu}_l,\varepsilon\rangle + \frac{\sqrt{2}-8\nu^{\frac{1}{p}}}{2}\langle d,\boldsymbol{\mu}_l\rangle\right)\right] \leq 3\nu\right) \tag{D.5}$$

We define the event

$$\mathcal{E} := \left\{1 + \langle\boldsymbol{\mu}_k,\varepsilon\rangle + \frac{\sqrt{2}-8\nu^{\frac{1}{p}}}{2}\langle d,\boldsymbol{\mu}_k\rangle \geq 0, \forall d \in \mathbb{S}^{D-1}\right\}, \tag{D.6}$$

Then, by Min & Vidal (2024, Lemma 2),

$$\text{(D.5)} = \mathbb{P}_{\varepsilon}\left(\min_{\|d\|\leq 1}\left[\sigma^p\left(1 + \langle\boldsymbol{\mu}_k,\varepsilon\rangle + \frac{\sqrt{2}-8\nu^{\frac{1}{p}}}{2}\langle d,\boldsymbol{\mu}_k\rangle\right)\right.\right.$$

$$\left.\left. - \sum_{K_1+1\leq l\leq K}\sigma^p\left(\langle\boldsymbol{\mu}_l,\varepsilon\rangle + \frac{\sqrt{2}-8\nu^{\frac{1}{p}}}{2}\langle d,\boldsymbol{\mu}_l\rangle\right)\right] \leq 3\nu\right)$$

$$\le \; \mathbb{P}_{\boldsymbol{\varepsilon}}\left(\min_{\|\boldsymbol{d}\|\le 1}\left[\sigma^p\left(1+\langle\boldsymbol{\mu}_k,\boldsymbol{\varepsilon}\rangle+\frac{\sqrt{2}-8\nu^{\frac{1}{p}}}{2}\langle\boldsymbol{d},\boldsymbol{\mu}_k\rangle\right)\right.\right.$$

$$\left.\left.-\sum_{K_1+1\le l\le K}\sigma^p\left(\langle\boldsymbol{\mu}_l,\boldsymbol{\varepsilon}\rangle+\frac{\sqrt{2}-8\nu^{\frac{1}{p}}}{2}\langle\boldsymbol{d},\boldsymbol{\mu}_l\rangle\right)\right]\le 3\nu,\mathcal{E}\right)+\mathbb{P}\left(\mathcal{E}^c\right)\qquad(\text{D.7})$$

Since under event $\mathcal{E}$, we have $\sigma^p\left(1+\langle\boldsymbol{\mu}_k,\boldsymbol{\varepsilon}\rangle+\frac{\sqrt{2}-8\nu^{\frac{1}{p}}}{2}\langle\boldsymbol{d},\boldsymbol{\mu}_k\rangle\right)=\left(1+\langle\boldsymbol{\mu}_k,\boldsymbol{\varepsilon}\rangle+\frac{\sqrt{2}-8\nu^{\frac{1}{p}}}{2}\langle\boldsymbol{d},\boldsymbol{\mu}_k\rangle\right)^p$, we can proceed with

$$(\text{D.7})=\mathbb{P}_{\boldsymbol{\varepsilon}}\left(\min_{\|\boldsymbol{d}\|\le 1}\left[\left(1+\langle\boldsymbol{\mu}_k,\boldsymbol{\varepsilon}\rangle+\frac{\sqrt{2}-8\nu^{\frac{1}{p}}}{2}\langle\boldsymbol{d},\boldsymbol{\mu}_k\rangle\right)^p\right.\right.$$

$$\left.\left.-\sum_{K_1+1\le l\le K}\sigma^p\left(\langle\boldsymbol{\mu}_l,\boldsymbol{\varepsilon}\rangle+\frac{\sqrt{2}-8\nu^{\frac{1}{p}}}{2}\langle\boldsymbol{d},\boldsymbol{\mu}_l\rangle\right)\right]\le 3\nu,\mathcal{E}\right)+\mathbb{P}\left(\mathcal{E}^c\right)$$

$$\le \mathbb{P}_{\boldsymbol{\varepsilon}}\left(\min_{\|\boldsymbol{d}\|\le 1}\left[\left(1+\langle\boldsymbol{\mu}_k,\boldsymbol{\varepsilon}\rangle+\frac{\sqrt{2}-8\nu^{\frac{1}{p}}}{2}\langle\boldsymbol{d},\boldsymbol{\mu}_k\rangle\right)^p\right.\right.$$

$$\left.\left.-\sum_{K_1+1\le l\le K}\left(\langle|\boldsymbol{\mu}_l,\boldsymbol{\varepsilon}\rangle|+\frac{\sqrt{2}-8\nu^{\frac{1}{p}}}{2}|\langle\boldsymbol{d},\boldsymbol{\mu}_l\rangle|\right)^p-3\nu\right]<0,\mathcal{E}\right)+\mathbb{P}\left(\mathcal{E}^c\right)$$

$$\le \mathbb{P}_{\boldsymbol{\varepsilon}}\left(\min_{\|\boldsymbol{d}\|\le 1}\left[1+\langle\boldsymbol{\mu}_k,\boldsymbol{\varepsilon}\rangle+\frac{\sqrt{2}-8\nu^{\frac{1}{p}}}{2}\langle\boldsymbol{d},\boldsymbol{\mu}_k\rangle\right.\right.$$

$$\left.\left.-\left(\sum_{K_1+1\le l\le K}\left(|\langle\boldsymbol{\mu}_l,\boldsymbol{\varepsilon}\rangle|+\frac{\sqrt{2}-8\nu^{\frac{1}{p}}}{2}|\langle\boldsymbol{d},\boldsymbol{\mu}_l\rangle|\right)^p+3\nu\right)^{1/p}\right]<0,\mathcal{E}\right)+\mathbb{P}\left(\mathcal{E}^c\right)$$

$$\le \mathbb{P}_{\boldsymbol{\varepsilon}}\left(\underbrace{\min_{\|\boldsymbol{d}\|\le 1}\left[1+\frac{\sqrt{2}-8\nu^{\frac{1}{p}}}{2}\langle\boldsymbol{d},\boldsymbol{\mu}_k\rangle-\frac{\sqrt{2}-8\nu^{\frac{1}{p}}}{2}\left(\sum_{K_1+1\le l\le K}|\langle\boldsymbol{d},\boldsymbol{\mu}_l\rangle|^p\right)^{1/p}\right]}_{:=M^*(\nu)}\right.$$

$$\left.-\left(\sum_{K_1+1\le l\le K}(|\langle\boldsymbol{\mu}_l,\boldsymbol{\varepsilon}\rangle|)^p+3\nu\right)^{1/p}-|\langle\boldsymbol{\mu}_k,\boldsymbol{\varepsilon}\rangle|<0,\mathcal{E}\right)+\mathbb{P}\left(\mathcal{E}^c\right)$$

$$\le \mathbb{P}_{\boldsymbol{\varepsilon}}\left(M^*(\nu)-\sum_{K_1+1\le l\le K}|\langle\boldsymbol{\mu}_l,\boldsymbol{\varepsilon}\rangle|-(3\nu)^{\frac{1}{p}}-|\langle\boldsymbol{\mu}_k,\boldsymbol{\varepsilon}\rangle|<0\right)+\mathbb{P}\left(\mathcal{E}^c\right),\qquad(\text{D.8})$$

From the proof of Min & Vidal (2024, Theorem 1), we have $M^*(\nu)=4\sqrt{2}\nu^{\frac{1}{p}}$. Therefore we have

$$(\text{D.8})=\mathbb{P}_{\boldsymbol{\varepsilon}}\left(\sum_{K_1+1\le l\le K}|\langle\boldsymbol{\mu}_l,\boldsymbol{\varepsilon}\rangle|+|\langle\boldsymbol{\mu}_k,\boldsymbol{\varepsilon}\rangle|>M^*(\nu)-(3\nu)^{\frac{1}{p}}\right)+\mathbb{P}\left(\mathcal{E}^c\right)$$

$$\ge \mathbb{P}_{\boldsymbol{\varepsilon}}\left(\sum_{K_1+1\le l\le K}|\langle\boldsymbol{\mu}_l,\boldsymbol{\varepsilon}\rangle|+|\langle\boldsymbol{\mu}_k,\boldsymbol{\varepsilon}\rangle|>\left(4\sqrt{2}-3^{\frac{1}{p}}\right)\nu^{\frac{1}{p}}\right)+\mathbb{P}\left(\mathcal{E}^c\right)$$

$$\ge \mathbb{P}_{\boldsymbol{\varepsilon}}\left(\sum_{K_1+1\le l\le K}|\langle\boldsymbol{\mu}_l,\boldsymbol{\varepsilon}\rangle|+|\langle\boldsymbol{\mu}_k,\boldsymbol{\varepsilon}\rangle|>\sqrt{2}\nu^{\frac{1}{p}}\right)+\mathbb{P}\left(\mathcal{E}^c\right)$$

$$\geq \mathbb{P}_{\boldsymbol{\varepsilon}} \left( \max_{1 \leq k \leq K} |\langle \boldsymbol{\mu}_k, \boldsymbol{\varepsilon} \rangle| > \frac{\sqrt{2} \nu^{\frac{1}{p}}}{K} \right) + \mathbb{P}\left(\mathcal{E}^c\right)$$

$$\geq K \mathbb{P}_{\boldsymbol{\varepsilon}} \left( |\langle \boldsymbol{\mu}_1, \boldsymbol{\varepsilon} \rangle| > \frac{\sqrt{2} \nu^{\frac{1}{p}}}{K} \right) + \mathbb{P}\left(\mathcal{E}^c\right) \geq 2K \exp\left( -\frac{D \nu^{\frac{2}{p}}}{K^2 \alpha^2} \right) + \mathbb{P}\left(\mathcal{E}^c\right) .$$

Therefore, we have

$$\mathbb{P}\left( \min_{\|\boldsymbol{d}\| \leq 1} \left[ f\left( \boldsymbol{x} + \frac{\sqrt{2} - 8\nu^{\frac{1}{p}}}{2} \boldsymbol{d} \right) y \right] \leq 0 \right) \leq 2K \exp\left( -\frac{D \nu^{\frac{2}{p}}}{K^2 \alpha^2} \right) + \mathbb{P}\left(\mathcal{E}^c\right) + \mathbb{P}\left( \|\boldsymbol{x}\|^2 > \frac{3}{2} \right) .$$

Finally, by

$$\mathbb{P}\left(\mathcal{E}^c\right) \leq \mathbb{P}\left( |\langle \boldsymbol{\mu}_k, \boldsymbol{\varepsilon} \rangle| \geq 1 - \frac{\sqrt{2}}{2} \right) \leq \mathbb{P}\left( |\langle \boldsymbol{\mu}_k, \boldsymbol{\varepsilon} \rangle| \geq \frac{2}{5} \right) \leq 2 \exp\left( -\frac{2D}{25\alpha^2} \right)$$

$$\mathbb{P}\left( \|\boldsymbol{x}\|^2 > \frac{17}{2} \right) \leq \mathbb{P}\left( \|\boldsymbol{\varepsilon}\| \geq \sqrt{\frac{17}{2}} - 1 \right) \leq 4 \exp\left( -\left( \sqrt{\frac{17}{2}} - 1 \right)^2 \frac{1}{8\alpha^2} \right) \leq 4 \exp\left( -\frac{3}{8\alpha^2} \right) ,$$

The proof is finished, notice that the bad event $\|\boldsymbol{x}\|^2 > \frac{17}{2}$ is chosen arbitrarily, so one can derive more general results by letting the results depend on the choice of a bad event. But for our purpose, we do not need it. $\qquad\square$

# E. PReLU converges to optimal $\ell_2$-robust classifier, Part Two: Basic results on neuron dynamics and good events

In this and the following sections, we let $\ell_i(t) := \ell(y_i, f^{(p)}(\boldsymbol{x}_i; \boldsymbol{\theta}(t)))$ denote the loss on data point $(\boldsymbol{x}_i, y_i)$, and $\nabla_{\hat{y}} \ell_i$ denotes the derivation of $\ell_i$ w.r.t. its second argument, the network output. Moreover, we let $c_{kj} := \cos(\boldsymbol{\mu}_k, \boldsymbol{w}_j(t))$ denote the cosine angle between cluster center $\boldsymbol{\mu}_k$ and neuron $\boldsymbol{w}_j$. Note: For simplicity, we drop the time dependence in $\boldsymbol{\theta}(t), v_j(t), \boldsymbol{w}_j(t), \mathcal{L}(t), \ell_i(t), c_{kj}(t)$ and write $\boldsymbol{\theta}, v_j, \boldsymbol{w}_j, \mathcal{L}, \ell_i, c_{kj}$ whenever it is clear that they come from the GF solution thus depend on time. Note: **It suffices to prove the case $\alpha = \alpha_0$, we thus use $\alpha$ to both denote the intra-class variance and the $\alpha_0$ we use to control the order of all the relevant quantities in our proofs.**

We also let $\mathcal{I}_k := \{i : (k-1)N + 1 \le i \le kN\}$, the index set of data sampled from $k$-th cluster.

## E.1. Results on neuron dynamics

**Neuron dynamics**: Under GF, we have

$$
\begin{aligned}
\frac{d}{dt} \boldsymbol{w}_j &= -\frac{1}{N} \sum_{i=1}^{KN} \nabla_{\hat{y}} \ell_i \, v_j \left( \frac{p[\sigma(\langle \boldsymbol{x}_i, \boldsymbol{w}_j \rangle)]^{p-1}}{\|\boldsymbol{w}_j\|^{p-1}} \boldsymbol{x}_i - (p-1) \frac{[\sigma(\langle \boldsymbol{x}_i, \boldsymbol{w}_j \rangle)]^p}{\|\boldsymbol{w}_j\|^{p+1}} \boldsymbol{w}_j \right) \\
&= -\frac{1}{N} \sum_{i:\langle \boldsymbol{x}_i, \boldsymbol{w}_j \rangle > 0} \nabla_{\hat{y}} \ell_i \, v_j \left( \frac{p[\langle \boldsymbol{x}_i, \boldsymbol{w}_j \rangle]^{p-1}}{\|\boldsymbol{w}_j\|^{p-1}} \boldsymbol{x}_i - (p-1) \frac{[\langle \boldsymbol{x}_k, \boldsymbol{w}_j \rangle]^p}{\|\boldsymbol{w}_j\|^{p+1}} \boldsymbol{w}_j \right)
\end{aligned}
$$

and similarly,

$$
\frac{d}{dt} v_j = -\frac{1}{N} \sum_{i:\langle \boldsymbol{x}_i, \boldsymbol{w}_j \rangle > 0} \nabla_{\hat{y}} \ell_i \frac{[\langle \boldsymbol{x}_i, \boldsymbol{w}_j \rangle]^p}{\|\boldsymbol{w}_j\|^{p-1}}
$$

**Balancedness**: We compute

$$
\begin{aligned}
\frac{d}{dt}(\boldsymbol{w}_j^\top \boldsymbol{w}_j) &= 2 \left\langle \frac{d}{dt} \boldsymbol{w}_j, \boldsymbol{w}_j \right\rangle \\
&= -\frac{1}{N} \sum_{i:\langle \boldsymbol{x}_i, \boldsymbol{w}_j \rangle > 0} \nabla_{\hat{y}} \ell_i \, v_j \left( \frac{p[\langle \boldsymbol{x}_i, \boldsymbol{w}_j \rangle]^p}{\|\boldsymbol{w}_j\|^{p-1}} - (p-1) \frac{[\langle \boldsymbol{x}_k, \boldsymbol{w}_j \rangle]^p}{\|\boldsymbol{w}_j\|^{p-1}} \right) \\
&= -\frac{2}{N} \sum_{i:\langle \boldsymbol{x}_i, \boldsymbol{w}_j \rangle > 0} \nabla_{\hat{y}} \ell_i \, v_j \frac{[\langle \boldsymbol{x}_i, \boldsymbol{w}_j \rangle]^p}{\|\boldsymbol{w}_j\|^{p-1}},
\end{aligned}
$$

and

$$
\frac{d}{dt} v_j^2 = 2 v_j \frac{d}{dt} v_j = -\frac{2}{N} \sum_{i:\langle \boldsymbol{x}_i, \boldsymbol{w}_j \rangle > 0} \nabla_{\hat{y}} \ell_i \, v_j \frac{[\langle \boldsymbol{x}_i, \boldsymbol{w}_j \rangle]^p}{\|\boldsymbol{w}_j\|^{p-1}}
$$

Therefore, we have

$$
\frac{d}{dt}(\boldsymbol{w}_j^\top \boldsymbol{w}_j - v_j^2) \equiv 0, \tag{E.1}
$$

thus $\boldsymbol{w}_j^\top(t)\boldsymbol{w}_j(t) - v_j^2(t) = \boldsymbol{w}_j^\top(0)\boldsymbol{w}_j(0) - v_j^2(0), \forall t$, since we have a balanced initialization such that $\boldsymbol{w}_j^\top(0)\boldsymbol{w}_j(0) - v_j^2(0), \forall j$. Such balancedness holds for all time $t$. Using this balancedness $v_j^2 \equiv \|\boldsymbol{w}_j\|^2, \forall j \in [h]$, we can write

$$
\frac{d}{dt} \boldsymbol{w}_j = -\frac{\text{sign}(v_j(0))}{N} \sum_{i:\langle \boldsymbol{x}_i, \boldsymbol{w}_j \rangle > 0} \nabla_{\hat{y}} \ell_i \, \|\boldsymbol{w}_j\| \left( p \left( \left\langle \boldsymbol{x}_i, \frac{\boldsymbol{w}_j}{\|\boldsymbol{w}_j\|} \right\rangle \right)^{p-1} \boldsymbol{x}_i - (p-1) \left( \left\langle \boldsymbol{x}_i, \frac{\boldsymbol{w}_j}{\|\boldsymbol{w}_j\|} \right\rangle \right)^p \frac{\boldsymbol{w}_j}{\|\boldsymbol{w}_j\|} \right), \tag{E.2}
$$

where we use that $\text{sign}(v_j(t)) = \text{sign}(v_j(0))$, which is another consequence of balancedness Boursier et al. (2022); Min et al. (2024). We will study the dynamics of $\boldsymbol{w}_j$ from now on, and one can write the time derivatives of the norm and direction of these neurons:

**Neuron norm dynamics**:

$$\frac{d}{dt}\|\boldsymbol{w}_j\|^2$$

$$= 2\left\langle \boldsymbol{w}_j, \frac{d}{dt}\boldsymbol{w}_j \right\rangle$$

$$= -2\frac{\text{sign}(v_j(0))}{N}\sum_{i:\langle \boldsymbol{x}_i, \boldsymbol{w}_j \rangle > 0}\nabla_{\hat{y}}\ell_i\,\|\boldsymbol{w}_j\|\left(p\left(\left\langle \boldsymbol{x}_i, \frac{\boldsymbol{w}_j}{\|\boldsymbol{w}_j\|}\right\rangle\right)^{p-1}\langle \boldsymbol{w}_j, \boldsymbol{x}_i\rangle - (p-1)\left(\left\langle \boldsymbol{x}_i, \frac{\boldsymbol{w}_j}{\|\boldsymbol{w}_j\|}\right\rangle\right)^p\|\boldsymbol{w}_j\|\right)$$

$$= -2\frac{\text{sign}(v_j(0))}{N}\sum_{i:\langle \boldsymbol{x}_i, \boldsymbol{w}_j \rangle > 0}\nabla_{\hat{y}}\ell_k\,\|\boldsymbol{w}_j\|\left(p\left(\left\langle \boldsymbol{x}_i, \frac{\boldsymbol{w}_j}{\|\boldsymbol{w}_j\|}\right\rangle\right)^p\|\boldsymbol{w}_j\| - (p-1)\left(\left\langle \boldsymbol{x}_i, \frac{\boldsymbol{w}_j}{\|\boldsymbol{w}_j\|}\right\rangle\right)^p\|\boldsymbol{w}_j\|\right)$$

$$= -2\frac{\text{sign}(v_j(0))}{N}\left(\sum_{i:\langle \boldsymbol{x}_i, \boldsymbol{w}_j \rangle > 0}\nabla_{\hat{y}}\ell_i\left(\left\langle \boldsymbol{x}_i, \frac{\boldsymbol{w}_j}{\|\boldsymbol{w}_j\|}\right\rangle\right)^p\right)\|\boldsymbol{w}_j\|^2 \tag{E.3}$$

**Neuron angular dynamics**:

$$\frac{d}{dt}\frac{\boldsymbol{w}_j}{\|\boldsymbol{w}_j\|}$$

$$= \left(I - \frac{\boldsymbol{w}_j\boldsymbol{w}_j^\top}{\|\boldsymbol{w}_j\|^2}\right)\frac{1}{\|\boldsymbol{w}_j\|}\frac{d}{dt}\boldsymbol{w}_j$$

$$= -\frac{\text{sign}(v_j(0))}{N}\sum_{i:\langle \boldsymbol{x}_i, \boldsymbol{w}_j \rangle > 0}\nabla_{\hat{y}}\ell_i\left(I - \frac{\boldsymbol{w}_j\boldsymbol{w}_j^\top}{\|\boldsymbol{w}_j\|^2}\right)\left(p\left(\left\langle \boldsymbol{x}_i, \frac{\boldsymbol{w}_j}{\|\boldsymbol{w}_j\|}\right\rangle\right)^{p-1}\boldsymbol{x}_i - (p-1)\left(\left\langle \boldsymbol{x}_i, \frac{\boldsymbol{w}_j}{\|\boldsymbol{w}_j\|}\right\rangle\right)^p\frac{\boldsymbol{w}_j}{\|\boldsymbol{w}_j\|}\right)$$

$$= -\frac{\text{sign}(v_j(0))}{N}\sum_{i:\langle \boldsymbol{x}_i, \boldsymbol{w}_j \rangle > 0}\nabla_{\hat{y}}\ell_i\,p\left(\left\langle \boldsymbol{x}_i, \frac{\boldsymbol{w}_j}{\|\boldsymbol{w}_j\|}\right\rangle\right)^{p-1}\left(\boldsymbol{x}_i - \left\langle \boldsymbol{x}_i, \frac{\boldsymbol{w}_j}{\|\boldsymbol{w}_j\|}\right\rangle\frac{\boldsymbol{w}_j}{\|\boldsymbol{w}_j\|}\right). \tag{E.4}$$

Finally, from the directional dynamics $\frac{d}{dt}\frac{\boldsymbol{w}_j}{\|\boldsymbol{w}_j\|}$, we obtain

$$\frac{d}{dt}c_{kj} = \left\langle \boldsymbol{\mu}_k, \frac{d}{dt}\frac{\boldsymbol{w}_j}{\|\boldsymbol{w}_j\|}\right\rangle$$

$$= -\frac{\text{sign}(v_j(0))}{N}\sum_{i:\langle \boldsymbol{x}_i, \boldsymbol{w}_j \rangle > 0}\nabla_{\hat{y}}\ell_i\,p\left(\left\langle \boldsymbol{x}_i, \frac{\boldsymbol{w}_j}{\|\boldsymbol{w}_j\|}\right\rangle\right)^{p-1}\left(\langle \boldsymbol{\mu}_k, \boldsymbol{x}_i\rangle - \left\langle \boldsymbol{x}_i, \frac{\boldsymbol{w}_j}{\|\boldsymbol{w}_j\|}\right\rangle c_{kj}\right), \tag{E.5}$$

and whenever $|c_{kj}| \neq 0$, we have

$$\frac{d}{dt}\log|c_{kj}|$$

$$= \frac{1}{c_{kj}}\frac{d}{dt}c_{kj}$$

$$= -\frac{\text{sign}(v_j(0))}{N}\sum_{i:\langle \boldsymbol{x}_i, \boldsymbol{w}_j \rangle > 0}\nabla_{\hat{y}}\ell_i\,p\left(\left\langle \boldsymbol{x}_i, \frac{\boldsymbol{w}_j}{\|\boldsymbol{w}_j\|}\right\rangle\right)^{p-1}\left(\frac{\langle \boldsymbol{\mu}_k, \boldsymbol{x}_i\rangle}{c_{kj}} - \left\langle \boldsymbol{x}_i, \frac{\boldsymbol{w}_j}{\|\boldsymbol{w}_j\|}\right\rangle\right) \tag{E.6}$$

Our proof has the same structure as prior works (Boursier et al., 2022; Min et al., 2024): We will study neuron's angular dynamics (E.5) at the early phase (alignment phase) of the GF training, and then study neuron's norm dynamics (E.3) at the later phase (convergence phase).

Lastly, in order to prove Lemma 7 and Proposition 4 in the next subsection, we need the following:

We let $\{\boldsymbol{\mu}_{K+1}, \cdots, \boldsymbol{\mu}_D\}$ be an orthonormal basis for the subspace that is orthogonal to $\text{span}\{\boldsymbol{\mu}_1, \cdots, \boldsymbol{\mu}_K\}$, and we can define $c_{kj} = \cos(\boldsymbol{\mu}_k, \boldsymbol{w}_j), k = K+1, \cdots, D$. Since $\{\boldsymbol{\mu}_1, \cdots, \boldsymbol{\mu}_D\}$ forms an orthonormal basis for the ambient space $\mathbb{R}^D$,

we have

$$\sum_{k=1}^{D} c_{kj}^2 = \sum_{k=1}^{D} \left| \left\langle \boldsymbol{\mu}_k, \frac{\boldsymbol{w}_j}{\|\boldsymbol{w}_j\|} \right\rangle \right|^2 = 1 \, . \tag{E.7}$$

Moreover, we can write the same time-derivatives $\frac{d}{dt} c_{kj}$, $\frac{d}{dt} \log |c_{kj}|$ for $c_{kj} = \cos(\boldsymbol{\mu}_k, \boldsymbol{w}_j), k = K+1, \cdots, D$ as in (E.5) and (E.6), respectively.

Lastly, the following inequality will be used frequently in our proof:

$$\sum_{l \neq k} c_{lj}^p \leq \sum_{1 \leq l \leq D, l \neq k} |c_{lj}|^p \leq \left( \sum_{1 \leq l \leq D, l \neq k} c_{lj}^2 \right)^{\frac{p}{2}} = \left( 1 - c_{kj}^2 \right)^{\frac{p}{2}} \tag{E.8}$$

**Note**: The sum operation $\sum_{l \neq k}$ implicitly assumes $l \leq K$. We will explicitly indicate the range of $l$ if it can take values between $K+1$ and $D$.

### E.2. Good Event

For a balanced dataset $\hat{\mathcal{D}} = \{\boldsymbol{x}_i, y_i\}_{i=1}^{KN}$, notice that $\boldsymbol{x}_i = \boldsymbol{\mu}_{\lceil \frac{i}{N} \rceil} + \boldsymbol{\varepsilon}_i$ for some $\boldsymbol{\varepsilon}_i \in \mathcal{N}\left(0, \frac{\alpha^2}{D} \boldsymbol{I}\right)$. We define the following good event w.r.t. these $\boldsymbol{\varepsilon}_i$s and show that they happen with high probability:

**Lemma 4.** *We define the event $\mathcal{E}_{good}$ when the following happens:*

1. $\|\boldsymbol{\varepsilon}_i\| \leq \sqrt{8 \log \frac{16KN}{\delta}} \alpha, \forall 1 \leq i \leq KN;$

2. $|\langle \boldsymbol{\mu}_k, \boldsymbol{\varepsilon}_i \rangle| \leq \sqrt{2 \log \frac{8K^2N}{\delta}} \frac{\alpha}{\sqrt{D}}, \forall 1 \leq i \leq KN, 1 \leq k \leq K;$

3. $\|\sum_{i \in \mathcal{I}_k} \boldsymbol{\varepsilon}_i\| \leq \sqrt{2 \log \frac{8K}{\delta}} \alpha \sqrt{N}, \forall 1 \leq k \leq K$

4. $\sum_{i \in \mathcal{I}_k} \|\boldsymbol{\varepsilon}_i\|^2 \leq 8 \log \frac{16K}{\delta} \alpha^2 N, \forall 1 \leq k \leq K$

*We have $\mathbb{P}\left(\mathcal{E}_{good}\right) \geq 1 - \delta$. Furthermore, for simplicity, we write*

1. $\|\boldsymbol{\varepsilon}_i\| \leq C \sqrt{\log \frac{K^2N}{\delta}} \alpha, \forall 1 \leq i \leq KN;$

2. $|\langle \boldsymbol{\mu}_k, \boldsymbol{\varepsilon}_i \rangle| \leq C \sqrt{\log \frac{K^2N}{\delta}} \frac{\alpha}{\sqrt{D}}, \forall 1 \leq i \leq KN, 1 \leq k \leq K;$

3. $\|\sum_{i \in \mathcal{I}_k} \boldsymbol{\varepsilon}_i\| \leq C \sqrt{\log \frac{K}{\delta}} \alpha \sqrt{N}, \forall 1 \leq k \leq K;$

4. $\sum_{i \in \mathcal{I}_k} \|\boldsymbol{\varepsilon}_i\|^2 \leq C \log \frac{K}{\delta} \alpha^2 N, \forall 1 \leq k \leq K,$

*for some universal constant $C > 0$.*

*Proof.* We proof relevent probabilities one by one:

1. By Lemma 3, we have

$$\mathbb{P}\left(\|\boldsymbol{\varepsilon}_i\| \geq t\right) \leq 4 \exp\left(-\frac{t^2}{8\alpha^2}\right) \, . \tag{E.9}$$

2. By Lemma 2, we have

$$\mathbb{P}\left(|\langle \boldsymbol{\mu}_k, \boldsymbol{\varepsilon}_i \rangle| \geq t\right) \leq 2 \exp\left(-\frac{Dt^2}{2\alpha^2}\right) \, . \tag{E.10}$$

3. Apply Lemma 3 to the vector $\sum_{i \in \mathcal{I}_k} \boldsymbol{\varepsilon}_i$, we have

$$\mathbb{P}\left(\left\|\sum_{i \in \mathcal{I}_k} \boldsymbol{\varepsilon}_i\right\| \geq t\right) \leq 4 \exp\left(-\frac{t^2}{8N\alpha^2}\right). \tag{E.11}$$

4. Apply Lemma 3 to the vector that is the concatenation of all $\boldsymbol{\varepsilon}_i$, $i \in \mathcal{N}_k$ and notice that its norm is equal to $\sqrt{\sum_{i \in \mathcal{I}_k} \|\boldsymbol{\varepsilon}_i\|^2}$, hence

$$\mathbb{P}\left(\sum_{i \in \mathcal{I}_k} \|\boldsymbol{\varepsilon}_i\|^2 \geq t^2\right) \leq 4 \exp\left(-\frac{t^2}{8N\alpha^2}\right). \tag{E.12}$$

Therefore,

$$\mathbb{P}\left(\|\boldsymbol{\varepsilon}_i\| \geq \sqrt{8 \log \frac{16KN}{\delta}}\alpha\right) \leq \frac{\delta}{4KN}, \qquad\qquad \forall 1 \leq i \leq KN,$$

$$\mathbb{P}\left(|\langle \boldsymbol{\mu}_k, \boldsymbol{\varepsilon}_i\rangle| \geq \sqrt{2 \log \frac{8K^2N}{\delta}}\frac{\alpha}{\sqrt{D}}\right) \leq \frac{\delta}{4K^2N}, \qquad\qquad \forall 1 \leq i \leq KN, 1 \leq k \leq K,$$

$$\mathbb{P}\left(\left\|\sum_{i \in \mathcal{I}_k} \boldsymbol{\varepsilon}_i\right\| \geq \sqrt{8 \log \frac{16K}{\delta}}\alpha\sqrt{N}\right) \leq \frac{\delta}{4K}, \qquad\qquad \forall 1 \leq k \leq K,$$

$$\mathbb{P}\left(\sum_{i \in \mathcal{I}_k} \|\boldsymbol{\varepsilon}_i\|^2 \geq 8 \log \frac{16K}{\delta}\alpha^2 N\right) \leq 4 \exp\left(-\frac{t^2}{8N\alpha^2}\right) \leq \frac{\delta}{4K}, \qquad\qquad \forall 1 \leq k \leq K.$$

The union bound shows that $\mathbb{P}\left(\mathcal{E}_{good}\right) \leq 1 - \delta$. $\qquad\qquad\qquad\square$

# F. PReLU converges to optimal $\ell_2$-robust classifier, Part Three: Alignment Phase

## F.1. Auxiliary lemmas

We need the following lemmas (proofs provided in Appendix H)

**Lemma 5.** *Given an initialization shape that satisfies Assumption 2 with non-degeneracy gap $\Delta > 0$, then for $j \in \mathcal{N}_k$, we have*

$$c_{kj}(0) = \cos(\boldsymbol{\mu}_k, \boldsymbol{w}_j(0)) \geq \sqrt{\frac{1}{2}\left(\frac{1}{(1-\Delta)^2} - 1\right)} := \tilde{\Delta}_1, \tag{F.1}$$

$$\frac{c_{lj}^{p-2}(0)}{c_{kj}^{p-2}(0)} \leq (1 - \sqrt{2\Delta})^{p-2} := 1 - \tilde{\Delta}_2, \forall l \neq k \text{ with } y_l = y_k \text{ and } c_{lj}(0) > 0 \tag{F.2}$$

**Lemma 6.** *Let $p > 2$. Condition on good event $\mathcal{E}_{good}$. Given some $1 \leq k \leq K$ and some $j \in \mathcal{N}_k$ and suppose the following is true at some point on the GF trajectory:*

*1. $c_{kj} \geq \tilde{\Delta}_1$;*

*2. $\frac{|c_{lj}|}{c_{kj}} \leq (1 - \sqrt{2\Delta}), \forall l \neq k.$*

*Then the following holds:*

$$\frac{d}{dt}c_{kj} \geq pc_{kj}^{p-1}\tilde{\Delta}_2(1 - c_{kj}) - C_1 \log\frac{K}{\delta}\alpha^2 - C_2 \max_k |f^{(p)}(\boldsymbol{\mu}_k; \boldsymbol{\theta}(t))|,$$

*for some universal constant $C_1, C_2$ that depends on $p$. If one further assume $c_{kj} \geq \sqrt{\frac{4}{5}}$, then the lower bound can be improved as*

$$\frac{d}{dt}c_{kj} \geq pc_{kj}^{p-1}\left(1 - \frac{1}{2^{p-2}}\right)(1 - c_{kj}) - C_1 \log\frac{K}{\delta}\alpha^2 - C_2 \max_i |f^{(p)}(\boldsymbol{x}_i; \boldsymbol{\theta})|,$$

**Lemma 7.** *Let $p > 2$. Condition on good event $\mathcal{E}_{good}$. Given an initialization shape that satisfies Assumption 2 with non-degeneracy gap $\Delta > 0$, define*

$$t_{1a} := \inf\left\{t : \max_i |f^{(p)}(\boldsymbol{x}_i; \boldsymbol{\theta}(t))| > \min\left\{\frac{\tilde{\Delta}_1^{p-1}\tilde{\Delta}_2(1 - \tilde{\Delta}_1)}{2^{p+1}}, \frac{\tilde{\Delta}_1^{p-1}\tilde{\Delta}_2(1 - \sqrt{2\Delta})}{2K2^{p+1}}\right\}\right\}. \tag{F.3}$$

*Then the following holds $\forall t \leq t_{1a}$:*

$$c_{kj}(t) \geq c_{kj}(0) \geq \tilde{\Delta}_1, \forall 1 \leq k \leq K, j \in \mathcal{N}_k, \tag{F.4}$$

*and*

$$\frac{|c_{lj}^{p-2}(t)|}{c_{kj}^{p-2}(t)} \leq \frac{|c_{lj}^{p-2}(0)|}{c_{kj}^{p-2}(0)} \leq 1 - \tilde{\Delta}_2 . \text{ and } \forall l \neq k, j \in \mathcal{N}_k . \tag{F.5}$$

**Lemma 8.** *Let $p > 2$. Condition on good event $\mathcal{E}_{good}$, then with any balanced initialization scale $\epsilon \leq \frac{1}{4\sqrt{h}W_{\max}^2}$, the solution to gradient flow dynamics satisfies*

$$\max_k |f^{(p)}(\boldsymbol{\mu}_k; \boldsymbol{\theta}(t))| \leq 2\epsilon\sqrt{h}W_{\max}^2, \quad \forall t \leq \frac{1}{2^{p+2}K}\log\left(\frac{1}{2^{p-1}\sqrt{h}\epsilon}\right). \tag{F.6}$$

The following lemma will be used to upper-bound the time each neuron spends until reaching a neighborhood of some data $\boldsymbol{\mu}_k$.

**Lemma 9.** *Let $p > 2$. Given some $C > 0$, if for some $z(t)$, the following holds*

$$\frac{d}{dt}z \geq Cz^{p-1}, \forall t \in [0, T], \; z(0) = z_0, \; z(T) = z_1, \tag{F.7}$$

*for some $0 < z_0 \leq z_1 < 1$. Then the travel time $T$ for $z(t)$ to go from $z_0$ to $z_1$ satifies:*

$$T \leq \frac{1}{(p-2)Cz_0^{p-2}}. \tag{F.8}$$

**Lemma 10.** *Let $p > 2$. Given some $C > 0$, if for some $z(t)$, the following holds*

$$\frac{d}{dt}z \geq C(1 - z), \forall t \in [0, T], \; z(0) = z_0, \; z(T) = z_1, \tag{F.9}$$

*for some $0 < z_0 \leq z_1 < 1$. Then the travel time $T$ for $z(t)$ to go from $z_0$ to $z_1$ satifies:*

$$T \leq \frac{1}{C} \log \frac{1}{1 - z_1}. \tag{F.10}$$

The following lemma will be used to lower-bound the time each neuron can stay around the neighborhood of some data $\boldsymbol{\mu}_k$.

### F.2. Proof of Proposition 4

**Proposition 2** (Restated). *Given the same assumptions as in Theorem 1 and consider the same GF solution $\boldsymbol{\theta}(t), t \geq 0$. There exist some $t_1 = \mathcal{O}\left(\log \frac{1}{\alpha}\right)$ and $t_2 = \mathcal{O}\left(\log \frac{1}{\epsilon}\right)$ such that $\forall k$ and $\forall j \in \mathcal{N}_k$, $\cos\left(\boldsymbol{\mu}_k, \boldsymbol{w}_j(t)\right) \geq 1 - \tilde{\mathcal{O}}(\alpha^2), \; \forall t \in [t_1, t_2]$.*

*Proof of Proposition 4.* **Breakdown the proofs** We let

$$t_1 := \inf\left\{t : \min_k \min_{j \in \mathcal{N}_k} c_{kj}(t) \geq 1 - C \log \frac{K}{\delta} \alpha^2\right\}. \tag{F.11}$$

We define

$$\epsilon_0 := \min\left\{ \frac{\tilde{\Delta}_1^{p-1}\tilde{\Delta}_2(1 - \tilde{\Delta}_1)}{2^{p+2}\sqrt{h}W_{\max}^2}, \right.$$
$$\frac{\tilde{\Delta}_1^{p-1}\tilde{\Delta}_2(1 - \sqrt{2\Delta})}{2K2^{p+2}\sqrt{h}W_{\max}^2},$$
$$\frac{p\tilde{\Delta}_1^{p-1}\tilde{\Delta}_2\alpha^2}{8\sqrt{h}W_{\max}^2},$$
$$\left. \frac{1}{\sqrt{h}}\exp\left(-4K\left(\frac{20}{(p-2)p\tilde{\Delta}_2\tilde{\Delta}_1^{p-2}} + \frac{2}{p(2^{p-1} - 2)}\log\frac{1}{C\log\frac{K}{\delta}\alpha^2}\right)\right)\right\}. \tag{F.12}$$

Our goal is to show that if the initialization scale $\epsilon \leq \epsilon_0$ (Notice that our assumption $\epsilon = \Theta(\alpha^{8K})$ can satisfies this inequality), then

1. $\min_k \min_{j \in \mathcal{N}_k} c_{kj}(t)$ grows above $1 - C \log \frac{K}{\delta}\alpha^2$ before
$\bar{t}_1 := \frac{20}{(p-2)p\tilde{\Delta}_2\tilde{\Delta}_1^{p-2}} + \frac{2}{p(2^{p-1} - 2)}\log\frac{1}{C\log\frac{K}{\delta}\alpha^2}$;

2. Any $c_{kj}(t)$ staying above $1 - C \log \frac{K}{\delta}\alpha^2$ during $[t_1, t_2]$, where $t_2 := \frac{1}{2^{p+2}K}\log\left(\frac{1}{2^{p-1}\sqrt{h}\epsilon}\right)$;

The remaining proof is to show them one by one.

**Upper bound on $t_1$** When $1 \leq k \leq K_1$, $j \in \mathcal{N}_k$ implies that $\boldsymbol{w}_{j0} \in \mathcal{R}_k$ and $\mathrm{sign}(v_j) = 1$. We shall primarily focus on this case as the proof is nearly identical for $K_1 + 1 \leq k \leq K$. We prove it by contradiction.

$\forall t \leq \bar{t}_1$, we have

$$\max_i |f^{(p)}(\boldsymbol{x}_i; \boldsymbol{\theta})| \leq \min \left\{ \frac{\tilde{\Delta}_1^{p-1} \tilde{\Delta}_2 (1 - \tilde{\Delta}_1)}{2^{p+1}}, \frac{\tilde{\Delta}_1^{p-1} \tilde{\Delta}_2 (1 - \sqrt{2\Delta})}{2K2^{p+1}} \right\} , \quad \text{(By Lemma 8 and (F.12))} \tag{F.13}$$

and

$$c_{kj}(t) \geq \tilde{\Delta}_1, \; \frac{c_{lj}^{p-2}}{c_{kj}^{p-2}} \leq 1 - \tilde{\Delta}_2 , \forall l \neq k, j \in \mathcal{N}_k . \qquad \text{(By (F.13) and Lemma 7)} \tag{F.14}$$

Suppose $t_1 \geq \bar{t}_1$, then $\exists k, j \in \mathcal{N}_k$ such that $t_{1j}^{(k)} := \inf\{t : c_{kj}(t) \geq 1 - \frac{\alpha^2}{2}\} > \bar{t}_1$. However, for $0 \leq t \leq \bar{t}_1$, we have, by Lemma 6, for this particular $k, j$,

Whenever $c_{kj} \geq \tilde{\Delta}_1$,

$$\frac{d}{dt} c_{kj} \geq p c_{kj}^{p-1} \tilde{\Delta}_2 (1 - c_{kj}) - C_1 \log \frac{K}{\delta} \alpha^2 - C_2 \max_k |f^{(p)}(\boldsymbol{\mu}_k; \boldsymbol{\theta}(t))| , \tag{F.15}$$

Whenever $c_{kj} \geq \sqrt{\frac{4}{5}}$,

$$\frac{d}{dt} c_{kj} \geq p c_{kj}^{p-1} \left(1 - \frac{1}{2^{p-2}}\right)(1 - c_{kj}) - C_1 \log \frac{K}{\delta} \alpha^2 - C_2 \max_k |f^{(p)}(\boldsymbol{\mu}_k; \boldsymbol{\theta}(t))| , \tag{F.16}$$

Notice that by Lemma 8 and (F.12), we have

$$\max_i |f^{(p)}(\boldsymbol{x}_i; \boldsymbol{\theta})| \leq \frac{p\tilde{\Delta}_1^{p-1} \tilde{\Delta}_2 \alpha^2}{4} \tag{F.17}$$

These suffices to show that $c_{kj}$ will reach $1 - \frac{C\alpha^2}{2}$ in less than $\bar{t}_1$ time.

For some choice of $C$ and sufficiently small $\alpha$, we have: Whenever, $\tilde{\Delta}_1 \leq c_{kj} \leq \sqrt{\frac{4}{5}}$,

$$\begin{aligned}
\frac{d}{dt} c_{kj} &\geq p c_{kj}^{p-1} \tilde{\Delta}_2 (1 - c_{kj}) - C_1 \log \frac{K}{\delta} \alpha^2 - C_2 \max_k |f^{(p)}(\boldsymbol{\mu}_k; \boldsymbol{\theta}(t))| \\
&\geq p c_{kj}^{p-1} \tilde{\Delta}_2 \left(1 - \sqrt{\frac{4}{5}}\right) - C_1 \log \frac{K}{\delta} \alpha^2 - C_2 \max_k |f^{(p)}(\boldsymbol{\mu}_k; \boldsymbol{\theta}(t))| \\
&\geq p c_{kj}^{p-1} \tilde{\Delta}_2 \left(1 - \sqrt{\frac{4}{5}}\right) - C_1 \log \frac{K}{\delta} \alpha^2 - C_2 \frac{p\tilde{\Delta}_1^{p-1} \tilde{\Delta}_2 \alpha^2}{4} . \\
&\geq \frac{p}{2} c_{kj}^{p-1} \tilde{\Delta}_2 \left(1 - \sqrt{\frac{4}{5}}\right) \geq \frac{p}{20} c_{kj}^{p-1} \tilde{\Delta}_2 ,
\end{aligned} \tag{F.18}$$

where we uses the fact that $c_{kj} \geq \tilde{\Delta}_1$ in the last inequality. Whenever, $\sqrt{\frac{4}{5}} \leq c_{kj} \leq 1 - \frac{C\alpha^2}{2}$,

$$\begin{aligned}
\frac{d}{dt} c_{kj} &\geq p c_{kj}^{p-1} \left(1 - \frac{1}{2^{p-2}}\right)(1 - c_{kj}) - C_1 \log \frac{K}{\delta} \alpha^2 - C_2 \max_k |f^{(p)}(\boldsymbol{\mu}_k; \boldsymbol{\theta}(t))| \\
&\geq p(2^{p-1} - 2)(1 - c_{kj}) - C_1 \log \frac{K}{\delta} \alpha^2 - C_2 \max_k |f^{(p)}(\boldsymbol{\mu}_k; \boldsymbol{\theta}(t))| \\
&\geq p(2^{p-1} - 2)(1 - c_{kj}) - C_1 \log \frac{K}{\delta} \alpha^2 - C_2 \frac{p\tilde{\Delta}_1^{p-1} \tilde{\Delta}_2 \alpha^2}{4} \\
&\geq \frac{p}{2}(2^{p-1} - 2)(1 - c_{kj}) ,
\end{aligned} \tag{F.19}$$

where we uses the fact that $c_{kj} \leq 1 - C \log \frac{K}{\delta} \alpha^2$ in the last inequality. The right-hand sides of (F.18) and (F.19) is positive, which proves that $c_{kj}$ is monotonically increasing before reaching $1 - C \log \frac{K}{\delta} \alpha^2$. Lastly,

1. by Lemma 9 and (F.18), it takes at most $\frac{20}{(p-2)p\tilde{\Delta}_2\tilde{\Delta}_1^{p-2}}$ time for $c_{kj}$ to travel from $\tilde{\Delta}_1$ to $\sqrt{\frac{4}{5}}$;

2. by Lemma 10 and (F.19), it takes at most $\frac{2}{p(2^{p-1}-2)}\log\frac{1}{C\log\frac{K}{\delta}\alpha^2}$ time for $c_{kj}$ to travel from $\sqrt{\frac{4}{5}}$ to $1-C\log\frac{K}{\delta}\alpha^2$.

Therefore, we have

$$t_{1j}^{(k)} := \inf\{t : c_{kj}(t) \geq 1 - \frac{\alpha^2}{2}\} \leq \frac{20}{(p-2)p\tilde{\Delta}_2\tilde{\Delta}_1^{p-2}} + \frac{2}{p(2^{p-1}-2)}\log\frac{1}{C\log\frac{K}{\delta}\alpha^2} = \bar{t}_1\,, \tag{F.20}$$

which contradicts our initial assumption that $c_{kj}(t) > \bar{t}_1$. Hence $t_1 \leq \bar{t}_1$.

**Maintaining** $C\log\frac{K}{\delta}\alpha^2$ **alignment until** $t_2$ We have shown that at some $t_1 \leq \bar{t}_1$, all $c_{kj}$ have grown above $1 - C\log\frac{K}{\delta}\alpha^2$. Now we show that any $c_{kj}(t)$ stays above $1 - C\log\frac{K}{\delta}\alpha^2$ between $[t_1, t_2]$. It suffices to show that for any $t \leq t_2$,

$$\left.\frac{d}{dt}c_{kj}\right|_{c_{kj}=1-C\log\frac{K}{\delta}\alpha^2} \geq 0\,. \tag{F.21}$$

Indeed, the inequality (F.16) is still valid before $t_2$, i.e.

$$\frac{d}{dt}c_{kj} \geq pc_{kj}^{p-1}\left(1-\frac{1}{2^{p-2}}\right)(1-c_{kj}) - C_1\log\frac{K}{\delta}\alpha^2 - C_2\max_k|f^{(p)}(\boldsymbol{\mu}_k;\boldsymbol{\theta}(t))|$$

$$\geq p(2^{p-1}-2)(1-c_{kj}) - C_1\log\frac{K}{\delta}\alpha^2 - C_2\frac{p\tilde{\Delta}_1^{p-1}\tilde{\Delta}_2\alpha^2}{4}\,.$$

Therefore, for some choice of $C$ and sufficiently small $\alpha$,

$$\left.\frac{d}{dt}c_{kj}\right|_{c_{kj}=1-C\log\frac{K}{\delta}\alpha^2} \geq p(2^{p-1}-2)C\log\frac{K}{\delta}\alpha^2 - C_1\log\frac{K}{\delta}\alpha^2 - C_2\frac{p\tilde{\Delta}_1^{p-1}\tilde{\Delta}_2\alpha^2}{4} \geq 0\,. \tag{F.22}$$

Hence

$$\min_k\min_{j\in\mathcal{N}_k}c_{kj}(t) \geq 1 - C\log\frac{K}{\delta}\alpha^2, \forall t \in [t_1, t_2]\,. \tag{F.23}$$

$\square$

# G. PReLU converges to optimal $\ell_2$-robust classifier, Part Four: Convergence Phase

## G.1. Axuiliary Lemmas

We need the following lemmas (proofs provided in Appendix H):

**Lemma 11.** *Let $p > 2$. Condition on good event $\mathcal{E}_{good}$. Suppose the following is true at some point on the GF trajectory:*

1. *$c_{kj}(t) \geq 1 - 2C_a \log \frac{K}{\delta} \alpha^2$, $\forall k, j \in \mathcal{N}_k$;*

2. *$\sum_{j \in \mathcal{N}_k} \|\boldsymbol{w}_j\|^2 \leq 1 + C_w \log \frac{K}{\delta} \alpha^2$, $\forall k$;*

3. *$\sum_{j \in \mathcal{N}_c} \|\boldsymbol{w}_j\|^2 = \tilde{o}(\alpha^2)$.*

*Then the following holds for every $1 \leq k \leq K$, $i \in \mathcal{I}_k$,*

$$f^{(p)}(\boldsymbol{x}_i; \boldsymbol{\theta}) \leq \sum_{j \in \mathcal{N}_k} \|\boldsymbol{w}_j\|^2 \left(1 + 2^{p+2}C\sqrt{\log \frac{K^2 N}{\delta}}\alpha^2\right) + 2KC\alpha^p;$$

$$f^{(p)}(\boldsymbol{x}_i; \boldsymbol{\theta}) \geq \sum_{j \in \mathcal{N}_k} \|\boldsymbol{w}_j\|^2 \left(1 - 4pC\sqrt{\log \frac{K^2 N}{\delta}}\alpha^2\right) - 2KC\alpha^p.$$

**Lemma 12.** *Let $p > 2$. Condition on good event $\mathcal{E}_{good}$. Suppose the following is true at some point on the GF trajectory:*

1. *$c_{kj}(t) \geq 1 - 2C_a \log \frac{K}{\delta} \alpha^2$, $\forall k, j \in \mathcal{N}_k$;*

2. *$\sum_{j \in \mathcal{N}_k} \|\boldsymbol{w}_j\|^2 \leq 1 + C_w \log \frac{K}{\delta} \alpha^2$, $\forall k$;*

3. *$\sum_{j \in \mathcal{N}_c} \|\boldsymbol{w}_j\|^2 = \tilde{o}(\alpha^2)$.*

*Furthermore, suppose additionally that for some $k, j \in \mathcal{N}_k$:*

$$1 - 2C_a \log \frac{K}{\delta}\alpha^2 \leq c_{kj}(t) \leq 1 - C_a \log \frac{K}{\delta}\alpha^2;$$

*Then the following holds for the same $k, j$,*

$$\frac{d}{dt}c_{kj} \geq -CK \log \frac{K^2 N}{\delta}\alpha^{\min\{p,4\}}.$$

**Lemma 13.** *Let $p > 2$. Condition on good event $\mathcal{E}_{good}$. Suppose the following is true at some point on the GF trajectory :*

1. *$c_{kj}(t) \geq 1 - 2C_a \log \frac{K}{\delta}\alpha^2$, $k, j \in \mathcal{N}_k$;*

2. *$\sum_{j \in \mathcal{N}_k} \|\boldsymbol{w}_j\|^2 \leq 1 + C_w \log \frac{K}{\delta}\alpha^2$, $\forall k$;*

3. *$\sum_{j \in \mathcal{N}_c} \|\boldsymbol{w}_j\|^2 = \tilde{o}(\alpha^2)$.*

*Then the following holds for every $1 \leq k \leq K$,*

$$\frac{d}{dt}\left(\sum_{j \in \mathcal{N}_k} \|\boldsymbol{w}_j\|^2\right) \leq 2\left(1 - \sum_{j \in \mathcal{N}_k} \|\boldsymbol{w}_j\|^2 + C \log \frac{K}{\delta}\alpha^2\right)\left(\sum_{j \in \mathcal{N}_k} \|\boldsymbol{w}_j\|^2\right),$$

*and*

$$\frac{d}{dt}\left(\sum_{j \in \mathcal{N}_k} \|\boldsymbol{w}_j\|^2\right) \geq 2\left(1 - \sum_{j \in \mathcal{N}_k} \|\boldsymbol{w}_j\|^2 - C \log \frac{K}{\delta}\alpha^2\right)\left(\sum_{j \in \mathcal{N}_k} \|\boldsymbol{w}_j\|^2\right),$$

*where $C$ is some universal constant such that $C < C_w$.*

**Lemma 14.** *Consider the same assumptions as in Proposition 4. Given the $t_1$ in Proposition 4, the following holds $\forall 1 \leq k \leq K$:*

$$\sum_{j \in \mathcal{N}_k} \|\boldsymbol{w}_j(t_1)\|^2 \geq \exp\left(-\frac{2p^{p+2}K}{p(p-2)\tilde{\Delta}_2 \tilde{\Delta}_1^{p-2}}\right) W_{\min}^2 \epsilon^2. \tag{G.1}$$

**Lemma 15.** *Given some $0 < \Delta < \frac{1}{4}$, if for some $z(t)$, the following holds*

$$\frac{d}{dt}z \geq (1 - z - \Delta)z, \ z(0) = z_0, \ z(T) = z_1, \tag{G.2}$$

*for some $0 < z_0 \leq \frac{1}{4}$, and $z_0 \leq z_1 < 1 - \Delta$. Then the travel time $T$ for $z(t)$ to go from $z_0$ to $z_1$ satisfies:*

$$T \leq 2\left(\log\frac{1}{1 - z_1 - \Delta} + \log\frac{1}{z_0}\right). \tag{G.3}$$

**Lemma 16.** *Condition on good event $\mathcal{E}_{good}$, we have*

$$\sum_{j \in \mathcal{N}_c} \|\boldsymbol{w}_j(t)\|^2 = \tilde{o}(\alpha^2), \ \forall t \leq T^*. \tag{G.4}$$

**Lemma 17.** *If the neurons $\{\boldsymbol{w}_j\}_{j=1}^h$ satisfies the following for some $0 \leq \delta \leq 1$ and $\nu, \zeta > 0$:*

- $\max_k \max_{j \in \mathcal{N}_k} c_{kj}(t) \geq 1 - \delta$;

- $\left|1 - \sum_{j \in \mathcal{N}_k} \|\boldsymbol{w}_j\|^2\right| \leq \nu$;

- $\sum_{j \in \mathcal{N}^c} \|\boldsymbol{w}_j\|^2 \leq \zeta$,

*then $\sup_{\boldsymbol{x} \in \mathbb{S}^{D-1}} \left|f^{(p)}(\boldsymbol{x}; \boldsymbol{\theta}) - F^{(p)}(\boldsymbol{x})\right| \leq K(1 + \nu)(2^p - 1)2\delta + K\nu + \zeta$*

### G.2. Proof of Theorem 1

**Theorem 3** (Restated). *Let $p > 2$. Given $0 \leq \delta \leq 1$ and a sufficiently small $\alpha_0^2$, assume the intra-cluster variance, the data dimension, and per-cluster sample size satisfy that $\alpha^2 \leq \alpha_0^2$, $D \geq \tilde{\Omega}(\alpha_0^{-2})$ and $\tilde{\Omega}(\alpha_0^{-2}) \leq N \leq \tilde{o}(\exp(\alpha_0^{-2}))$, respectively. Suppose that the initialization $\boldsymbol{\theta}(0)$ is balanced, $\epsilon$-small (Assumption 1) with $\epsilon = \tilde{\Theta}\left(\alpha_0^{8K}\right)$, and satisfies Assumption 2 with a non-degeneracy gap $\Delta = \Theta(1)$. Then with probability at least $1 - \delta$, the GF dynamics (7) with a sampled balanced dataset $\hat{\mathcal{D}} = \{\boldsymbol{x}_i, y_i\}_{i=1}^{KN}$, starting from $\boldsymbol{\theta}(0)$, has its solution $\boldsymbol{\theta}(t)$ satisfying that: for some $t^* = \tilde{\mathcal{O}}\left(\log\frac{1}{\alpha_0}\right)$ and $T^* = \tilde{\Theta}\left(\log\frac{1}{\alpha_0}\right) + \tilde{\Omega}\left(\alpha_0^{-\min\{p-2,2\}}\right)$ with $[t^*, T^*] \neq \emptyset$, we have $\mathcal{L}(\boldsymbol{\theta}(t)) = \tilde{\mathcal{O}}(\alpha_0^4), \forall t \in [t^*, T^*]$, and*

$$\sup_{t \in [t^*, T^*]} \sup_{\boldsymbol{x} \in \mathbb{S}^{D-1}} \left|f^{(p)}(\boldsymbol{x}; \boldsymbol{\theta}(t)) - F^{(p)}(\boldsymbol{x})\right| \leq \tilde{\mathcal{O}}(\alpha_0^2). \tag{G.5}$$

*Proof.* We have shown in Proposition 4, and Lemma 14 that:

1. Any $c_{kj}(t)$ staying above $1 - C_a \log\frac{K}{\delta}\alpha^2$ during $[t_1, t_2]$;

2. $\sum_{j \in \mathcal{N}_k} \|\boldsymbol{w}_j(t_1)\|^2 \geq \exp\left(-\frac{2p^{p+2}K}{p(p-2)\tilde{\Delta}_2 \tilde{\Delta}_1^{p-2}}\right) W_{\min}^2 \epsilon^2$, for every $1 \leq k \leq K$.

We define

$$t^* = \underbrace{t_1}_{\mathcal{O}(1)} + 2\left(\log\frac{1}{(C - C_w)\log\frac{K}{\delta}\alpha^2} + \frac{2p^{p+2}K}{p(p-2)\tilde{\Delta}_2\tilde{\Delta}_1^{p-2}} + \log\frac{1}{W_{\min}^2\epsilon^2}\right) \tag{G.6}$$

$$T^* = \underbrace{t_2}_{\Theta(\log\frac{1}{\epsilon})} + \frac{C}{\log\frac{K^2 N}{\delta}}\alpha^{\max\{2-p, -2\}} \tag{G.7}$$

Since $\epsilon = \Theta(\alpha^{8K})$. For sufficiently small $\alpha$, we have $\mathcal{O}(\log\frac{1}{\alpha}) = t^* \leq T^* = \Theta(\alpha^{\min\{2-p, -2\}})$. Our goal is to show that

1. Before $T^*$, one must have $\max_k \max_{j \in \mathcal{N}_k} c_{kj}(t) \geq 1 - 2C_a \log \frac{K}{\delta} \alpha^2$ and $\sum_{j \in \mathcal{N}_k} \|\boldsymbol{w}_j(t)\|^2 \leq 1 + C_w \log \frac{K}{\delta} \alpha^2$;

2. Before $T^*$, for all $k$, whenever $\sum_{j \in \mathcal{N}_k} \|\boldsymbol{w}_j(t)\|^2$ reaches $1 - C_w \log \frac{K}{\delta} \alpha^2$, it can not drop below $1 - C_w \log \frac{K}{\delta} \alpha^2$;

3. After $t^*$, for all $k$, one must have $\sum_{j \in \mathcal{N}_k} \|\boldsymbol{w}_j\|^2 \geq 1 - 2C_w \log \frac{K}{\delta} \alpha^2$.

We also have $\sum_{j \in \mathcal{N}_c} \|\boldsymbol{w}_j\|^2 = \tilde{o}(\alpha^2)$, then applying Lemma 17 gives the desired result. The statement that $\mathcal{L}(t) = \tilde{O}(\alpha^4)$ is due to the fact that $|y_i - f^{(p)}(\boldsymbol{x}_i; \boldsymbol{\theta}(t))| = \tilde{O}(\alpha^2)$ during $[t^*, T^*]$.

**First claim**: The two inequalities hold before $t_2$, thus it suffices to study

$$\tau_3 := \inf \left\{ t \geq t_2 : \max_k \max_{j \in \mathcal{N}_k} c_{kj}(t) \leq 1 - 2C \log \frac{K}{\delta} \alpha^2 \right\},$$

$$\tau_4 := \inf \left\{ t \geq t_2 : \sum_{j \in \mathcal{N}_k} \|\boldsymbol{w}_j\|^2 \geq 1 + C \log \frac{K}{\delta} \alpha^2 \right\},$$

and show that $\min\{\tau_3, \tau_4\} \geq T^*$. We proof it by contradiction, suppose $\min\{\tau_3, \tau_4\} \leq T^*$, then it must be either $\tau_3 = \min\{\tau_3, \tau_4\} \leq T^*$ or $\tau_4 = \min\{\tau_3, \tau_4\} \leq T^*$.

Consider the first case that $\tau_3 = \min\{\tau_3, \tau_4\} \leq T^*$, then there exists some $k$ and $j \in \mathcal{N}_k$ and some $\tau_{3-} \geq t_2$ such that

$$1 - 2C \log \frac{K}{\delta} \alpha^2 \leq c_{kj}(t) \leq 1 - C \log \frac{K}{\delta} \alpha^2, \forall t \in [\tau_{3-}, \tau_3], \tag{G.8}$$

$$c_{kj}(\tau_{3-}) = 1 - C \log \frac{K}{\delta} \alpha^2, \ c_{kj}(\tau_3) = 1 - 2C \log \frac{K}{\delta} \alpha^2 \tag{G.9}$$

since $c_{kj}(t)$ is continuous and has to travel from $1 - C \log \frac{K}{\delta} \alpha^2$ to $1 - 2C \log \frac{K}{\delta} \alpha^2$. By Lemma 6, we have

$$\frac{d}{dt} c_{kj} \geq -CK \log \frac{K^2 N}{\delta} \alpha^{\min\{p,4\}}, \forall t \in [\tau_{3-}, \tau_3].$$

Then by the fundamental theorem of calculus, we have

$$-C \log \frac{K}{\delta} \alpha^2 = c_{kj}(\tau_3) - c_{kj}(\tau_{3-}) = \int_{\tau_{3-}}^{\tau_3} \frac{d}{dt} c_{kj} \geq \int_{\tau_{3-}}^{\tau_3} -CK \log \frac{K^2 N}{\delta} \alpha^{\min\{p,4\}}$$

$$= -(\tau_3 - \tau_{3-}) CK \log \frac{K^2 N}{\delta} \alpha^{\min\{p,4\}}, \tag{G.10}$$

Therefore, for some constant $C > 0$,

$$(\tau_3 - \tau_{3-}) \geq \frac{C}{\log \frac{K^2 N}{\delta}} \alpha^{\max\{2-p,-2\}} \Rightarrow (\tau_3 - t_2) \geq \frac{C}{\log \frac{K^2 N}{\delta}} \alpha^{\max\{2-p,-2\}}. \tag{G.11}$$

$[t_2, \tau_3]$ has length at least $\frac{C}{\log \frac{K^2 N}{\delta}} \alpha^{\max\{2-p,-2\}}$ thus is an interval that contains $[t_2, T^*]$. Contradicting our assumption that $\tau_3 \leq T^*$. The case one is thus eliminated.

Consider the second case that $\tau_4 = \min\{\tau_3, \tau_4\} \leq T^*$, then by the continuity of $\|w_j\|$, we know that there exists some $k$ such that $\sum_{j \in \mathcal{N}_k} \|\boldsymbol{w}_j(\tau_4)\|^2 = 1 + C_w \log \frac{K}{\delta} \alpha^2$. However, by Lemma 11, we have, at $\tau_4$,

$$\frac{d}{dt} \left( \sum_{j \in \mathcal{N}_k} \|\boldsymbol{w}_j\|^2 \right) \leq 2 \left( 1 - \underbrace{\sum_{j \in \mathcal{N}_k} \|\boldsymbol{w}_j\|^2}_{=1 + C_w \log \frac{K}{\delta} \alpha^2} + C \log \frac{K}{\delta} \alpha^2 \right) \left( \sum_{j \in \mathcal{N}_k} \|\boldsymbol{w}_j\|^2 \right),$$

$$= 2 \left( (C - C_w) \log \frac{K}{\delta} \alpha^2 \right) \left( \sum_{j \in \mathcal{N}_k} \|\boldsymbol{w}_j\|^2 \right) < 0 \, ,$$

which indicates that $\sum_{j \in \mathcal{N}_k} \|\boldsymbol{w}_j\|^2$ can not surpass $1 + C_w \log \frac{K}{\delta} \alpha^2$ after $\tau_4$, violating the definition of $\tau_4$, leading to a contradiction. Therefore the second case is eliminated as well. We must have $\min\{\tau_3, \tau_4\} \geq T^*$. The first claim is proved.

**Second claim** By Lemma 13 (it applies to any $t \leq T^*$ given the proof in our first step), we have

$$\frac{d}{dt} \left( \sum_{j \in \mathcal{N}_k} \|\boldsymbol{w}_j\|^2 \right) \Bigg|_{\sum_{j \in \mathcal{N}_k} \|\boldsymbol{w}_j\|^2 = 1 - C_w \log \frac{K}{\delta} \alpha^2} \geq 2 \left( C_w \log \frac{K}{\delta} \alpha^2 - C \log \frac{K}{\delta} \alpha^2 \right) \left( \sum_{j \in \mathcal{N}_k} \|\boldsymbol{w}_j\|^2 \right) ,$$

$$\geq 0 \, .$$

Therefore, whenever $\sum_{j \in \mathcal{N}_k} \|\boldsymbol{w}_j(t)\|^2$ reaches $1 - C_w \log \frac{K}{\delta} \alpha^2$, it can not drop below $1 - C_w \log \frac{K}{\delta} \alpha^2$. The second claim is proved.

**Third claim** Lastly, we just need an upper bound on the travel time for $\sum_{j \in \mathcal{N}_k} \|\boldsymbol{w}_j(t)\|^2$ to go from $\sum_{j \in \mathcal{N}_k} \|\boldsymbol{w}_j(t_1)\|^2$ to $1 - C_w \log \frac{K}{\delta} \alpha^2$, for which we simply combine Lemma 13, 14, and 15 to see the travel time is upper bounded by

$$2 \left( \log \frac{1}{(C - C_w) \log \frac{K}{\delta} \alpha^2} + \frac{2p^{p+2}K}{p(p-2)\tilde{\Delta}_2 \tilde{\Delta}_1^{p-2}} + \log \frac{1}{W_{\min}^2 \epsilon^2} \right) . \tag{G.12}$$

Thus $\sum_{j \in \mathcal{N}_k} \|\boldsymbol{w}_j(t)\|^2$ must reach $1 - C_w \log \frac{K}{\delta} \alpha^2$ by $t^*$. $\qquad\square$

# H. PReLU converges to optimal $\ell_2$-robust classifier, Part Five: Proofs for auxiliary lemmas

**Lemma 5** (Restated). *Given an initialization shape that satisfies Assumption 2 with non-degeneracy gap $\Delta > 0$, then for $j \in \mathcal{N}_k$, we have*

$$c_{kj}(0) = \cos(\boldsymbol{\mu}_k, \boldsymbol{w}_j(0)) \geq \sqrt{\frac{1}{2}\left(\frac{1}{(1-\Delta)^2} - 1\right)} := \tilde{\Delta}_1, \tag{H.1}$$

$$\frac{c_{lj}^{p-2}(0)}{c_{kj}^{p-2}(0)} \leq (1 - \sqrt{2\Delta})^{p-2} := 1 - \tilde{\Delta}_2, \forall l \neq k \text{ with } y_l = y_k \text{ and } c_{lj}(0) > 0 \tag{H.2}$$

*Proof.* We prove both inequalities by contradiction.

**First inequality** Suppose $0 < c_{kj}(0) = \cos(\boldsymbol{\mu}_k, \boldsymbol{w}_j(0)) = \cos(\boldsymbol{\mu}_k, \boldsymbol{w}_{j0}) < \tilde{\Delta}_1$, then consider $\tilde{\boldsymbol{w}}_{j0} = \frac{\boldsymbol{w}_{j0}}{\|\boldsymbol{w}_{j0}\|}$, and

$$\tilde{\boldsymbol{w}} = \tilde{\boldsymbol{w}}_{j0} - \frac{c_{kj}(0)}{1 - c_{kj}(0)}(\boldsymbol{\mu}_k - \tilde{\boldsymbol{w}}_{j0}). \tag{H.3}$$

Notice that here $c_{kj}(0) = \cos(\boldsymbol{\mu}_k, \boldsymbol{w}_{j0}) = \langle \boldsymbol{\mu}_k, \tilde{\boldsymbol{w}}_{j0} \rangle$. It is easy to verify that $\langle \boldsymbol{\mu}_l, \tilde{\boldsymbol{w}} \rangle = 0, \forall 1 \leq l \leq K$, thus $\tilde{\boldsymbol{w}} \in \partial\left(\bigcup_{k \in \mathcal{K}} \mathcal{R}_k\right)$, and

$$d\left(\boldsymbol{w}_{j0}, \partial\left(\bigcup_{k \in \mathcal{K}} \mathcal{R}_k\right)\right) = 1 - \sup_{\boldsymbol{w} \in \partial\left(\bigcup_{k \in \mathcal{K}} \mathcal{R}_k\right)} \cos\left(\tilde{\boldsymbol{w}}_{j0}, \boldsymbol{w}\right) \leq 1 - \cos\left(\tilde{\boldsymbol{w}}_{j0}, \tilde{\boldsymbol{w}}\right), \tag{H.4}$$

Since one can compute

$$\cos(\tilde{\boldsymbol{w}}_{j0}, \tilde{\boldsymbol{w}}) = \frac{\langle \tilde{\boldsymbol{w}}_{j0}, \tilde{\boldsymbol{w}} \rangle}{\|\tilde{\boldsymbol{w}}_{j0}\|\|\tilde{\boldsymbol{w}}\|} = \frac{1 + c_{kj}(0)(1 - c_{kj}(0))}{\sqrt{1 + 2c_{kj}^2(0)}} \geq \frac{1}{\sqrt{1 + 2c_{kj}^2(0)}} > 1 - \Delta, \tag{H.5}$$

where the last inequality is due to our assumption that $c_{kj}(0) < \tilde{\Delta}_1$. Combining (H.4)(H.5), we have

$$d\left(\boldsymbol{w}_{j0}, \partial\left(\bigcup_{k \in \mathcal{K}} \mathcal{R}_k\right)\right) < \Delta, \tag{H.6}$$

which contradicts our assumption that the non-degeneracy gap is at least $\Delta$.

**Second inequality** Suppose there exists an $l \neq k$ such that $y_l = y_k$ and $\frac{c_{lj}^{p-2}(0)}{c_{kj}^{p-2}(0)} > (1 - \sqrt{2\Delta})^{p-2}$ and $c_{lj}(0) > 0$, we pick the $l$ that has the largest $c_{lj}(0)$, then consider $\tilde{\boldsymbol{w}}_{j0} = \frac{\boldsymbol{w}_{j0}}{\|\boldsymbol{w}_{j0}\|}$, and

$$\tilde{\boldsymbol{w}} = \tilde{\boldsymbol{w}}_{j0} - \frac{c_{kj}(0) - c_{lj}(0)}{2}(\boldsymbol{\mu}_k - \boldsymbol{\mu}_l). \tag{H.7}$$

It can be verified that $\|\tilde{\boldsymbol{w}}\| = 1$, $\cos(\boldsymbol{\mu}_k, \tilde{\boldsymbol{w}}) = \cos(\boldsymbol{\mu}_l, \tilde{\boldsymbol{w}}) = \frac{c_{kj}(0) + c_{lj}(0)}{2}$, and $\cos(\boldsymbol{\mu}_m, \tilde{\boldsymbol{w}}) = \cos(\boldsymbol{\mu}_m, \tilde{\boldsymbol{w}}_{j0}) \leq \cos(\boldsymbol{\mu}_l, \tilde{\boldsymbol{w}}), \forall m \neq k$ or $l$. All of the above together implies $\tilde{\boldsymbol{w}} \in (\partial\mathcal{R}_k) \cap (\partial\mathcal{R}_l) \subset \partial\left(\bigcup_{k \in \mathcal{K}} \mathcal{R}_k\right)$, and

$$d\left(\boldsymbol{w}_{j0}, \partial\left(\bigcup_{k \in \mathcal{K}} \mathcal{R}_k\right)\right) = 1 - \sup_{\boldsymbol{w} \in \partial\left(\bigcup_{k \in \mathcal{K}} \mathcal{R}_k\right)} \cos\left(\tilde{\boldsymbol{w}}_{j0}, \boldsymbol{w}\right) \leq 1 - \cos\left(\tilde{\boldsymbol{w}}_{j0}, \tilde{\boldsymbol{w}}\right), \tag{H.8}$$

One can compute

$$\cos(\tilde{\boldsymbol{w}}_{j0}, \tilde{\boldsymbol{w}}) = \frac{\langle \tilde{\boldsymbol{w}}_{j0}, \tilde{\boldsymbol{w}} \rangle}{\|\tilde{\boldsymbol{w}}_{j0}\|\|\tilde{\boldsymbol{w}}\|} = 1 - \frac{(c_{kj}(0) - c_{lj}(0))^2}{2}$$

$$\geq 1 - \frac{\left(1 - \frac{c_{lj}(0)}{c_{kj}(0)}\right)^2}{2}$$

$$\geq 1 - \frac{\left(1 - \left(\frac{c_{lj}^{p-2}(0)}{c_{kj}^{p-2}(0)}\right)^{\frac{1}{p-2}}\right)^2}{2}$$

$$\geq 1 - \frac{\left(1 - (1 - \tilde{\Delta}_2)^{\frac{1}{p-2}}\right)^2}{2} = 1 - \Delta, \tag{H.9}$$

where the last inequality is due to our assumption that $\frac{c_{lj}^{p-2}(0)}{c_{kj}^{p-2}(0)} > (1 - \sqrt{2\Delta})^{p-2}$. Combining (H.4)(H.5), we have

$$d\left(\boldsymbol{w}_{j0}, \partial\left(\bigcup_{k \in \mathcal{K}} \mathcal{R}_k\right)\right) < \Delta, \tag{H.10}$$

which contradicts our assumption that the non-degeneracy gap is at least $\Delta$. $\square$

**Lemma 6** (Restated). *Let $p > 2$. Condition on good event $\mathcal{E}_{good}$. Given some $1 \leq k \leq K$ and some $j \in \mathcal{N}_k$ and suppose the following is true at some point on the GF trajectory:*

1. $c_{kj} \geq \tilde{\Delta}_1$;

2. $\frac{|c_{lj}|}{c_{kj}} \leq (1 - \sqrt{2\Delta}), \forall l \neq k.$

*Then the following holds:*

$$\frac{d}{dt}c_{kj} \geq pc_{kj}^{p-1}\tilde{\Delta}_2(1 - c_{kj}) - C_1 \log\frac{K}{\delta}\alpha^2 - C_2 \max_k |f^{(p)}(\boldsymbol{\mu}_k; \boldsymbol{\theta}(t))|,$$

*for some universal constant $C_1, C_2$ that depends on $p$. If one further assume $c_{kj} \geq \sqrt{\frac{4}{5}}$, then the lower bound can be improved as*

$$\frac{d}{dt}c_{kj} \geq pc_{kj}^{p-1}\left(1 - \frac{1}{2^{p-2}}\right)(1 - c_{kj}) - C_1 \log\frac{K}{\delta}\alpha^2 - C_2 \max_i |f^{(p)}(\boldsymbol{x}_i; \boldsymbol{\theta})|,$$

*Proof.* When $1 \leq k \leq K_1$, $j \in \mathcal{N}_k$ implies that $j \in \mathcal{N}_+$ thus $\mathrm{sign}(v_j) = 1$. We shall primarily focus on this case as the proof is nearly identical for $K_1 + 1 \leq k \leq K$.

$$\frac{d}{dt}c_{kj}$$

$$= -\frac{1}{N}\sum_{i:\langle\boldsymbol{x}_i,\boldsymbol{w}_j\rangle>0}\nabla_{\hat{y}}\ell_i\, p\left(\left\langle\boldsymbol{x}_i, \frac{\boldsymbol{w}_j}{\|\boldsymbol{w}_j\|}\right\rangle\right)^{p-1}\left(\langle\boldsymbol{\mu}_k, \boldsymbol{x}_i\rangle - \left\langle\boldsymbol{x}_i, \frac{\boldsymbol{w}_j}{\|\boldsymbol{w}_j\|}\right\rangle c_{kj}\right)$$

$$= \frac{1}{N}\sum_{i:\langle\boldsymbol{x}_i,\boldsymbol{w}_j\rangle>0}(y_i - f^{(p)}(\boldsymbol{x}_i; \boldsymbol{\theta}))\, p\left(\left\langle\boldsymbol{x}_i, \frac{\boldsymbol{w}_j}{\|\boldsymbol{w}_j\|}\right\rangle\right)^{p-1}\left(\langle\boldsymbol{\mu}_k, \boldsymbol{x}_i\rangle - \left\langle\boldsymbol{x}_i, \frac{\boldsymbol{w}_j}{\|\boldsymbol{w}_j\|}\right\rangle c_{kj}\right)$$

$$= \frac{1}{N}\sum_{i:\langle\boldsymbol{x}_i,\boldsymbol{w}_j\rangle>0}y_i\, p\left(\left\langle\boldsymbol{x}_i, \frac{\boldsymbol{w}_j}{\|\boldsymbol{w}_j\|}\right\rangle\right)^{p-1}\left(\langle\boldsymbol{\mu}_k, \boldsymbol{x}_i\rangle - \left\langle\boldsymbol{x}_i, \frac{\boldsymbol{w}_j}{\|\boldsymbol{w}_j\|}\right\rangle c_{kj}\right)$$

$$\underbrace{- \frac{1}{N}\sum_{i:\langle\boldsymbol{x}_i,\boldsymbol{w}_j\rangle>0}f^{(p)}(\boldsymbol{x}_i; \boldsymbol{\theta})\, p\left(\left\langle\boldsymbol{x}_i, \frac{\boldsymbol{w}_j}{\|\boldsymbol{w}_j\|}\right\rangle\right)^{p-1}\left(\langle\boldsymbol{\mu}_k, \boldsymbol{x}_i\rangle - \left\langle\boldsymbol{x}_i, \frac{\boldsymbol{w}_j}{\|\boldsymbol{w}_j\|}\right\rangle c_{kj}\right)}_{:=\Gamma_1\,(\text{will be treated later})}$$

$$= \left(\underbrace{\frac{1}{N}\sum_{i\in\mathcal{I}_k:\langle\boldsymbol{x}_i,\boldsymbol{w}_j\rangle>0}y_i\, p\left(\left\langle\boldsymbol{x}_i, \frac{\boldsymbol{w}_j}{\|\boldsymbol{w}_j\|}\right\rangle\right)^{p-1}\left(\langle\boldsymbol{\mu}_k, \boldsymbol{x}_i\rangle - \left\langle\boldsymbol{x}_i, \frac{\boldsymbol{w}_j}{\|\boldsymbol{w}_j\|}\right\rangle c_{kj}\right)}_{(a)}\right.$$

$$-\frac{1}{N}\underbrace{\sum_{l\neq k}\sum_{i\in\mathcal{I}_l:\langle \boldsymbol{x}_i,\boldsymbol{w}_j\rangle>0} y_i\, p\left(\left\langle \boldsymbol{x}_i,\frac{\boldsymbol{w}_j}{\|\boldsymbol{w}_j\|}\right\rangle\right)^{p-1}\left(\langle\boldsymbol{\mu}_k,\boldsymbol{x}_i\rangle-\left\langle \boldsymbol{x}_i,\frac{\boldsymbol{w}_j}{\|\boldsymbol{w}_j\|}\right\rangle c_{kj}\right)}_{(b)}\Bigg) + \Gamma_1 \tag{H.11}$$

We handle these two terms differently:

$$
\begin{aligned}
(a) &= \frac{1}{N}\sum_{i\in\mathcal{I}_k:\langle \boldsymbol{x}_i,\boldsymbol{w}_j\rangle>0} y_i\, p\left(\left\langle \boldsymbol{x}_i,\frac{\boldsymbol{w}_j}{\|\boldsymbol{w}_j\|}\right\rangle\right)^{p-1}\left(\langle\boldsymbol{\mu}_k,\boldsymbol{x}_i\rangle-\left\langle \boldsymbol{x}_i,\frac{\boldsymbol{w}_j}{\|\boldsymbol{w}_j\|}\right\rangle c_{kj}\right)\\
&= \frac{1}{N}\sum_{i\in\mathcal{I}_k} p\left(\left\langle \boldsymbol{\mu}_k+\boldsymbol{\varepsilon}_i,\frac{\boldsymbol{w}_j}{\|\boldsymbol{w}_j\|}\right\rangle\right)^{p-1}\left(\langle\boldsymbol{\mu}_k,\boldsymbol{\mu}_k+\boldsymbol{\varepsilon}_i\rangle-\left\langle \boldsymbol{\mu}_k+\boldsymbol{\varepsilon}_i,\frac{\boldsymbol{w}_j}{\|\boldsymbol{w}_j\|}\right\rangle c_{kj}\right)\\
&= \frac{1}{N}\sum_{i\in\mathcal{I}_k} p\left(\left\langle \boldsymbol{\mu}_k+\boldsymbol{\varepsilon}_i,\frac{\boldsymbol{w}_j}{\|\boldsymbol{w}_j\|}\right\rangle\right)^{p-1}\left(1-c_{kj}^2+\langle\boldsymbol{\mu}_k,\boldsymbol{\varepsilon}_i\rangle-\left\langle \boldsymbol{\varepsilon}_i,\frac{\boldsymbol{w}_j}{\|\boldsymbol{w}_j\|}\right\rangle c_{kj}\right)\\
&= \frac{1}{N}\sum_{i\in\mathcal{I}_k} p\left(\left\langle \boldsymbol{\mu}_k+\boldsymbol{\varepsilon}_i,\frac{\boldsymbol{w}_j}{\|\boldsymbol{w}_j\|}\right\rangle\right)^{p-1}\left(1-c_{kj}^2-\left\langle \boldsymbol{\varepsilon}_i,\frac{\boldsymbol{w}_j}{\|\boldsymbol{w}_j\|}\right\rangle c_{kj}\right)\\
&\qquad\qquad + \underbrace{\frac{1}{N}\sum_{i\in\mathcal{I}_k} y_i\, p\left(\left\langle \boldsymbol{\mu}_k+\boldsymbol{\varepsilon}_i,\frac{\boldsymbol{w}_j}{\|\boldsymbol{w}_j\|}\right\rangle\right)^{p-1}\langle\boldsymbol{\mu}_k,\boldsymbol{\varepsilon}_i\rangle}_{:=\Gamma_2\,(\text{will be treated later})}\\
&= \frac{1}{N}\sum_{i\in\mathcal{I}_k} p\left(c_{kj}+\left\langle \boldsymbol{\varepsilon}_i,\frac{\boldsymbol{w}_j}{\|\boldsymbol{w}_j\|}\right\rangle\right)^{p-1}\left(1-c_{kj}^2-\left\langle \boldsymbol{\varepsilon}_i,\frac{\boldsymbol{w}_j}{\|\boldsymbol{w}_j\|}\right\rangle c_{kj}\right)+\Gamma_2
\end{aligned}
\tag{H.12}
$$

With the Taylor expansion

$$\left(c_{kj}+\left\langle \boldsymbol{\varepsilon}_i,\frac{\boldsymbol{w}_j}{\|\boldsymbol{w}_j\|}\right\rangle\right)^{p-1}=c_{kj}^{p-1}+(p-1)c_{kj}^{p-2}\left\langle \boldsymbol{\varepsilon}_i,\frac{\boldsymbol{w}_j}{\|\boldsymbol{w}_j\|}\right\rangle+R_L\left|\left\langle \boldsymbol{\varepsilon}_i,\frac{\boldsymbol{w}_j}{\|\boldsymbol{w}_j\|}\right\rangle\right|^2, \tag{H.13}$$

where $R_L=\frac{(p-1)(p-2)(c_{kj}+\zeta_L)^{p-2}}{2}$ and $\zeta_L$ between $0$ and $\left\langle \boldsymbol{\varepsilon}_i,\frac{\boldsymbol{w}_j}{\|\boldsymbol{w}_j\|}\right\rangle$ comes from the Lagrange residual. Clearly $|R_L|\leq 2^{p-3}p^2$. Combining (H.12)(H.13), we have

$$
\begin{aligned}
&(a)\\
&= (\text{H.12})\\
&= pc_{kj}^{p-1}(1-c_{kj}^2)+\left(-pc_{kj}^p+p(p-1)c_{kj}^{p-2}(1-c_{kj}^2)\right)\sum_{i\in\mathcal{I}_k}\left\langle \boldsymbol{\varepsilon}_i,\frac{\boldsymbol{w}_j}{\|\boldsymbol{w}_j\|}\right\rangle\\
&\qquad\qquad +\frac{1}{N}\left(-p(p-1)c_{kj}+pR_L(1-c_{kj}^2)\right)\sum_{i\in\mathcal{I}_k}\left|\left\langle \boldsymbol{\varepsilon}_i,\frac{\boldsymbol{w}_j}{\|\boldsymbol{w}_j\|}\right\rangle\right|^2\\
&\qquad\qquad -\frac{1}{N}pc_{kj}R_L\sum_{i\in\mathcal{I}_k}\left|\left\langle \boldsymbol{\varepsilon}_i,\frac{\boldsymbol{w}_j}{\|\boldsymbol{w}_j\|}\right\rangle\right|^2\left\langle \boldsymbol{\varepsilon}_i,\frac{\boldsymbol{w}_j}{\|\boldsymbol{w}_j\|}\right\rangle+\Gamma_2\\
&\geq pc_{kj}^{p-1}(1-c_{kj}^2)-\frac{1}{N}p^2\left\|\sum_{i\in\mathcal{I}_k}\boldsymbol{\varepsilon}_i\right\|-\frac{1}{N}2^{p-1}p^2\sum_{i\in\mathcal{I}_k}\|\boldsymbol{\varepsilon}_i\|^2-\frac{1}{N}2^{p-3}p^3\sum_{i\in\mathcal{I}_k}\|\boldsymbol{\varepsilon}_i\|^3+\Gamma_2\\
&\geq pc_{kj}^{p-1}(1-c_{kj}^2)-\frac{1}{N}p^2\left\|\sum_{i\in\mathcal{I}_k}\boldsymbol{\varepsilon}_i\right\|-\frac{1}{N}2^{p-1}p^2\sum_{i\in\mathcal{I}_k}\|\boldsymbol{\varepsilon}_i\|^2-\frac{1}{N}2^{p-3}p^3\sum_{i\in\mathcal{I}_k}\|\boldsymbol{\varepsilon}_i\|^2\max_i\|\boldsymbol{\varepsilon}_i\|+\Gamma_2\\
&\geq pc_{kj}^{p-1}(1-c_{kj}^2)-Cp^2\sqrt{\log\frac{K}{\delta}}\frac{\alpha}{\sqrt{N}}-2^{p-1}p^3C^2\log\frac{K}{\delta}\alpha^2-2^{p-3}p^3C^3\log\frac{K^2N}{\delta}\alpha^3+\Gamma_2\,.
\end{aligned}
\tag{H.14}
$$

We leave the bound as the last one for now and turn to the other term:

$$(b)$$

$$= -\frac{1}{N} \sum_{l \neq k} \sum_{i \in \mathcal{I}_l : \langle \boldsymbol{x}_i, \boldsymbol{w}_j \rangle > 0} y_i \, p \left( \left\langle \boldsymbol{x}_i, \frac{\boldsymbol{w}_j}{\|\boldsymbol{w}_j\|} \right\rangle \right)^{p-1} \left( \langle \boldsymbol{\mu}_k, \boldsymbol{x}_i \rangle - \left\langle \boldsymbol{x}_i, \frac{\boldsymbol{w}_j}{\|\boldsymbol{w}_j\|} \right\rangle c_{kj} \right)$$

$$= \frac{1}{N} \sum_{l \neq k} \sum_{i \in \mathcal{I}_l : \langle \boldsymbol{x}_i, \boldsymbol{w}_j \rangle > 0} y_i \, p \left( c_{lj} + \left\langle \boldsymbol{\varepsilon}_i, \frac{\boldsymbol{w}_j}{\|\boldsymbol{w}_j\|} \right\rangle \right)^{p} c_{kj}$$

$$\underbrace{- \frac{1}{N} \sum_{l \neq k} \sum_{i \in \mathcal{I}_l : \langle \boldsymbol{x}_i, \boldsymbol{w}_j \rangle > 0} y_i \, p \left( c_{lj} + \left\langle \boldsymbol{\varepsilon}_i, \frac{\boldsymbol{w}_j}{\|\boldsymbol{w}_j\|} \right\rangle \right)^{p-1} \langle \boldsymbol{\mu}_k, \boldsymbol{\varepsilon}_i \rangle}_{:= \Gamma_3 \, (\text{will be treated later})}$$

$$\geq -\frac{1}{N} \sum_{l \neq k} \sum_{i \in \mathcal{I}_l} p \left( |c_{lj}| + \left| \left\langle \boldsymbol{\varepsilon}_i, \frac{\boldsymbol{w}_j}{\|\boldsymbol{w}_j\|} \right\rangle \right| \right)^{p} c_{kj} + \Gamma_3 \tag{H.15}$$

With the Taylor expansion

$$\left( |c_{lj}| + \left| \left\langle \boldsymbol{\varepsilon}_i, \frac{\boldsymbol{w}_j}{\|\boldsymbol{w}_j\|} \right\rangle \right| \right)^{p} = |c_{lj}|^p + p|c_{lj}|^{p-1} \left| \left\langle \boldsymbol{\varepsilon}_i, \frac{\boldsymbol{w}_j}{\|\boldsymbol{w}_j\|} \right\rangle \right| + R_L \left| \left\langle \boldsymbol{\varepsilon}_i, \frac{\boldsymbol{w}_j}{\|\boldsymbol{w}_j\|} \right\rangle \right|^2, \tag{H.16}$$

where $R_L = \frac{p(p-1)(|c_{lj}| + \zeta_L)^{p-2}}{2}$ and $\zeta_L$ between 0 and $\left| \left\langle \boldsymbol{\varepsilon}_i, \frac{\boldsymbol{w}_j}{\|\boldsymbol{w}_j\|} \right\rangle \right|$ comes from the Lagrange residual. Clearly $|R_L| \leq 2^{p-2} p^2$. Combining (H.12)(H.13), we have

$$(b)$$

$$= (\text{H.15})$$

$$= -\sum_{l \neq k} p|c_{lj}|^p c_{kj} - \frac{1}{N} \sum_{l \neq k} \sum_{i \in \mathcal{I}_l} p(p-1)|c_{lj}|^{p-1} c_{kj} \left| \left\langle \boldsymbol{\varepsilon}_i, \frac{\boldsymbol{w}_j}{\|\boldsymbol{w}_j\|} \right\rangle \right| - \frac{1}{N} \sum_{l \neq k} \sum_{i \in \mathcal{I}_l} p R_L c_{kj} \left| \left\langle \boldsymbol{\varepsilon}_i, \frac{\boldsymbol{w}_j}{\|\boldsymbol{w}_j\|} \right\rangle \right|^2 + \Gamma_3$$

$$\geq -\sum_{l \neq k} p|c_{lj}|^p c_{kj} - \frac{1}{N} \sum_{l \neq k} \sum_{i \in \mathcal{I}_l} p(p-1)|c_{lj}|^{p-1} c_{kj} \left| \left\langle \boldsymbol{\varepsilon}_i, \frac{\boldsymbol{w}_j}{\|\boldsymbol{w}_j\|} \right\rangle \right| - K 2^{p-2} p^3 C^2 \log \frac{K}{\delta} \alpha^2 + \Gamma_3,$$

$$\geq -\sum_{l \neq k} p|c_{lj}|^p c_{kj} - \frac{1}{N} \sum_{l \neq k} \sum_{i \in \mathcal{I}_l} p(p-1)|c_{lj}|^{p-1} c_{kj} \left( \|\boldsymbol{\varepsilon}_i\| \sqrt{1 - c_{kj}^2} + |\langle \boldsymbol{\varepsilon}_i, \boldsymbol{\mu}_k \rangle| \right) - K 2^{p-2} p^3 C^2 \log \frac{K}{\delta} \alpha^2 + \Gamma_3,$$

$$\geq -\sum_{l \neq k} p|c_{lj}|^p c_{kj} - \sum_{l \neq k} p^2 |c_{lj}|^{p-1} c_{kj} C \sqrt{\log \frac{K^2 N}{\delta}} \alpha \sqrt{1 - c_{kj}^2} - K 2^{p-2} p^3 C^2 \log \frac{K}{\delta} \alpha^2 + \Gamma_3$$

$$+ \underbrace{-\frac{1}{N} \sum_{l \neq k} \sum_{i \in \mathcal{I}_l} p(p-1)|c_{lj}|^{p-1} c_{kj} |\langle \boldsymbol{\varepsilon}_i, \boldsymbol{\mu}_k \rangle|}_{:= \Gamma_4 \, (\text{will be treated later})}$$

$$= -p c_{kj}^{p-1} \left( \sum_{l \neq k} \frac{|c_{lj}|^{p-2}}{c_{kj}^{p-2}} |c_{lj}|^2 - \sum_{l \neq k} p \frac{|c_{lj}|^{p-2}}{c_{kj}^{p-2}} |c_{lj}| C \sqrt{\log \frac{K^2 N}{\delta}} \alpha \sqrt{1 - c_{kj}^2} \right) - K 2^{p-2} p^3 C^2 \log \frac{K}{\delta} \alpha^2 + \Gamma_3 + \Gamma_4$$

$$= -p c_{kj}^{p-1} \left( \max_{l \neq k} \frac{|c_{lj}|^{p-2}}{c_{kj}^{p-2}} \right) \left( \sum_{l \neq k} |c_{lj}|^2 - \sum_{l \neq k} p|c_{lj}| C \sqrt{\log \frac{K^2 N}{\delta}} \alpha \sqrt{1 - c_{kj}^2} \right) - K 2^{p-2} p^3 C^2 \log \frac{K}{\delta} \alpha^2 + \Gamma_3 + \Gamma_4$$

$$\geq -p c_{kj}^{p-1} \left( \max_{l \neq k} \frac{|c_{lj}|^{p-2}}{c_{kj}^{p-2}} \right) \left( \sum_{l \neq k} |c_{lj}|^2 - pC \sqrt{\log \frac{K^2 N}{\delta}} \alpha \sqrt{1 - c_{kj}^2} \sum_{l \neq k} |c_{lj}| \right) - K 2^{p-2} p^3 C^2 \log \frac{K}{\delta} \alpha^2 + \Gamma_3 + \Gamma_4$$

$$\geq -p c_{kj}^{p-1} \left( \max_{l \neq k} \frac{|c_{lj}|^{p-2}}{c_{kj}^{p-2}} \right) \left( \sum_{l \neq k} |c_{lj}|^2 - pC \sqrt{K \log \frac{K^2 N}{\delta}} \alpha \sqrt{1 - c_{kj}^2} \sqrt{\sum_{l \neq k} |c_{lj}|^2} \right) - K 2^{p-2} p^3 C^2 \log \frac{K}{\delta} \alpha^2 + \Gamma_3 + \Gamma_4$$

$$\geq -pc_{kj}^{p-1} \left( \max_{l\neq k} \frac{|c_{lj}|^{p-2}}{c_{kj}^{p-2}} (1-c_{kj}^2) \right) \left( 1 - pC\sqrt{K\log\frac{K^2N}{\delta}}\alpha(1-c_{kj}^2) \right) - K2^{p-2}p^3C^2\log\frac{K}{\delta}\alpha^2 + \Gamma_3 + \Gamma_4$$

$$\geq -\frac{p}{2}c_{kj}^{p-1} \left( \max_{l\neq k} \frac{|c_{lj}|^{p-2}}{c_{kj}^{p-2}} \right)(1-c_{kj}^2) - K2^{p-2}p^3C^2\log\frac{K}{\delta}\alpha^2 + \Gamma_3 + \Gamma_4 \qquad\text{(H.17)}$$

Finally, combining (H.14)(H.17), we have

$$\begin{aligned}
\frac{d}{dt}c_{kj} &\geq pc_{kj}^{p-1}\left(1 - \max_{l\neq k}\frac{|c_{lj}|^{p-2}}{c_{kj}^{p-2}}\right)(1-c_{kj}^2) - C_1'\sqrt{\log\frac{K}{\delta}}\frac{\alpha}{\sqrt{N}} - C_2'\log\frac{K}{\delta}\alpha^2 - C_3'\log\frac{K^2N}{\delta}\alpha^3 \\
&\quad - |\Gamma_1| - |\Gamma_2| - |\Gamma_3| - |\Gamma_4| \\
&\geq pc_{kj}^{p-1}\left(1 - \max_{l\neq k}\frac{|c_{lj}|^{p-2}}{c_{kj}^{p-2}}\right)(1-c_{kj}) - C_1'\sqrt{\log\frac{K}{\delta}}\frac{\alpha}{\sqrt{N}} - C_2'\log\frac{K}{\delta}\alpha^2 - C_3'\log\frac{K^2N}{\delta}\alpha^3 \\
&\quad - |\Gamma_1| - |\Gamma_2| - |\Gamma_3| - |\Gamma_4|,
\end{aligned}$$

where the readers should be able to find universal constants $C_1', C_2', C_3'$ from the derivation. It remains to bound these $|\Gamma_i|, i = 1, \cdots, 4$. Indeed, we can find the following bound:

$$\begin{aligned}
|\Gamma_1| &= \left| \frac{1}{N} \sum_{i:\langle x_i, w_j\rangle > 0} f^{(p)}(x_i; \theta)\, p\left(\left\langle x_i, \frac{w_j}{\|w_j\|}\right\rangle\right)^{p-1}\left(\langle \mu_k, x_i\rangle - \left\langle x_i, \frac{w_j}{\|w_j\|}\right\rangle c_{kj}\right) \right| \\
&\leq \max_i |f^{(p)}(x_i;\theta)| \frac{1}{N}\sum_{i:\langle x_i, w_j\rangle > 0} \left| p\|x_i\|^{p-1}(2\|x_i\|) \right| \\
&\leq p2^{p+1}\max_i |f^{(p)}(x_i;\theta)|,
\end{aligned}$$

$$\begin{aligned}
|\Gamma_2| &= \left| \frac{1}{N}\sum_{i\in\mathcal{I}_k} y_i\, p\left(\left\langle \mu_k + \varepsilon_i, \frac{w_j}{\|w_j\|}\right\rangle\right)^{p-1}\langle \mu_k, \varepsilon_i\rangle \right| \\
&\leq \frac{1}{N}\sum_{i\in\mathcal{I}_k} p\|x_i\|^{p-1}|\langle \mu_k, \varepsilon_i\rangle| \leq p2^{p-1}C\sqrt{\log\frac{K^2N}{\delta}}\frac{\alpha}{\sqrt{D}}
\end{aligned}$$

$$\begin{aligned}
|\Gamma_3| &= \left| -\frac{1}{N}\sum_{l\neq k}\sum_{i\in\mathcal{I}_l:\langle x_i, w_j\rangle > 0} y_i\, p\left(c_{lj} + \left\langle \varepsilon_i, \frac{w_j}{\|w_j\|}\right\rangle\right)^{p-1}\langle \mu_k, \varepsilon_i\rangle \right| \\
&\leq \frac{1}{N}\sum_{i\in\mathcal{I}_k} p\|x_i\|^{p-1}|\langle \mu_k, \varepsilon_i\rangle| \leq p2^{p-1}C\sqrt{\log\frac{K^2N}{\delta}}\frac{\alpha}{\sqrt{D}}
\end{aligned}$$

$$\begin{aligned}
|\Gamma_4| &= \left| -\frac{1}{N}\sum_{l\neq k}\sum_{i\in\mathcal{I}_l} p(p-1)|c_{lj}|^{p-1}c_{kj}\,|\langle \varepsilon_i, \mu_k\rangle| \right| \\
&\leq \frac{1}{N}\sum_{l\neq k}\sum_{i\in\mathcal{I}_k} p^2\,|\langle \mu_k, \varepsilon_i\rangle| \leq Kp^2C\sqrt{\log\frac{K^2N}{\delta}}\frac{\alpha}{\sqrt{D}}.
\end{aligned}$$

With these norm bounds, we have

$$\begin{aligned}
\frac{d}{dt}c_{kj} &\\
&\geq pc_{kj}^{p-1}\left(1 - \max_{l\neq k}\frac{|c_{lj}|^{p-2}}{c_{kj}^{p-2}}\right)(1-c_{kj}^2) - C_1'\sqrt{\log\frac{K}{\delta}}\frac{\alpha}{\sqrt{N}} - C_2'\log\frac{K}{\delta}\alpha^2 \\
&\quad - C_3'\log\frac{K^2N}{\delta}\alpha^3 - C_4'\max_i|f^{(p)}(x_i;\theta)| - C_5'\sqrt{\log\frac{K^2N}{\delta}}\frac{\alpha}{\sqrt{D}}
\end{aligned}$$

$$\geq pc_{kj}^{p-1}\left(1 - \max_{l \neq k}\frac{|c_{lj}|^{p-2}}{c_{kj}^{p-2}}\right)(1 - c_{kj}) - C_1 \log \frac{K}{\delta}\alpha^2 - C_2 \max_i |f^{(p)}(\boldsymbol{x}_i; \boldsymbol{\theta})|.$$

Lastly, our bound is

1. When we only assumed $c_{kj} \geq \tilde{\Delta}_1$:

$$\frac{d}{dt}c_{kj} \geq pc_{kj}^{p-1}\left(1 - \max_{l \neq k}\frac{|c_{lj}|^{p-2}}{c_{kj}^{p-2}}\right)(1 - c_{kj}) - C_1 \log \frac{K}{\delta}\alpha^2 - C_2 \max_i |f^{(p)}(\boldsymbol{x}_i; \boldsymbol{\theta})|$$

$$\geq pc_{kj}^{p-1}\tilde{\Delta}_2(1 - c_{kj}) - C_1 \log \frac{K}{\delta}\alpha^2 - C_2 \max_i |f^{(p)}(\boldsymbol{x}_i; \boldsymbol{\theta})|$$

2. When we further assume $c_{kj} \geq \sqrt{\frac{4}{5}}$, we have that $\sum_{l \neq k} c_{lj}^2 = 1 - c_{kj}^2 \leq \frac{1}{5}$, then $\max_{l \neq k}|c_{lj}| \leq \sqrt{\frac{1}{5}}$. Therefore

$$\left(1 - \max_{l \neq k}\frac{|c_{lj}|^{p-2}}{c_{kj}^{p-2}}\right) = \left(1 - \left(\frac{\max_{l \neq k}|c_{lj}|}{c_{kj}}\right)^{p-2}\right) \geq 1 - \frac{1}{2^{p-2}}, \tag{H.18}$$

which leads to

$$\frac{d}{dt}c_{kj} \geq pc_{kj}^{p-1}\left(1 - \frac{1}{2^{p-2}}\right)(1 - c_{kj}) - C_1 \log \frac{K}{\delta}\alpha^2 - C_2 \max_i |f^{(p)}(\boldsymbol{x}_i; \boldsymbol{\theta})|.$$

$\square$

**Lemma 7** (Restated). *Let $p > 2$. Condition on good event $\mathcal{E}_{good}$. Given an initialization shape that satisfies Assumption 2 with non-degeneracy gap $\Delta > 0$, define*

$$t_{1a} := \inf\left\{t : \max_i |f^{(p)}(\boldsymbol{x}_i; \boldsymbol{\theta}(t))| > \min\left\{\frac{\tilde{\Delta}_1^{p-1}\tilde{\Delta}_2(1 - \tilde{\Delta}_1)}{2^{p+1}}, \frac{\tilde{\Delta}_1^{p-1}\tilde{\Delta}_2(1 - \sqrt{2\Delta})}{2K2^{p+1}}\right\}\right\}. \tag{H.19}$$

*Then the following holds $\forall t \leq t_{1a}$:*

$$c_{kj}(t) \geq c_{kj}(0) \geq \tilde{\Delta}_1, \forall 1 \leq k \leq K, j \in \mathcal{N}_k, \tag{H.20}$$

*and*

$$\frac{|c_{lj}^{p-2}(t)|}{c_{kj}^{p-2}(t)} \leq \frac{|c_{lj}^{p-2}(0)|}{c_{kj}^{p-2}(0)} \leq 1 - \tilde{\Delta}_2 . \text{ and } \forall l \neq k, j \in \mathcal{N}_k . \tag{H.21}$$

*Proof.* When $1 \leq k \leq K_1$, $j \in \mathcal{N}_k$ implies that $j \in \mathcal{N}_+$ thus $\text{sign}(v_j) = 1$. We shall primarily focus on this case as the proof is nearly identical for $K_1 + 1 \leq k \leq K$.

**Overview of the proof**: We will prove by contradiction, we let $\tau_1 := \inf\{t : \exists k, j \in \mathcal{N}_k, \text{ s.t. } c_{kj}(t) < c_{kj}(0)\}$ and $\tau_2 := \inf\left\{t : \exists k, j \in \mathcal{N}_k, \& l \neq k, \text{ s.t. } \frac{|c_{lj}^{p-2}(t)|}{c_{kj}(t)} > \frac{|c_{lj}^{p-2}(0)|}{c_{kj}(0)}\right\}$, by the continuity of every $c_{kj}(t)$ and every $\frac{|c_{lj}^{p-2}(t)|}{c_{kj}(t)}$ on the interval $[0, \tau_1]$ and $[0, \tau_2]$ respectively, we know that $c_{kj}(\tau_1) = c_{kj}(0)$ for some $k, j$ and $\frac{|c_{lj}^{p-2}(\tau_2)|}{c_{kj}(\tau_2)} = \frac{|c_{lj}^{p-2}(0)|}{c_{kj}(0)}$ for some $k, j, l$. If $\min\{\tau_1, \tau_2\} > t_{1a}$ then there is nothing to be proved, otherwise, there are two cases:

1. When $\tau_1 = \min\{\tau_1, \tau_2\} \leq t_{1a}$, we show that for the $k, j$ such that $c_{kj}(\tau_1) = c_{kj}(0)$

$$\frac{d}{dt}c_{kj}\bigg|_{t=\tau_1} \geq 0, \tag{H.22}$$

which says $c_{kj}(\tau_1 + \Delta t) \geq c_{kj}(0)$ for every sufficiently small $\Delta t$, contradicting the definition of $\tau_1$.

2. When $\tau_2 = \min\{\tau_1, \tau_2\} \leq t_{1a}$, we show that for the $k, j, l$ such that $\frac{|c_{lj}^{p-2}(\tau_2)|}{c_{kj}(\tau_2)} = \frac{|c_{lj}^{p-2}(0)|}{c_{kj}(0)}$

$$\frac{d}{dt} \log \frac{|c_{lj}|}{c_{kj}}\bigg|_{t=\tau_2} \leq 0, \tag{H.23}$$

which says $\frac{|c_{lj}(\tau_1 + \Delta t)|}{c_{kj}(\tau_1 + \Delta t)} \leq c_{kj}(0)$ for every sufficiently small $\Delta t$ (due to the monotonicity of log function), contradicting the definition of $\tau_2$.

**Time derivatives of log cosine angles** We have shown in (E.6) that for every $1 \leq l \leq D$, whenever $|c_{lj}| > 0$,

$$\frac{d}{dt} \log |c_{lj}|$$

$$= -\frac{1}{N} \sum_{i:\langle \boldsymbol{x}_i, \boldsymbol{w}_j \rangle > 0} \nabla_{\hat{y}} \ell_i \, p \left( \left\langle \boldsymbol{x}_i, \frac{\boldsymbol{w}_j}{\|\boldsymbol{w}_j\|} \right\rangle \right)^{p-1} \left( \frac{\langle \boldsymbol{\mu}_l, \boldsymbol{x}_i \rangle}{c_{lj}} - \left\langle \boldsymbol{x}_i, \frac{\boldsymbol{w}_j}{\|\boldsymbol{w}_j\|} \right\rangle \right)$$

$$= -\frac{1}{N} \sum_{i:\langle \boldsymbol{x}_i, \boldsymbol{w}_j \rangle > 0} \nabla_{\hat{y}} \ell_i \, p \left( \left\langle \boldsymbol{x}_i, \frac{\boldsymbol{w}_j}{\|\boldsymbol{w}_j\|} \right\rangle \right)^{p-1} \frac{\langle \boldsymbol{\mu}_l, \boldsymbol{x}_i \rangle}{c_{lj}}$$

$$+ \frac{1}{N} \sum_{i:\langle \boldsymbol{x}_i, \boldsymbol{w}_j \rangle > 0} \nabla_{\hat{y}} \ell_i \, p \left( \left\langle \boldsymbol{x}_i, \frac{\boldsymbol{w}_j}{\|\boldsymbol{w}_j\|} \right\rangle \right)^{p} .$$

**Case One**: $\tau_1 = \min\{\tau_1, \tau_2\}$. This case is relatively easier as we have already shown Lemma 6. For the $k, j$ such that $c_{kj} = \tilde{\Delta}$

$$\frac{d}{dt} c_{kj}\bigg|_{t=\tau_1} \geq p c_{kj}^{p-1} \tilde{\Delta}_2 (1 - c_{kj}) - C_1 \log \frac{K}{\delta} \alpha^2 - \underbrace{p 2^{p+1} \max_i |f^{(p)}(\boldsymbol{x}_i; \boldsymbol{\theta})|}_{(*)},$$

by Lemma 6 (conditions are satisified at $t = \tau_1$ and one should be able to get $(*)$ using the intermediate results in the proof of Lemma 6). Then

$$\frac{d}{dt} c_{kj}\bigg|_{t=\tau_1} \geq p \tilde{\Delta}_1^{p-1} \tilde{\Delta}_2 (1 - \tilde{\Delta}_1) - C_1 \log \frac{K}{\delta} \alpha^2 - p 2^{p+1} \max_i |f^{(p)}(\boldsymbol{x}_i; \boldsymbol{\theta})|,$$

$$\overset{(\tau_1 \leq t_{1a})}{\geq} p \tilde{\Delta}_1^{p-1} \tilde{\Delta}_2 (1 - \tilde{\Delta}_1) - C_1 \log \frac{K}{\delta} \alpha^2 - \frac{1}{2} p \tilde{\Delta}_1^{p-1} \tilde{\Delta}_2 (1 - \tilde{\Delta}_1)$$

$$\geq \frac{1}{2} p \tilde{\Delta}_1^{p-1} \tilde{\Delta}_2 (1 - \tilde{\Delta}_1) - C_1 \log \frac{K}{\delta} \alpha^2 \geq 0,$$

for sufficiently small $\alpha$.

**Case Two**: $\tau_2 = \min\{\tau_1, \tau_2\}$. For the $k, j, l$ such that $\frac{|c_{lj}^{p-2}(\tau_2)|}{c_{kj}(\tau_2)} = \frac{|c_{lj}^{p-2}(0)|}{c_{kj}(0)}$, we have (although we omit the notation, **all the derivations are at** $\tau_2$, so that $c_{lj}$ can appear in the denominator of a fraction.)

$$\frac{d}{dt} \log \frac{|c_{lj}|}{c_{kj}}$$

$$= \frac{d}{dt} \log |c_{lj}| - \frac{d}{dt} \log c_{kj}$$

$$= -\frac{1}{N} \sum_{i:\langle \boldsymbol{x}_i, \boldsymbol{w}_j \rangle > 0} \nabla_{\hat{y}} \ell_i \, p \left( \left\langle \boldsymbol{x}_i, \frac{\boldsymbol{w}_j}{\|\boldsymbol{w}_j\|} \right\rangle \right)^{p-1} \left( \frac{\langle \boldsymbol{\mu}_l, \boldsymbol{x}_i \rangle}{c_{lj}} - \frac{\langle \boldsymbol{\mu}_k, \boldsymbol{x}_i \rangle}{c_{kj}} \right)$$

$$= \frac{1}{N} \sum_{i:\langle \boldsymbol{x}_i, \boldsymbol{w}_j \rangle > 0} (y_i - f^{(p)}(\boldsymbol{x}_i; \boldsymbol{\theta})) \, p \left( \left\langle \boldsymbol{x}_i, \frac{\boldsymbol{w}_j}{\|\boldsymbol{w}_j\|} \right\rangle \right)^{p-1} \left( \frac{\langle \boldsymbol{\mu}_l, \boldsymbol{x}_i \rangle}{c_{lj}} - \frac{\langle \boldsymbol{\mu}_k, \boldsymbol{x}_i \rangle}{c_{kj}} \right)$$

$$= \frac{1}{N} \sum_{i:\langle \boldsymbol{x}_i, \boldsymbol{w}_j \rangle > 0} y_i \, p \left( \left\langle \boldsymbol{x}_i, \frac{\boldsymbol{w}_j}{\|\boldsymbol{w}_j\|} \right\rangle \right)^{p-1} \left( \frac{\langle \boldsymbol{\mu}_l, \boldsymbol{x}_i \rangle}{c_{lj}} - \frac{\langle \boldsymbol{\mu}_k, \boldsymbol{x}_i \rangle}{c_{kj}} \right)$$

$$\underbrace{- \frac{1}{N} \sum_{i:\langle \boldsymbol{x}_i, \boldsymbol{w}_j \rangle > 0} f^{(p)}(\boldsymbol{x}_i; \boldsymbol{\theta}) \, p \left( \left\langle \boldsymbol{x}_i, \frac{\boldsymbol{w}_j}{\|\boldsymbol{w}_j\|} \right\rangle \right)^{p-1} \left( \frac{\langle \boldsymbol{\mu}_l, \boldsymbol{x}_i \rangle}{c_{lj}} - \frac{\langle \boldsymbol{\mu}_k, \boldsymbol{x}_i \rangle}{c_{kj}} \right)}_{:=\Gamma_1}$$

$$= \frac{1}{N} \sum_{i \in \mathcal{I}_k : \langle \boldsymbol{x}_i, \boldsymbol{w}_j \rangle > 0} y_i \, p \left( \left\langle \boldsymbol{\mu}_k + \boldsymbol{\varepsilon}_i, \frac{\boldsymbol{w}_j}{\|\boldsymbol{w}_j\|} \right\rangle \right)^{p-1} \left( \frac{\langle \boldsymbol{\mu}_l, \boldsymbol{\mu}_k + \boldsymbol{\varepsilon}_i \rangle}{c_{lj}} - \frac{\langle \boldsymbol{\mu}_k, \boldsymbol{\mu}_k + \boldsymbol{\varepsilon}_i \rangle}{c_{kj}} \right)$$

$$\frac{1}{N} \sum_{i \in \mathcal{I}_l : \langle \boldsymbol{x}_i, \boldsymbol{w}_j \rangle > 0} y_i \, p \left( \left\langle \boldsymbol{\mu}_l + \boldsymbol{\varepsilon}_i, \frac{\boldsymbol{w}_j}{\|\boldsymbol{w}_j\|} \right\rangle \right)^{p-1} \left( \frac{\langle \boldsymbol{\mu}_l, \boldsymbol{\mu}_l + \boldsymbol{\varepsilon}_i \rangle}{c_{lj}} - \frac{\langle \boldsymbol{\mu}_k, \boldsymbol{\mu}_l + \boldsymbol{\varepsilon}_i \rangle}{c_{kj}} \right)$$

$$\frac{1}{N} \sum_{\substack{1 \leq l' \leq K \\ l' \neq l, l' \neq k}} \sum_{i \in \mathcal{I}_{l'} : \langle \boldsymbol{x}_i, \boldsymbol{w}_j \rangle > 0} y_i \, p \left( \left\langle \boldsymbol{\mu}_{l'} + \boldsymbol{\varepsilon}_i, \frac{\boldsymbol{w}_j}{\|\boldsymbol{w}_j\|} \right\rangle \right)^{p-1} \left( \frac{\langle \boldsymbol{\mu}_l, \boldsymbol{\mu}_{l'} + \boldsymbol{\varepsilon}_i \rangle}{c_{lj}} - \frac{\langle \boldsymbol{\mu}_k, \boldsymbol{\mu}_{l'} + \boldsymbol{\varepsilon}_i \rangle}{c_{kj}} \right) + \Gamma_1$$

$$= \frac{1}{N} \sum_{i \in \mathcal{I}_k : \langle \boldsymbol{x}_i, \boldsymbol{w}_j \rangle > 0} y_i \, p \left( c_{kj} + \left\langle \boldsymbol{\varepsilon}_i, \frac{\boldsymbol{w}_j}{\|\boldsymbol{w}_j\|} \right\rangle \right)^{p-1} \left( -\frac{1}{c_{kj}} + \frac{\langle \boldsymbol{\mu}_l, \boldsymbol{\varepsilon}_i \rangle}{c_{lj}} - \frac{\langle \boldsymbol{\mu}_k, \boldsymbol{\varepsilon}_i \rangle}{c_{kj}} \right)$$

$$\frac{1}{N} \sum_{i \in \mathcal{I}_l : \langle \boldsymbol{x}_i, \boldsymbol{w}_j \rangle > 0} y_i \, p \left( c_{lj} + \left\langle \boldsymbol{\varepsilon}_i, \frac{\boldsymbol{w}_j}{\|\boldsymbol{w}_j\|} \right\rangle \right)^{p-1} \left( \frac{1}{c_{lj}} + \frac{\langle \boldsymbol{\mu}_l, \boldsymbol{\varepsilon}_i \rangle}{c_{lj}} - \frac{\langle \boldsymbol{\mu}_k, \boldsymbol{\varepsilon}_i \rangle}{c_{kj}} \right)$$

$$\frac{1}{N} \sum_{\substack{1 \leq l' \leq K \\ l' \neq l, l' \neq k}} \sum_{i \in \mathcal{I}_{l'} : \langle \boldsymbol{x}_i, \boldsymbol{w}_j \rangle > 0} y_i \, p \left( c_{l'j} + \left\langle \boldsymbol{\varepsilon}_i, \frac{\boldsymbol{w}_j}{\|\boldsymbol{w}_j\|} \right\rangle \right)^{p-1} \left( \frac{\langle \boldsymbol{\mu}_l, \boldsymbol{\varepsilon}_i \rangle}{c_{lj}} - \frac{\langle \boldsymbol{\mu}_k, \boldsymbol{\varepsilon}_i \rangle}{c_{kj}} \right) + \Gamma_1$$

$$= \frac{1}{N} \sum_{i \in \mathcal{I}_k : \langle \boldsymbol{x}_i, \boldsymbol{w}_j \rangle > 0} y_i \, p \left( c_{kj} + \left\langle \boldsymbol{\varepsilon}_i, \frac{\boldsymbol{w}_j}{\|\boldsymbol{w}_j\|} \right\rangle \right)^{p-1} \left( -\frac{1}{c_{kj}} \right)$$

$$\frac{1}{N} \sum_{i \in \mathcal{I}_l : \langle \boldsymbol{x}_i, \boldsymbol{w}_j \rangle > 0} y_i \, p \left( c_{lj} + \left\langle \boldsymbol{\varepsilon}_i, \frac{\boldsymbol{w}_j}{\|\boldsymbol{w}_j\|} \right\rangle \right)^{p-1} \left( \frac{1}{c_{lj}} \right) + \Gamma_1 + \Gamma_2 , \tag{H.24}$$

We view $\Gamma_1, \Gamma_2$ as "perturbation term" and will control their norms later. For the first two terms in (H.24), we have, respectively:

$$\frac{1}{N} \sum_{i \in \mathcal{I}_k : \langle \boldsymbol{x}_i, \boldsymbol{w}_j \rangle > 0} y_i \, p \left( c_{kj} + \left\langle \boldsymbol{\varepsilon}_i, \frac{\boldsymbol{w}_j}{\|\boldsymbol{w}_j\|} \right\rangle \right)^{p-1} \left( -\frac{1}{c_{kj}} \right)$$

$$= -\frac{p}{N} \sum_{i \in \mathcal{I}_k} \left( c_{kj} + \left\langle \boldsymbol{\varepsilon}_i, \frac{\boldsymbol{w}_j}{\|\boldsymbol{w}_j\|} \right\rangle \right)^{p-1} \frac{1}{c_{kj}}$$

$$= -\frac{p}{N} \sum_{i \in \mathcal{I}_k} c_{kj}^{p-2} \left( 1 - \frac{\left| \left\langle \boldsymbol{\varepsilon}_i, \frac{\boldsymbol{w}_j}{\|\boldsymbol{w}_j\|} \right\rangle \right|}{c_{kj}} \right)^{p-1}$$

$$\leq -\frac{p}{N} \sum_{i \in \mathcal{I}_k} c_{kj}^{p-2} \left( 1 - \frac{\left| \left\langle \boldsymbol{\varepsilon}_i, \frac{\boldsymbol{w}_j}{\|\boldsymbol{w}_j\|} \right\rangle \right|}{c_{kj}} \right)^{p-1}$$

$$\leq -\frac{p}{N} \sum_{i \in \mathcal{I}_k} c_{kj}^{p-2} \left( 1 - (p-1) \frac{\left| \left\langle \boldsymbol{\varepsilon}_i, \frac{\boldsymbol{w}_j}{\|\boldsymbol{w}_j\|} \right\rangle \right|}{c_{kj}} \right)$$

$$\leq -p c_{kj}^{p-2} + p(p-1) \frac{\max_i \|\boldsymbol{\varepsilon}_i\|}{c_{kj}(0)} \geq -p c_{kj}^{p-2} + p(p-1) \sqrt{8 \log \frac{4K^2 N}{\delta}} \alpha ,$$

and similarly,

$$\frac{1}{N} \sum_{i \in \mathcal{I}_l : \langle \boldsymbol{x}_i, \boldsymbol{w}_j \rangle > 0} y_i \, p \left( c_{lj} + \left\langle \boldsymbol{\varepsilon}_i, \frac{\boldsymbol{w}_j}{\|\boldsymbol{w}_j\|} \right\rangle \right)^{p-1} \left( \frac{1}{c_{lj}} \right)$$

$$\leq \frac{p}{N} \sum_{i \in \mathcal{I}_l : \langle \boldsymbol{x}_i, \boldsymbol{w}_j \rangle > 0} |y_i| \, |c_{lj}|^{p-2} \left( 1 + \frac{\left| \left\langle \boldsymbol{\varepsilon}_i, \frac{\boldsymbol{w}_j}{\|\boldsymbol{w}_j\|} \right\rangle \right|}{c_{lj}} \right)^{p-1}$$

$$\leq \frac{p}{N} \sum_{i \in \mathcal{I}_l : \langle \boldsymbol{x}_i, \boldsymbol{w}_j \rangle > 0} |c_{lj}|^{p-2} \left( 1 + (p-1) \frac{\left| \left\langle \boldsymbol{\varepsilon}_i, \frac{\boldsymbol{w}_j}{\|\boldsymbol{w}_j\|} \right\rangle \right|}{c_{lj}} \right)$$

$$\begin{cases} \leq p|c_{lj}|^{p-2} + p(p-1)\sqrt{8 \log \frac{4K^2 N}{\delta}} \alpha, & 1 \leq l \leq K \\ = 0, & K < l \leq D \end{cases}$$

Therefore we have

$$\frac{d}{dt} \log \frac{|c_{lj}|}{c_{kj}} \leq -p(c_{kj}^{p-2} - |c_{lj}|^{p-2} \mathbb{1}_{l \leq K}) + 2p(p-1)\sqrt{8 \log \frac{4K^2 N}{\delta}} \alpha - |\Gamma_1| - |\Gamma_2|$$

$$\leq -p(c_{kj}^{p-2} - |c_{lj}|^{p-2}) + 2p(p-1)\sqrt{8 \log \frac{4K^2 N}{\delta}} \alpha - |\Gamma_1| - |\Gamma_2|$$

$$\leq -pc_{kj}^{p-2}\left( 1 - \frac{|c_{lj}|^{p-2}}{c_{kj}^{p-2}} \right) + 2p(p-1)\sqrt{8 \log \frac{4K^2 N}{\delta}} \alpha - |\Gamma_1| - |\Gamma_2|$$

$$\leq -p\tilde{\Delta}_1^{p-2} \tilde{\Delta}_2 + 2p(p-1)\sqrt{8 \log \frac{4K^2 N}{\delta}} \alpha - |\Gamma_1| - |\Gamma_2| \, .$$

It remains to bound these $|\Gamma_1|, |\Gamma_2|$. Indeed, we can find the following bound[6] (note that at $\tau_2$, we have $|c_{lj}| = c_{kj}(1 - \sqrt{2\Delta})$) and $c_{kj} \geq \tilde{\Delta}_1$):

$$|\Gamma_1| = \left| \frac{1}{N} \sum_{i : \langle \boldsymbol{x}_i, \boldsymbol{w}_j \rangle > 0} f^{(p)}(\boldsymbol{x}_i; \boldsymbol{\theta}) \, p \left( \left\langle \boldsymbol{x}_i, \frac{\boldsymbol{w}_j}{\|\boldsymbol{w}_j\|} \right\rangle \right)^{p-1} \left( \frac{\langle \boldsymbol{\mu}_l, \boldsymbol{x}_i \rangle}{c_{lj}} - \frac{\langle \boldsymbol{\mu}_k, \boldsymbol{x}_i \rangle}{c_{kj}} \right) \right|$$

$$\leq \left| \frac{1}{N} \sum_{1 \leq i \leq KN} |f^{(p)}(\boldsymbol{x}_i; \boldsymbol{\theta})| \, p \left| \left\langle \boldsymbol{x}_i, \frac{\boldsymbol{w}_j}{\|\boldsymbol{w}_j\|} \right\rangle \right|^{p-1} \left( \frac{|\langle \boldsymbol{\mu}_l, \boldsymbol{x}_i \rangle|}{c_{kj}(1 - \sqrt{2\Delta})} + \frac{|\langle \boldsymbol{\mu}_k, \boldsymbol{x}_i \rangle|}{c_{kj}} \right) \right|$$

$$\leq \max_i |f^{(p)}(\boldsymbol{x}_i; \boldsymbol{\theta})| \frac{Kp2^{p+1}}{\tilde{\Delta}_1(1 - \sqrt{2\Delta})} \overset{(\tau_2 \leq t_{1a})}{\leq} \frac{1}{2} p\tilde{\Delta}_1^{p-2} \tilde{\Delta}_2 \, ,$$

$$|\Gamma_2| = \left| \frac{1}{N} \sum_{i : \langle \boldsymbol{x}_i, \boldsymbol{w}_j \rangle > 0} y_i \, p \left( \left\langle \boldsymbol{x}_i, \frac{\boldsymbol{w}_j}{\|\boldsymbol{w}_j\|} \right\rangle \right)^{p-1} \left( \frac{\langle \boldsymbol{\mu}_l, \boldsymbol{\varepsilon}_i \rangle}{c_{lj}} - \frac{\langle \boldsymbol{\mu}_k, \boldsymbol{\varepsilon}_i \rangle}{c_{kj}} \right) \right|$$

$$\leq \left| \frac{1}{N} \sum_{1 \leq i \leq KN} p \left| \left\langle \boldsymbol{x}_i, \frac{\boldsymbol{w}_j}{\|\boldsymbol{w}_j\|} \right\rangle \right|^{p-1} \left( \frac{|\langle \boldsymbol{\mu}_l, \boldsymbol{\varepsilon}_i \rangle|}{c_{kj}(1 - \sqrt{2\Delta})} + \frac{|\langle \boldsymbol{\mu}_k, \boldsymbol{\varepsilon}_i \rangle|}{c_{kj}} \right) \right|$$

$$\leq \max_{i,k} |\langle \boldsymbol{\mu}_k, \boldsymbol{\varepsilon}_i \rangle| \frac{Kp2^{p+1}}{\tilde{\Delta}_1(1 - \sqrt{2\Delta})} \leq \frac{CKp2^{p+1}}{\tilde{\Delta}_1(1 - \sqrt{2\Delta})} \sqrt{\log \frac{K^2 N}{\delta}} \frac{\alpha}{\sqrt{D}}$$

Finally, we arrived at

$$\frac{d}{dt} \log \frac{|c_{lj}|}{c_{kj}} \bigg|_{t = \tau_2}$$

---

[6] It may take some time to recollect the terms we omitted in (H.24) and regroup them into $\Gamma_2$

$$\leq -p\tilde{\Delta}_1^{p-2}\tilde{\Delta}_2 + 2p(p-1)\sqrt{8\log\frac{4K^2N}{\delta}}\alpha + \frac{1}{2}p\tilde{\Delta}_1^{p-2}\tilde{\Delta}_2 + \frac{CKp2^{p+1}}{\tilde{\Delta}_1(1-\sqrt{2\Delta})}\sqrt{\log\frac{K^2N}{\delta}}\frac{\alpha}{\sqrt{D}}$$

$$\leq -\frac{1}{2}p\tilde{\Delta}_1^{p-2}\tilde{\Delta}_2 + 2p(p-1)\sqrt{8\log\frac{4K^2N}{\delta}}\alpha + \frac{CKp2^{p+1}}{\tilde{\Delta}_1(1-\sqrt{2\Delta})}\sqrt{\log\frac{K^2N}{\delta}}\frac{\alpha}{\sqrt{D}} \leq 0, \tag{H.25}$$

for sufficiently small $\alpha$. $\qquad\square$

**Lemma 8** (Restated). *Let $p > 2$. Condition on good event $\mathcal{E}_{good}$, then with any balanced initialization scale $\epsilon \leq \frac{1}{4\sqrt{h}W_{\max}^2}$, the solution to gradient flow dynamics satisfies*

$$\max_k |f^{(p)}(\boldsymbol{\mu}_k; \boldsymbol{\theta}(t))| \leq 2\epsilon\sqrt{h}W_{\max}^2, \quad \forall t \leq \frac{1}{2^{p+2}K}\log\left(\frac{1}{2^{p-1}\sqrt{h}\epsilon}\right). \tag{H.26}$$

*Proof.* Let $T := \inf\{t : \max_i |f(\boldsymbol{x}_k; \boldsymbol{\theta}(t))| > 2\epsilon\sqrt{h}W_{\max}^2\}$, then $\forall t \leq T, j \in [h]$, we have

$$\begin{aligned}
\frac{d}{dt}\|\boldsymbol{w}_j\|^2 &= -2\frac{\text{sign}(v_j(0))}{N}\left(\sum_{i:\langle\boldsymbol{x}_i,\boldsymbol{w}_j\rangle>0}\nabla_{\hat{y}}\ell_i\left(\left\langle\boldsymbol{x}_i,\frac{\boldsymbol{w}_j}{\|\boldsymbol{w}_j\|}\right\rangle\right)^p\right)\|\boldsymbol{w}_j\|^2 \\
&\leq 2\frac{1}{N}\sum_{i=1}^{KN}|\nabla_{\hat{y}}\ell_i|\|\boldsymbol{w}_j\|^2\frac{(\langle\boldsymbol{x}_i,\boldsymbol{w}_j\rangle)^p}{\|\boldsymbol{w}_j\|^p} \\
&\leq 2\frac{1}{N}\sum_{i=1}^{KN}|\nabla_{\hat{y}}\ell_i|\|\boldsymbol{w}_j\|^2\|\boldsymbol{x}_i\|^p \\
&\leq \frac{2^{p+1}}{N}\sum_{i=1}^{KN}(1+|f(\boldsymbol{x}_k;\boldsymbol{\theta}(t))|)\|\boldsymbol{w}_j\|^2 \\
&\leq \frac{2^{p+1}}{N}\sum_{i=1}^{KN}(1+4\epsilon\sqrt{h}W_{\max}^2)\|\boldsymbol{w}_j\|^2 \\
&\leq 2^{p+1}K(1+4\epsilon\sqrt{h}W_{\max}^2)\|\boldsymbol{w}_j\|^2. \tag{H.27}
\end{aligned}$$

Let $\tau_j := \inf\{t : \|\boldsymbol{w}_j(t)\|^2 > \frac{2\epsilon M^2}{2^{p-1}\sqrt{h}}\}$, and let $j^* := \arg\min_j \tau_j$, then $\tau_{j^*} = \min_j \tau_j \leq T$ due to the fact that

$$|f(\boldsymbol{x}_i;\boldsymbol{\theta})| = \left|\sum_{j\in[h]}\mathbb{1}_{\langle\boldsymbol{w}_j,\boldsymbol{x}_i\rangle>0}v_j\frac{(\langle\boldsymbol{w}_j,\boldsymbol{x}_k\rangle)^p}{\|\boldsymbol{w}_j\|^{p-1}}\right| \leq 2^p\sum_{j\in[h]}\|\boldsymbol{w}_j\|^2 \leq 2^p h\max_{j\in[h]}\|\boldsymbol{w}_j\|^2,$$

which implies "$|f(\boldsymbol{x}_k;\boldsymbol{\theta}(t))| > 2\epsilon\sqrt{h}W_{\max}^2 \Rightarrow \exists j, s.t. \|\boldsymbol{w}_j(t)\|^2 > \frac{\epsilon W_{\max}^2}{2^{p-1}\sqrt{h}}$".

Then for $t \leq \tau_{j^*}$, we have

$$\frac{d}{dt}\|\boldsymbol{w}_{j^*}\|^2 \leq 2^{p+1}K(+4\epsilon\sqrt{h}W_{\max}^2)\|\boldsymbol{w}_{j^*}\|^2. \tag{H.28}$$

By Gr$\acute{}$onwall's inequality, we have $\forall t \leq \tau_{j^*}$

$$\begin{aligned}
\|\boldsymbol{w}_{j^*}(t)\|^2 &\leq \exp\left(2^{p+1}K(1+4\epsilon\sqrt{h}W_{\max}^2)t\right)\|\boldsymbol{w}_{j^*}(0)\|^2, \\
&= \exp\left(2^{p+1}K(1+4\epsilon\sqrt{h}W_{\max}^2)t\right)\epsilon^2\|\boldsymbol{w}_{j^*0}\|^2 \\
&\leq \exp\left(2^{p+1}K(1+4\epsilon\sqrt{h}W_{\max}^2)t\right)\epsilon^2W_{\max}^2.
\end{aligned}$$

Suppose $\tau_{j^*} < \frac{1}{2^{p+2}K}\log\left(\frac{1}{2^{p-1}\sqrt{h}\epsilon}\right)$, then by the continuity of $\|w_{j^*}(t)\|^2$, we have

$$\frac{2\epsilon W_{\max}^2}{2^{p-1}\sqrt{h}} \leq \|\boldsymbol{w}_{j^*}(\tau_{j^*})\|^2 \leq \exp\left(2^{p+1}K(1+4\epsilon\sqrt{h}W_{\max}^2)\tau_{j^*}\right)\epsilon^2W_{\max}^2$$

$$\leq \exp\left(2^{p+1}K(1 + 4\epsilon\sqrt{h}W_{\max}^2)\frac{1}{2^{p+2}K}\log\left(\frac{1}{2^{p-1}\sqrt{h}\epsilon}\right)\right)\epsilon^2 W_{\max}^2$$

$$\leq \exp\left(\frac{1 + 4\epsilon\sqrt{h}W_{\max}^2}{2}\log\left(\frac{1}{2^{p-1}\sqrt{h}\epsilon}\right)\right)\epsilon^2 W_{\max}^2$$

$$\leq \exp\left(\log\left(\frac{1}{2^{p-1}\sqrt{h}\epsilon}\right)\right)\epsilon^2 W_{\max}^2 = \frac{\epsilon W_{\max}^2}{2^{p-1}\sqrt{h}},$$

which leads to a contradiction $2\epsilon \leq \epsilon$. Therefore, one must have $T \geq \tau_{j^*} \geq \frac{1}{2^{p+2}K}\log\left(\frac{1}{2^{p-1}\sqrt{h}\epsilon}\right)$. This finishes the proof. $\qquad\square$

**Lemma 9** (Restated). *Let $p > 2$. Given some $C > 0$, if for some $z(t)$, the following holds*

$$\frac{d}{dt}z \geq Cz^{p-1}, \forall t \in [0, T], \ z(0) = z_0, \ z(T) = z_1, \tag{H.29}$$

*for some $0 < z_0 \leq z_1 < 1$. Then the travel time $T$ for $z(t)$ to go from $z_0$ to $z_1$ satifies:*

$$T \leq \frac{1}{(p-2)Cz_0^{p-2}}. \tag{H.30}$$

*Proof.* We have

$$\int_{z_0}^{z_1} \frac{1}{Cz^{p-1}}dz \geq \int_0^T dt, \tag{H.31}$$

thus

$$T \leq \frac{1}{(p-2)C}\left(\frac{1}{z_0^{p-2}} - \frac{1}{z_1^{p-2}}\right) \leq \frac{1}{(p-2)Cz_0^{p-2}}. \tag{H.32}$$

$\qquad\square$

**Lemma 10** (Restated). *Let $p > 2$. Given some $C > 0$, if for some $z(t)$, the following holds*

$$\frac{d}{dt}z \geq C(1 - z), \forall t \in [0, T], \ z(0) = z_0, \ z(T) = z_1, \tag{H.33}$$

*for some $0 < z_0 \leq z_1 < 1$. Then the travel time $T$ for $z(t)$ to go from $z_0$ to $z_1$ satifies:*

$$T \leq \frac{1}{C}\log\frac{1}{1 - z_1}. \tag{H.34}$$

*Proof.* We have

$$\int_{z_0}^{z_1} \frac{1}{C(1-z)}dz \geq \int_0^T dt, \tag{H.35}$$

thus

$$T \leq \frac{1}{C}\left(\log\frac{1 - z_0}{1 - z_1}\right) \leq \frac{1}{C}\log\frac{1}{1 - z_1}. \tag{H.36}$$

$\qquad\square$

**Lemma 11** (Restated). *Let $p > 2$. Condition on good event $\mathcal{E}_{good}$. Suppose the following is true at some point on the GF trajectory:*

1. $c_{kj}(t) \geq 1 - 2C_a\log\frac{K}{\delta}\alpha^2, \ \forall k, j \in \mathcal{N}_k$;

2. $\sum_{j \in \mathcal{N}_k}\|\boldsymbol{w}_j\|^2 \leq 1 + C_w\log\frac{K}{\delta}\alpha^2, \ \forall k$;

3. $\sum_{j \in \mathcal{N}_c}\|\boldsymbol{w}_j\|^2 = \tilde{o}(\alpha^2)$.

*Then the following holds for every $1 \leq k \leq K$, $i \in \mathcal{I}_k$,*

$$f^{(p)}(\boldsymbol{x}_i; \boldsymbol{\theta}) \leq \sum_{j \in \mathcal{N}_k} \|\boldsymbol{w}_j\|^2 \left(1 + 2^{p+2}C\sqrt{\log \frac{K^2 N}{\delta}}\alpha^2\right) + 2KC\alpha^p;$$

$$f^{(p)}(\boldsymbol{x}_i; \boldsymbol{\theta}) \geq \sum_{j \in \mathcal{N}_k} \|\boldsymbol{w}_j\|^2 \left(1 - 4pC\sqrt{\log \frac{K^2 N}{\delta}}\alpha^2\right) - 2KC\alpha^p.$$

*Proof.* Our proof ignores terms related to neurons in $\mathcal{N}_c$ as they only introduce a $\tilde{o}(\alpha^2)$ perturbation.

$$f^{(p)}(\boldsymbol{x}_i; \boldsymbol{\theta})$$

$$= \sum_{j=1}^{h} v_j \frac{\sigma^p(\langle \boldsymbol{w}_j, \boldsymbol{x}_i \rangle)}{\|\boldsymbol{w}_j\|^{p-1}}$$

$$= \sum_{j=1}^{h} \|\boldsymbol{w}_j\|^2 \sigma^p \left(\left\langle \frac{\boldsymbol{w}_j}{\|\boldsymbol{w}_j\|}, \boldsymbol{x}_i \right\rangle\right)$$

$$= \sum_{j \in \mathcal{N}_k} \|\boldsymbol{w}_j\|^2 \left(\left\langle \frac{\boldsymbol{w}_j}{\|\boldsymbol{w}_j\|}, \boldsymbol{x}_i \right\rangle\right)^p + \sum_{l \neq k} \sum_{j \in \mathcal{N}_l} \|\boldsymbol{w}_j\|^2 \sigma^p \left(\left\langle \frac{\boldsymbol{w}_j}{\|\boldsymbol{w}_j\|}, \boldsymbol{x}_i \right\rangle\right)$$

$$= \sum_{j \in \mathcal{N}_k} \|\boldsymbol{w}_j\|^2 \left(c_{kj} + \left\langle \frac{\boldsymbol{w}_j}{\|\boldsymbol{w}_j\|}, \boldsymbol{\varepsilon}_i \right\rangle\right)^p + \sum_{l \neq k} \sum_{j \in \mathcal{N}_l} \|\boldsymbol{w}_j\|^2 \sigma^p \left(c_{lj} + \left\langle \frac{\boldsymbol{w}_j}{\|\boldsymbol{w}_j\|}, \boldsymbol{\varepsilon}_i \right\rangle\right) \qquad \text{(H.37)}$$

**Upper bound**:

$$f^{(p)}(\boldsymbol{x}_i; \boldsymbol{\theta})$$

$$= \text{(H.37)}$$

$$\leq \underbrace{\sum_{j \in \mathcal{N}_k} \|\boldsymbol{w}_j\|^2 \left(c_{kj} + \left|\left\langle \frac{\boldsymbol{w}_j}{\|\boldsymbol{w}_j\|}, \boldsymbol{\varepsilon}_i \right\rangle\right|\right)^p}_{(a)} + \underbrace{\left|\sum_{l \neq k} \sum_{j \in \mathcal{N}_l} \|\boldsymbol{w}_j\|^2 \sigma^p \left(c_{lj} + \left\langle \frac{\boldsymbol{w}_j}{\|\boldsymbol{w}_j\|}, \boldsymbol{\varepsilon}_i \right\rangle\right)\right|}_{(b)}.$$

For the first term, we have

$$(a) \leq \sum_{j \in \mathcal{N}_k} \|\boldsymbol{w}_j\|^2 \left(1 + \|\boldsymbol{\varepsilon}_i\|\sqrt{1 - c_{kj}^2} + |\langle \boldsymbol{\mu}_k, \boldsymbol{\varepsilon}_i \rangle|\right)^p$$

$$\leq \sum_{j \in \mathcal{N}_k} \|\boldsymbol{w}_j\|^2 \left(1 + \|\boldsymbol{\varepsilon}_i\|\sqrt{2(1 - c_{kj})} + |\langle \boldsymbol{\mu}_k, \boldsymbol{\varepsilon}_i \rangle|\right)^p$$

$$\leq \sum_{j \in \mathcal{N}_k} \|\boldsymbol{w}_j\|^2 \left(1 + 2C\sqrt{\log \frac{K^2 N}{\delta}}\alpha^2 + C\sqrt{\log \frac{K^2 N}{\delta}}\frac{\alpha}{\sqrt{D}}\right)^p$$

$$\leq \sum_{j \in \mathcal{N}_k} \|\boldsymbol{w}_j\|^2 \left(1 + 2^{p+2}C\sqrt{\log \frac{K^2 N}{\delta}}\alpha^2\right),$$

for sufficiently small $\alpha$. For the second term, we have

$$(b) \leq 2 \sum_{l \neq k} \left(|c_{lj}| + \left|\left\langle \boldsymbol{\varepsilon}_i, \frac{\boldsymbol{w}_j}{\|\boldsymbol{w}_j\|} \right\rangle\right|\right)^p$$

$$\leq 2K \left(\sqrt{1 - c_{kj}^2} + \|\boldsymbol{\varepsilon}_i\|\sqrt{1 - c_{kj}^2} + |\langle \boldsymbol{\mu}_k, \boldsymbol{\varepsilon}_i \rangle|\right)^p$$

$$\leq 2K \left(\sqrt{2(1 - c_{kj})} + \|\boldsymbol{\varepsilon}_i\|\sqrt{(1 - c_{kj})} + |\langle \boldsymbol{\mu}_k, \boldsymbol{\varepsilon}_i \rangle|\right)^p$$

$$\leq 2K \left( C\alpha + C\sqrt{\log \frac{K^2 N}{\delta}} \alpha^2 + C\sqrt{\log \frac{K^2 N}{\delta}} \frac{\alpha}{\sqrt{D}} \right)^p \leq 2KC\alpha^p . \tag{H.38}$$

Therefore

$$f^{(p)}(\boldsymbol{x}_i; \boldsymbol{\theta}) \leq \sum_{j \in \mathcal{N}_k} \|\boldsymbol{w}_j\|^2 \left( 1 + 2^{p+2} C\sqrt{\log \frac{K^2 N}{\delta}} \alpha^2 \right) + 2KC\alpha^p . \tag{H.39}$$

Lower bound:

$$f^{(p)}(\boldsymbol{x}_i; \boldsymbol{\theta})$$
$$= \text{(H.37)}$$
$$\geq \underbrace{\sum_{j \in \mathcal{N}_k} \|\boldsymbol{w}_j\|^2 \left( c_{kj} + \left| \left\langle \frac{\boldsymbol{w}_j}{\|\boldsymbol{w}_j\|}, \boldsymbol{\varepsilon}_i \right\rangle \right| \right)^p}_{(a)} - \underbrace{\left| \sum_{l \neq k} \sum_{j \in \mathcal{N}_l} \|\boldsymbol{w}_j\|^2 \sigma^p \left( c_{lj} + \left\langle \frac{\boldsymbol{w}_j}{\|\boldsymbol{w}_j\|}, \boldsymbol{\varepsilon}_i \right\rangle \right) \right|}_{\leq \text{(H.38)}} .$$

For the first term, we have

$$(a) \geq \sum_{j \in \mathcal{N}_k} \|\boldsymbol{w}_j\|^2 \left( 1 - \|\boldsymbol{\varepsilon}_i\| \sqrt{1 - c_{kj}^2} - |\langle \boldsymbol{\mu}_k, \boldsymbol{\varepsilon}_i \rangle| \right)^p$$
$$\geq \sum_{j \in \mathcal{N}_k} \|\boldsymbol{w}_j\|^2 \left( 1 - \|\boldsymbol{\varepsilon}_i\| \sqrt{2(1 - c_{kj})} - |\langle \boldsymbol{\mu}_k, \boldsymbol{\varepsilon}_i \rangle| \right)^p$$
$$\geq \sum_{j \in \mathcal{N}_k} \|\boldsymbol{w}_j\|^2 \left( 1 - 2C\sqrt{\log \frac{K^2 N}{\delta}} \alpha^2 - C\sqrt{\log \frac{K^2 N}{\delta}} \frac{\alpha}{\sqrt{D}} \right)^p$$
$$\geq \sum_{j \in \mathcal{N}_k} \|\boldsymbol{w}_j\|^2 \left( 1 - 4pC\sqrt{\log \frac{K^2 N}{\delta}} \alpha^2 \right) ,$$

for sufficiently small $\alpha$. Therefore

$$f^{(p)}(\boldsymbol{x}_i; \boldsymbol{\theta}) \geq \sum_{j \in \mathcal{N}_k} \|\boldsymbol{w}_j\|^2 \left( 1 - 4pC\sqrt{\log \frac{K^2 N}{\delta}} \alpha^2 \right) - 2KC\alpha^p . \tag{H.40}$$

$\square$

**Lemma 12** (Restated). *Let $p > 2$. Condition on good event $\mathcal{E}_{good}$. Suppose the following is true at some point on the GF trajectory:*

*1. $c_{kj}(t) \geq 1 - 2C_a \log \frac{K}{\delta} \alpha^2, \ \forall k, j \in \mathcal{N}_k$;*

*2. $\sum_{j \in \mathcal{N}_k} \|\boldsymbol{w}_j\|^2 \leq 1 + C_w \log \frac{K}{\delta} \alpha^2, \ \forall k$;*

*3. $\sum_{j \in \mathcal{N}_c} \|\boldsymbol{w}_j\|^2 = \tilde{o}(\alpha^2)$.*

*Furthermore, suppose additionally that for some $k, j \in \mathcal{N}_k$:*

$$1 - 2C_a \log \frac{K}{\delta} \alpha^2 \leq c_{kj}(t) \leq 1 - C_a \log \frac{K}{\delta} \alpha^2;$$

*Then the following holds for the same $k, j$,*

$$\frac{d}{dt} c_{kj} \geq -CK \log \frac{K^2 N}{\delta} \alpha^{\min\{p, 4\}} .$$

*Proof.* When $1 \leq k \leq K_1$, $j \in \mathcal{N}_k$ implies that $j \in \mathcal{N}_+$ thus $\text{sign}(v_j) = 1$. We shall primarily focus on this case as the proof is nearly identical for $K_1 + 1 \leq k \leq K$.

$$\frac{d}{dt} c_{kj}$$

$$= -\frac{1}{N} \sum_{i:\langle \boldsymbol{x}_i, \boldsymbol{w}_j \rangle > 0} \nabla_{\hat{y}} \ell_i \, p \left( \left\langle \boldsymbol{x}_i, \frac{\boldsymbol{w}_j}{\|\boldsymbol{w}_j\|} \right\rangle \right)^{p-1} \left( \langle \boldsymbol{\mu}_k, \boldsymbol{x}_i \rangle - \left\langle \boldsymbol{x}_i, \frac{\boldsymbol{w}_j}{\|\boldsymbol{w}_j\|} \right\rangle c_{kj} \right)$$

$$= \frac{1}{N} \sum_{i:\langle \boldsymbol{x}_i, \boldsymbol{w}_j \rangle > 0} (y_i - f^{(p)}(\boldsymbol{x}_i; \boldsymbol{\theta})) \, p \left( \left\langle \boldsymbol{x}_i, \frac{\boldsymbol{w}_j}{\|\boldsymbol{w}_j\|} \right\rangle \right)^{p-1} \left( \langle \boldsymbol{\mu}_k, \boldsymbol{x}_i \rangle - \left\langle \boldsymbol{x}_i, \frac{\boldsymbol{w}_j}{\|\boldsymbol{w}_j\|} \right\rangle c_{kj} \right)$$

$$= \underbrace{\frac{1}{N} \sum_{i \in \mathcal{I}_k} \left( 1 - \sum_{j \in \mathcal{N}_k} \|\boldsymbol{w}_j\|^2 \right) \, p \left( \left\langle \boldsymbol{x}_i, \frac{\boldsymbol{w}_j}{\|\boldsymbol{w}_j\|} \right\rangle \right)^{p-1} \left( \langle \boldsymbol{\mu}_k, \boldsymbol{x}_i \rangle - \left\langle \boldsymbol{x}_i, \frac{\boldsymbol{w}_j}{\|\boldsymbol{w}_j\|} \right\rangle c_{kj} \right)}_{(a)}$$

$$+ \underbrace{\frac{1}{N} \sum_{i \in \mathcal{I}_k} \left( \sum_{j \in \mathcal{N}_k} \|\boldsymbol{w}_j\|^2 - f^{(p)}(\boldsymbol{x}_i; \boldsymbol{\theta}) \right) \, p \left( \left\langle \boldsymbol{x}_i, \frac{\boldsymbol{w}_j}{\|\boldsymbol{w}_j\|} \right\rangle \right)^{p-1} \left( \langle \boldsymbol{\mu}_k, \boldsymbol{x}_i \rangle - \left\langle \boldsymbol{x}_i, \frac{\boldsymbol{w}_j}{\|\boldsymbol{w}_j\|} \right\rangle c_{kj} \right)}_{(b)}$$

$$+ \underbrace{\frac{1}{N} \sum_{l \neq k} \sum_{i \in \mathcal{I}_l : \langle \boldsymbol{x}_i, \boldsymbol{w}_j \rangle > 0} (y_i - f^{(p)}(\boldsymbol{x}_i; \boldsymbol{\theta})) \, p \left( \left\langle \boldsymbol{x}_i, \frac{\boldsymbol{w}_j}{\|\boldsymbol{w}_j\|} \right\rangle \right)^{p-1} \left( \langle \boldsymbol{\mu}_k, \boldsymbol{x}_i \rangle - \left\langle \boldsymbol{x}_i, \frac{\boldsymbol{w}_j}{\|\boldsymbol{w}_j\|} \right\rangle c_{kj} \right)}_{(c)} . \qquad \text{(H.41)}$$

We deal with these terms one by one:

Since $\sum_{j \in \mathcal{N}_k} \|\boldsymbol{w}_j\|^2 \leq 1 + C\alpha^2$, for $(a)$, there are two cases:

1. When $1 - \sum_{j \in \mathcal{N}_k} \|\boldsymbol{w}_j\|^2 \geq 0$, Follow the same derivations from (H.12) to (H.14), we have

$$(a) = \left( 1 - \sum_{j \in \mathcal{N}_k} \|\boldsymbol{w}_j\|^2 \right) \frac{1}{N} \sum_{i \in \mathcal{I}_k} p \left( \left\langle \boldsymbol{x}_i, \frac{\boldsymbol{w}_j}{\|\boldsymbol{w}_j\|} \right\rangle \right)^{p-1} \left( \langle \boldsymbol{\mu}_k, \boldsymbol{x}_i \rangle - \left\langle \boldsymbol{x}_i, \frac{\boldsymbol{w}_j}{\|\boldsymbol{w}_j\|} \right\rangle c_{kj} \right)$$

$$\geq \left( 1 - \sum_{j \in \mathcal{N}_k} \|\boldsymbol{w}_j\|^2 \right) \left( p c_{kj}^{p-1} (1 - c_{kj}^2) - C p^2 \sqrt{\log \frac{K}{\delta}} \frac{\alpha}{\sqrt{N}} - 2^{p-1} p^3 C^2 \log \frac{K}{\delta} \alpha^2 - o(\alpha^2) \right)$$

$$\geq \left( 1 - \sum_{j \in \mathcal{N}_k} \|\boldsymbol{w}_j\|^2 \right) \left( p(1 - C\alpha^2)^{p-1} \left( C\alpha^2 - \frac{C^2}{4}\alpha^4 \right) - C p^2 \sqrt{\log \frac{K}{\delta}} \frac{\alpha}{\sqrt{N}} - 2^{p-1} p^3 C^2 \log \frac{K}{\delta} \alpha^2 - o(\alpha^2) \right)$$

$$\geq 0 ,$$

for some choice of $C$ and sufficiently small $\alpha$.

2. When $-C\alpha^2 \leq 1 - \sum_{j \in \mathcal{N}_k} \|\boldsymbol{w}_j\|^2 \leq 0$, we have

$$(a) = \left( 1 - \sum_{j \in \mathcal{N}_k} \|\boldsymbol{w}_j\|^2 \right) \frac{1}{N} \sum_{i \in \mathcal{I}_k} p \left( \left\langle \boldsymbol{x}_i, \frac{\boldsymbol{w}_j}{\|\boldsymbol{w}_j\|} \right\rangle \right)^{p-1} \left( \langle \boldsymbol{\mu}_k, \boldsymbol{x}_i \rangle - \left\langle \boldsymbol{x}_i, \frac{\boldsymbol{w}_j}{\|\boldsymbol{w}_j\|} \right\rangle c_{kj} \right)$$

$$\geq - \left| 1 - \sum_{j \in \mathcal{N}_k} \|\boldsymbol{w}_j\|^2 \right| \frac{1}{N} \sum_{i \in \mathcal{I}_k} p \left( \left\langle \boldsymbol{x}_i, \frac{\boldsymbol{w}_j}{\|\boldsymbol{w}_j\|} \right\rangle \right)^{p-1} \left( 1 - c_{kj}^2 + |\langle \boldsymbol{\mu}_k, \boldsymbol{\varepsilon}_i \rangle| + \left| \left\langle \boldsymbol{\varepsilon}_i, \frac{\boldsymbol{w}_j}{\|\boldsymbol{w}_j\|} \right\rangle \right| c_{kj} \right)$$

$$\geq - \left| 1 - \sum_{j \in \mathcal{N}_k} \|\boldsymbol{w}_j\|^2 \right| p 2^{p-1} \left( 1 - c_{kj}^2 + 2|\langle \boldsymbol{\mu}_k, \boldsymbol{\varepsilon}_i \rangle| + \|\boldsymbol{\varepsilon}_i\| \sqrt{1 - c_{kj}^2} c_{kj} \right)$$

$$\geq C\sqrt{\log \frac{K^2 N}{\delta}}\alpha^4 \,, \tag{H.42}$$

Therefore, we always have

$$(a) \geq C\sqrt{\log \frac{K^2 N}{\delta}}\alpha^4 \,. \tag{H.43}$$

The second term $(b)$ is easy: by Lemma 11, we know that $\left|\sum_{j\in\mathcal{N}_k}\|\boldsymbol{w}_j\|^2 - f^{(p)}(\boldsymbol{x}_i;\boldsymbol{\theta})\right| = \mathcal{O}(\sqrt{\log \frac{K^2 N}{\delta}}\alpha^2)$, then by the a similar derivation as in (H.42), we have

$$
\begin{aligned}
(b) &= \frac{1}{N}\sum_{i\in\mathcal{I}_k}\left(\sum_{j\in\mathcal{N}_k}\|\boldsymbol{w}_j\|^2 - f^{(p)}(\boldsymbol{x}_i;\boldsymbol{\theta})\right) p\left(\left\langle \boldsymbol{x}_i, \frac{\boldsymbol{w}_j}{\|\boldsymbol{w}_j\|}\right\rangle\right)^{p-1}\left(\langle\boldsymbol{\mu}_k,\boldsymbol{x}_i\rangle - \left\langle \boldsymbol{x}_i, \frac{\boldsymbol{w}_j}{\|\boldsymbol{w}_j\|}\right\rangle c_{kj}\right) \\
&\geq -\left|\sum_{j\in\mathcal{N}_k}\|\boldsymbol{w}_j\|^2 - f^{(p)}(\boldsymbol{x}_i;\boldsymbol{\theta})\right|\frac{1}{N}\sum_{i\in\mathcal{I}_k} p\left(\left\langle \boldsymbol{x}_i, \frac{\boldsymbol{w}_j}{\|\boldsymbol{w}_j\|}\right\rangle\right)^{p-1}\left(1 - c_{kj}^2 + |\langle\boldsymbol{\mu}_k,\boldsymbol{\varepsilon}_i\rangle| + \left|\left\langle \boldsymbol{\varepsilon}_i, \frac{\boldsymbol{w}_j}{\|\boldsymbol{w}_j\|}\right\rangle\right| c_{kj}\right) \\
&\geq C\log \frac{K^2 N}{\delta}\alpha^4 \,,
\end{aligned} \tag{H.44}
$$

For the last term, we have

$$
\begin{aligned}
(c) &= \frac{1}{N}\sum_{l\neq k}\sum_{i\in\mathcal{I}_l:\langle\boldsymbol{x}_i,\boldsymbol{w}_j\rangle>0}(y_i - f^{(p)}(\boldsymbol{x}_i;\boldsymbol{\theta})) p\left(\left\langle \boldsymbol{x}_i, \frac{\boldsymbol{w}_j}{\|\boldsymbol{w}_j\|}\right\rangle\right)^{p-1}\left(\langle\boldsymbol{\mu}_k,\boldsymbol{x}_i\rangle - \left\langle \boldsymbol{x}_i, \frac{\boldsymbol{w}_j}{\|\boldsymbol{w}_j\|}\right\rangle c_{kj}\right) \\
&\geq -\frac{2}{N}\sum_{l\neq k}\sum_{i\in\mathcal{I}_l} p\left(c_{lj} + \left|\left\langle \boldsymbol{\varepsilon}_i, \frac{\boldsymbol{w}_j}{\|\boldsymbol{w}_j\|}\right\rangle\right|\right)^{p-1}\left(\langle\boldsymbol{\mu}_k,\boldsymbol{\varepsilon}_i\rangle + c_{lj}c_{kj} + \left|\left\langle \boldsymbol{\varepsilon}_i, \frac{\boldsymbol{w}_j}{\|\boldsymbol{w}_j\|}\right\rangle\right| c_{kj}\right) \\
&\geq -\frac{2}{N}\sum_{l\neq k}\sum_{i\in\mathcal{I}_l} p\left(\sqrt{1 - c_{kj}^2} + \|\boldsymbol{\varepsilon}_i\|\sqrt{1 - c_{kj}^2}\right)^{p-1}\left(\langle\boldsymbol{\mu}_k,\boldsymbol{\varepsilon}_i\rangle + \sqrt{1 - c_{kj}^2}c_{kj} + \|\boldsymbol{\varepsilon}_i\|\sqrt{1 - c_{kj}^2}c_{kj}\right) \\
&\geq -CK\sqrt{\log \frac{K^2 N}{\delta}}\alpha^p \,.
\end{aligned}
$$

Finally, we can conclude that

$$\frac{d}{dt}c_{kj} \geq -CK\log \frac{K^2 N}{\delta}\alpha^{\min\{p,4\}} \,. \tag{H.45}$$

$\square$

**Lemma 13** (Restated). *Let $p > 2$. Condition on good event $\mathcal{E}_{good}$. Suppose the following is true at some point on the GF trajectory :*

1. $c_{kj}(t) \geq 1 - 2C_a \log \frac{K}{\delta}\alpha^2, \ k,j \in \mathcal{N}_k;$

2. $\sum_{j\in\mathcal{N}_k}\|\boldsymbol{w}_j\|^2 \leq 1 + C_w \log \frac{K}{\delta}\alpha^2, \ \forall k;$

3. $\sum_{j\in\mathcal{N}_c}\|\boldsymbol{w}_j\|^2 = \tilde{o}(\alpha^2).$

*Then the following holds for every $1 \leq k \leq K$,*

$$\frac{d}{dt}\left(\sum_{j\in\mathcal{N}_k}\|\boldsymbol{w}_j\|^2\right) \leq 2\left(1 - \sum_{j\in\mathcal{N}_k}\|\boldsymbol{w}_j\|^2 + C\log \frac{K}{\delta}\alpha^2\right)\left(\sum_{j\in\mathcal{N}_k}\|\boldsymbol{w}_j\|^2\right),$$

*and*

$$\frac{d}{dt}\left(\sum_{j\in\mathcal{N}_k}\|\boldsymbol{w}_j\|^2\right) \geq 2\left(1 - \sum_{j\in\mathcal{N}_k}\|\boldsymbol{w}_j\|^2 - C\log \frac{K}{\delta}\alpha^2\right)\left(\sum_{j\in\mathcal{N}_k}\|\boldsymbol{w}_j\|^2\right),$$

*where $C$ is some universal constant such that $C < C_w$.*

*Proof.* When $1 \leq k \leq K_1$, $j \in \mathcal{N}_k$ implies that $j \in \mathcal{N}_+$ thus $\text{sign}(v_j) = 1$. We shall primarily focus on this case as the proof is nearly identical for $K_1 + 1 \leq k \leq K$. We start with (E.3):

$$\frac{d}{dt}\|\boldsymbol{w}_j\|^2 = \frac{2}{N}\left(\sum_{i:\langle\boldsymbol{x}_i,\boldsymbol{w}_j\rangle>0}(y_i - f^{(p)}(\boldsymbol{x}_i;\boldsymbol{\theta}))\left(\left\langle\boldsymbol{x}_i,\frac{\boldsymbol{w}_j}{\|\boldsymbol{w}_j\|}\right\rangle\right)^p\right)\|\boldsymbol{w}_j\|^2$$

$$= 2\left(\underbrace{\frac{1}{N}\sum_{i\in\mathcal{I}_k:\langle\boldsymbol{x}_i,\boldsymbol{w}_j\rangle>0}(y_i - f^{(p)}(\boldsymbol{x}_i;\boldsymbol{\theta}))\left(\left\langle\boldsymbol{x}_i,\frac{\boldsymbol{w}_j}{\|\boldsymbol{w}_j\|}\right\rangle\right)^p}_{(a)}\right.$$

$$\left.+\underbrace{\frac{1}{N}\sum_{l\neq k}\sum_{i\in\mathcal{I}_l:\langle\boldsymbol{x}_i,\boldsymbol{w}_j\rangle>0}(y_i - f^{(p)}(\boldsymbol{x}_i;\boldsymbol{\theta}))\left(\left\langle\boldsymbol{x}_i,\frac{\boldsymbol{w}_j}{\|\boldsymbol{w}_j\|}\right\rangle\right)^p}_{:=\Gamma_1}\|\boldsymbol{w}_j\|^2\right)$$

For the first term, we have

$$(a) = \frac{1}{N}\sum_{i\in\mathcal{I}_k:\langle\boldsymbol{x}_i,\boldsymbol{w}_j\rangle>0}(y_i - f^{(p)}(\boldsymbol{x}_i;\boldsymbol{\theta}))\left(\left\langle\boldsymbol{x}_i,\frac{\boldsymbol{w}_j}{\|\boldsymbol{w}_j\|}\right\rangle\right)^p$$

$$= \frac{1}{N}\sum_{i\in\mathcal{I}_k}(1 - f^{(p)}(\boldsymbol{x}_i;\boldsymbol{\theta}))\left(\left\langle\boldsymbol{x}_i,\frac{\boldsymbol{w}_j}{\|\boldsymbol{w}_j\|}\right\rangle\right)^p$$

$$= \frac{1}{N}\sum_{i\in\mathcal{I}_k}\left(1 - \sum_{j\in\mathcal{N}_k}\|\boldsymbol{w}_j\|^2\left(\left\langle\boldsymbol{x}_i,\frac{\boldsymbol{w}_j}{\|\boldsymbol{w}_j\|}\right\rangle\right)^p + \sum_{l\neq k}\sum_{j\in\mathcal{N}_l}\|\boldsymbol{w}_j\|^2\sigma^p\left(\left\langle\boldsymbol{x}_i,\frac{\boldsymbol{w}_j}{\|\boldsymbol{w}_j\|}\right\rangle\right)\right)\left(\left\langle\boldsymbol{x}_i,\frac{\boldsymbol{w}_j}{\|\boldsymbol{w}_j\|}\right\rangle\right)^p$$

$$= \frac{1}{N}\sum_{i\in\mathcal{I}_k}\left(1 - \sum_{j\in\mathcal{N}_k}\|\boldsymbol{w}_j\|^2\left(\left\langle\boldsymbol{x}_i,\frac{\boldsymbol{w}_j}{\|\boldsymbol{w}_j\|}\right\rangle\right)^p\right)\left(\left\langle\boldsymbol{x}_i,\frac{\boldsymbol{w}_j}{\|\boldsymbol{w}_j\|}\right\rangle\right)^p + \underbrace{\frac{1}{N}\sum_{i\in\mathcal{I}_k}\sum_{l\neq k}\sum_{j\in\mathcal{N}_l}\|\boldsymbol{w}_j\|^2\sigma^{2p}\left(\left\langle\boldsymbol{x}_i,\frac{\boldsymbol{w}_j}{\|\boldsymbol{w}_j\|}\right\rangle\right)}_{:=\Gamma_2}.$$

$$= \frac{1}{N}\sum_{i\in\mathcal{I}_k}\left(1 - \sum_{j\in\mathcal{N}_k}\|\boldsymbol{w}_j\|^2\left(c_{kj} + \left\langle\boldsymbol{\varepsilon}_i,\frac{\boldsymbol{w}_j}{\|\boldsymbol{w}_j\|}\right\rangle\right)^p\right)\left(c_{kj} + \left\langle\boldsymbol{\varepsilon}_i,\frac{\boldsymbol{w}_j}{\|\boldsymbol{w}_j\|}\right\rangle\right)^p + \Gamma_2.$$

We shall focus on the first term. With the Taylor expansion

$$\left(c_{kj} + \left\langle\boldsymbol{\varepsilon}_i,\frac{\boldsymbol{w}_j}{\|\boldsymbol{w}_j\|}\right\rangle\right)^p = c_{kj}^p + pc_{kj}^{p-1}\left\langle\boldsymbol{\varepsilon}_i,\frac{\boldsymbol{w}_j}{\|\boldsymbol{w}_j\|}\right\rangle + R_L\left|\left\langle\boldsymbol{\varepsilon}_i,\frac{\boldsymbol{w}_j}{\|\boldsymbol{w}_j\|}\right\rangle\right|^2, \tag{H.46}$$

where $R_L = \frac{p(p-1)(c_{kj}+\zeta_L)^{p-2}}{2}$ and $\zeta_L$ between 0 and $\left|\left\langle\boldsymbol{\varepsilon}_i,\frac{\boldsymbol{w}_j}{\|\boldsymbol{w}_j\|}\right\rangle\right|$ comes from the Lagrange residual. Clearly $|R_L| \leq 2^{p-2}p^2$. Then we have

$$\frac{1}{N}\sum_{i\in\mathcal{I}_k}\left(1 - \sum_{j\in\mathcal{N}_k}\|\boldsymbol{w}_j\|^2\left(c_{kj} + \left\langle\boldsymbol{\varepsilon}_i,\frac{\boldsymbol{w}_j}{\|\boldsymbol{w}_j\|}\right\rangle\right)^p\right)\left(c_{kj} + \left\langle\boldsymbol{\varepsilon}_i,\frac{\boldsymbol{w}_j}{\|\boldsymbol{w}_j\|}\right\rangle\right)^p$$

$$= c_{kj}^p - \sum_{j\in\mathcal{N}_k}\|\boldsymbol{w}_j\|^2c_{kj}^{2p}$$

$$\left(pc_{kj}^{p-1} - 2\sum_{j\in\mathcal{N}_k}\|\boldsymbol{w}_j\|^2c_{kj}^{2p-1}\right)\frac{1}{N}\sum_{i\in\mathcal{I}_k}\left\langle\boldsymbol{\varepsilon}_i,\frac{\boldsymbol{w}_j}{\|\boldsymbol{w}_j\|}\right\rangle$$

$$\left(R_L - p^2 c_{kj}^{2p-2} - 2c_{kj}^p R_L\right) \frac{1}{N} \sum_{i \in \mathcal{I}_k} \left|\left\langle \boldsymbol{\varepsilon}_i, \frac{\boldsymbol{w}_j}{\|\boldsymbol{w}_j\|}\right\rangle\right|^2 + o\left(\left|\left\langle \boldsymbol{\varepsilon}_i, \frac{\boldsymbol{w}_j}{\|\boldsymbol{w}_j\|}\right\rangle\right|^2\right)$$

Finally, we are ready to derive the upper and lower bound. For lower bound,

$$\frac{d}{dt}\|\boldsymbol{w}_j\|^2$$
$$= 2\left((a) + \Gamma_1\right)\|\boldsymbol{w}_j\|^2$$
$$\geq 2\left(\frac{1}{N}\sum_{i \in \mathcal{I}_k}\left(1 - \sum_{j \in \mathcal{N}_k}\|\boldsymbol{w}_j\|^2\left(c_{kj} + \left\langle \boldsymbol{\varepsilon}_i, \frac{\boldsymbol{w}_j}{\|\boldsymbol{w}_j\|}\right\rangle\right)^p\right)\left(c_{kj} + \left\langle \boldsymbol{\varepsilon}_i, \frac{\boldsymbol{w}_j}{\|\boldsymbol{w}_j\|}\right\rangle\right)^p - |\Gamma_1| - |\Gamma_2|\right)\|\boldsymbol{w}_j\|^2$$
$$\geq 2\left(c_{kj}^p - \sum_{j \in \mathcal{N}_k}\|\boldsymbol{w}_j\|^2 c_{kj}^{2p} - C_1\frac{1}{N}\left|\left\langle \sum_{i \in \mathcal{I}_k}\boldsymbol{\varepsilon}_i, \frac{\boldsymbol{w}_j}{\|\boldsymbol{w}_j\|}\right\rangle\right| - C_2\frac{1}{N}\sum_{i \in \mathcal{I}_k}\left|\left\langle \boldsymbol{\varepsilon}_i, \frac{\boldsymbol{w}_j}{\|\boldsymbol{w}_j\|}\right\rangle\right|^2\right.$$
$$\left. - o\left(\left|\left\langle \boldsymbol{\varepsilon}_i, \frac{\boldsymbol{w}_j}{\|\boldsymbol{w}_j\|}\right\rangle\right|^2\right) - |\Gamma_1| - |\Gamma_2|\right)\|\boldsymbol{w}_j\|^2$$
$$\geq 2\left(c_{kj}^p - \sum_{j \in \mathcal{N}_k}\|\boldsymbol{w}_j\|^2 c_{kj}^{2p} - C\sqrt{\log\frac{K}{\delta}}\frac{\alpha}{\sqrt{N}} - C^2\log\frac{K}{\delta}\alpha^2 - o(\alpha^2) - |\Gamma_1| - |\Gamma_2|\right)\|\boldsymbol{w}_j\|^2$$
$$\geq 2\left(\left(1 - \frac{C\alpha^2}{2}\right)^p - \sum_{j \in \mathcal{N}_k}\|\boldsymbol{w}_j\|^2 - C\sqrt{\log\frac{K}{\delta}}\frac{\alpha}{\sqrt{N}} - C^2\log\frac{K}{\delta}\alpha^2 - o(\alpha^2) - |\Gamma_1| - |\Gamma_2|\right)\|\boldsymbol{w}_j\|^2$$
$$\geq 2\left(1 - p\frac{C\alpha^2}{2} - \sum_{j \in \mathcal{N}_k}\|\boldsymbol{w}_j\|^2 - C\sqrt{\log\frac{K}{\delta}}\frac{\alpha}{\sqrt{N}} - C^2\log\frac{K}{\delta}\alpha^2 - o(\alpha^2) - |\Gamma_1| - |\Gamma_2|\right)\|\boldsymbol{w}_j\|^2$$

It remains to bound these $|\Gamma_1|, |\Gamma_2|$. Indeed, we can find the following bound:

$$|\Gamma_1| = \left|\frac{1}{N}\sum_{l \neq k}\sum_{i \in \mathcal{I}_l : \langle \boldsymbol{x}_i, \boldsymbol{w}_j\rangle > 0}(y_i - f^{(p)}(\boldsymbol{x}_i; \boldsymbol{\theta}))\left(\left\langle \boldsymbol{x}_i, \frac{\boldsymbol{w}_j}{\|\boldsymbol{w}_j\|}\right\rangle\right)^p\right|$$
$$\leq \left|\frac{1}{N}\sum_{l \neq k}\sum_{i \in \mathcal{I}_l}|y_i - f^{(p)}(\boldsymbol{x}_i; \boldsymbol{\theta}))|\left(\left\langle \boldsymbol{x}_i, \frac{\boldsymbol{w}_j}{\|\boldsymbol{w}_j\|}\right\rangle\right)^p\right|$$
$$\leq \left|\frac{2}{N}\sum_{l \neq k}\sum_{i \in \mathcal{I}_l}\left(\left\langle \boldsymbol{x}_i, \frac{\boldsymbol{w}_j}{\|\boldsymbol{w}_j\|}\right\rangle\right)^p\right|$$
$$\leq \left|\frac{2}{N}\sum_{i \in \mathcal{I}_l}\sum_{l \neq k}\left(\left\langle \boldsymbol{x}_i, \frac{\boldsymbol{w}_j}{\|\boldsymbol{w}_j\|}\right\rangle\right)^p\right|$$
$$\leq \left|\frac{2}{N}\sum_{i \in \mathcal{I}_l}\sum_{l \neq k}\left(c_{lj} + \left\langle \boldsymbol{\varepsilon}_i, \frac{\boldsymbol{w}_j}{\|\boldsymbol{w}_j\|}\right\rangle\right)^p\right|$$
$$\leq \left|\frac{2}{N}\sum_{i \in \mathcal{I}_l}\sum_{l \neq k}\left(\sqrt{1 - c_{kj}^2} + \|\boldsymbol{\varepsilon}_i\|\sqrt{1 - c_{kj}^2} + \langle \boldsymbol{\mu}_k, \boldsymbol{\varepsilon}_i\rangle\right)^p\right| \leq KC\alpha^p,$$
$$|\Gamma_2| = \left|\frac{1}{N}\sum_{i \in \mathcal{I}_k}\sum_{l \neq k}\sum_{j \in \mathcal{N}_l}\|\boldsymbol{w}_j\|^2 \sigma^{2p}\left(\left\langle \boldsymbol{x}_i, \frac{\boldsymbol{w}_j}{\|\boldsymbol{w}_j\|}\right\rangle\right)\right|$$

$$\leq \left| \frac{2}{N} \sum_{i\in\mathcal{I}_k} \sum_{l\neq k} \left( \left\langle \boldsymbol{x}_i, \frac{\boldsymbol{w}_j}{\|\boldsymbol{w}_j\|} \right\rangle \right)^{2p} \right| \leq KC\alpha^{2p} .$$

Therefore,

$$\frac{d}{dt}\|\boldsymbol{w}_j\|^2 \geq 2 \left( 1 - \sum_{j\in\mathcal{N}_k} \|\boldsymbol{w}_j\|^2 - C\log\frac{K}{\delta}\alpha^2 \right) \|\boldsymbol{w}_j\|^2 ,$$

since when $\alpha$ is sufficiently small, the dominant term is of order $\alpha^2$.

Similarly, for the upper bound, we can have

$$\frac{d}{dt}\|\boldsymbol{w}_j\|^2$$
$$= 2\left((a) + \Gamma_1\right)\|\boldsymbol{w}_j\|^2$$
$$\leq 2\left( c_{kj}^p - \sum_{j\in\mathcal{N}_k} \|\boldsymbol{w}_j\|^2 c_{kj}^{2p} + C\sqrt{\log\frac{K}{\delta}}\frac{\alpha}{\sqrt{N}} + C^2\log\frac{K}{\delta}\alpha^2 + o(\alpha^2) + |\Gamma_1| + |\Gamma_2| \right)\|\boldsymbol{w}_j\|^2$$
$$\leq 2\left( 1 - \sum_{j\in\mathcal{N}_k} \|\boldsymbol{w}_j\|^2 \left(1 - \frac{C\alpha^2}{2}\right)^{2p} - C\sqrt{\log\frac{K}{\delta}}\frac{\alpha}{\sqrt{N}} + C^2\log\frac{K}{\delta}\alpha^2 + o(\alpha^2) + |\Gamma_1| + |\Gamma_2| \right)\|\boldsymbol{w}_j\|^2$$
$$\leq 2\left( 1 - \sum_{j\in\mathcal{N}_k} \|\boldsymbol{w}_j\|^2 + 4p\frac{C\alpha^2}{2} + C\sqrt{\log\frac{K}{\delta}}\frac{\alpha}{\sqrt{N}} + C^2\log\frac{K}{\delta}\alpha^2 + o(\alpha^2) + |\Gamma_1| + |\Gamma_2| \right)\|\boldsymbol{w}_j\|^2$$
$$\leq 2\left( 1 - \sum_{j\in\mathcal{N}_k} \|\boldsymbol{w}_j\|^2 + C\log\frac{K}{\delta}\alpha^2 \right)\|\boldsymbol{w}_j\|^2$$

$\square$

**Lemma 14** (Restated). *Consider the same assumptions as in Proposition 4. Given the $t_1$ in Proposition 4, the following holds $\forall 1 \leq k \leq K$:*

$$\sum_{j\in\mathcal{N}_k} \|\boldsymbol{w}_j(t_1)\|^2 \geq \exp\left( -\frac{2p^{p+2}K}{p(p-2)\tilde{\Delta}_2\tilde{\Delta}_1^{p-2}} \right) W_{\min}^2\epsilon^2 . \tag{H.47}$$

*Proof.* **The proof will be in two parts**: first, we define, for each $k$,

$$t_{\text{aux}}^{(k)} := \inf\left\{ t: \min_{j\in\mathcal{N}_k} c_{kj}(t) \geq \frac{2}{3} \right\} \overset{\text{(By its definition)}}{\leq} t_1 , \tag{H.48}$$

and show that

$$\sum_{j\in\mathcal{N}_k} \|\boldsymbol{w}_j(t_{\text{aux}}^{(k)})\|^2 \geq \exp\left( -\frac{2p^{p+2}K}{p(p-2)\tilde{\Delta}_2\tilde{\Delta}_1^{p-2}} \right) \sum_{j\in\mathcal{N}_k} \|\boldsymbol{w}_j(0)\|^2 . \tag{H.49}$$

Then we show that $\sum_{j\in\mathcal{N}_k} \|\boldsymbol{w}_j(t_1)\|^2$ is non-decreasing during $[t_{\text{aux}}^{(k)}, t_1]$.

**Lower bound at $t_{\text{aux}}^{(k)}$:** We shall focus on the case $1 \leq k \leq K_1$. In the proofs of Proposition 4, we have shown in (F.18) that when $t \leq t_{\text{aux}}^{(k)} \leq \bar{t}_1$, the following is true: $\forall j \in \mathcal{N}_k$

$$\frac{d}{dt}c_{kj} \geq \tilde{\Delta}_2 p c_{kj}^{p-1} , \tag{H.50}$$

By Lemma 9, we have

$$t_{\text{aux}}^{(k)} = \inf\left\{ t: c_{kj} \geq \frac{2}{3} \right\} \leq \frac{1}{p(p-2)\tilde{\Delta}_2\tilde{\Delta}_1^{p-2}} . \tag{H.51}$$

Now we are ready to lower bound $\sum_{j \in \mathcal{N}_k} \|\boldsymbol{w}_j(t_{\text{aux}}^{(k)})\|^2$: In the same way we derived (H.27), we can also obtain: for $t \leq t_1$,

$$\frac{d}{dt}\|\boldsymbol{w}_j\|^2 \geq -2^{p+1}K(1+4\epsilon\sqrt{h}W_{\max}^2)\|\boldsymbol{w}_j\|^2 \geq -2^{p+2}K\|\boldsymbol{w}_j\|^2, \tag{H.52}$$

thus

$$\frac{d}{dt}\sum_{j \in \mathcal{N}_k}\|\boldsymbol{w}_j\|^2 \geq -2^{p+2}K\sum_{j \in \mathcal{N}_k}\|\boldsymbol{w}_j\|^2. \tag{H.53}$$

Finally, by Gr$\acute{}$onwall's inequality, we have

$$\sum_{j \in \mathcal{N}_k}\|\boldsymbol{w}_j(t_{\text{aux}}^{(k)})\|^2 \geq \exp\left(-2p^{p+2}Kt_{\text{aux}}^{(k)}\right)\sum_{j \in \mathcal{N}_k}\|\boldsymbol{w}_j(0)\|^2$$

$$\geq \exp\left(-\frac{2p^{p+2}K}{p(p-2)\tilde{\Delta}_2\tilde{\Delta}_1^{p-2}}\right)W_{\min}^2\epsilon^2.$$

**Norm is non-decreasing afterward** The techniques we will be using here is similar to those used in proving previous lemma, so we describe the argument briefly.

Suppose $1 \leq k \leq K$, we have the norm dynamics

$$\frac{d}{dt}\|\boldsymbol{w}_j\|^2$$

$$= -\frac{2}{N}\left(\sum_{i:\langle\boldsymbol{x}_i,\boldsymbol{w}_j\rangle>0}\nabla_{\hat{y}}\ell_i\left(\left\langle\boldsymbol{x}_i,\frac{\boldsymbol{w}_j}{\|\boldsymbol{w}_j\|}\right\rangle\right)^p\right)\|\boldsymbol{w}_j\|^2$$

$$= \frac{2}{N}\left(\sum_{i:\langle\boldsymbol{x}_i,\boldsymbol{w}_j\rangle>0}y_i\left(\left\langle\boldsymbol{x}_i,\frac{\boldsymbol{w}_j}{\|\boldsymbol{w}_j\|}\right\rangle\right)^p\right)\|\boldsymbol{w}_j\|^2 + \underbrace{\mathcal{O}(\epsilon)}_{\text{Recall how we handle }\Gamma_1\text{ in the proof of Lemma 6}}$$

$$\geq \frac{2}{N}\left(\sum_{i\in\mathcal{I}_k}\left|\left\langle\boldsymbol{x}_i,\frac{\boldsymbol{w}_j}{\|\boldsymbol{w}_j\|}\right\rangle\right|^p - \sum_{l\neq k}\sum_{i\in\mathcal{I}_l}\left|\left\langle\boldsymbol{x}_i,\frac{\boldsymbol{w}_j}{\|\boldsymbol{w}_j\|}\right\rangle\right|^p\right)\|\boldsymbol{w}_j\|^2 + \mathcal{O}(\epsilon)$$

$$\geq \frac{2}{N}\left(\sum_{i\in\mathcal{I}_k}\underbrace{\left(c_{kj}+\left\langle\boldsymbol{x}_i,\frac{\boldsymbol{w}_j}{\|\boldsymbol{w}_j\|}\right\rangle\right)^p}_{\text{Taylor expansion, refer to (H.46)}} - \sum_{l\neq k}\sum_{i\in\mathcal{I}_l}\underbrace{\left(|c_{lj}|+\left|\left\langle\boldsymbol{x}_i,\frac{\boldsymbol{w}_j}{\|\boldsymbol{w}_j\|}\right\rangle\right|\right)^p}_{\text{Taylor expansion, refer to (H.16)}}\right)\|\boldsymbol{w}_j\|^2 + \mathcal{O}(\epsilon)$$

$$\geq 2\left(c_{kj}^p - \sum_{l\neq k}|c_{lj}|^p - \mathcal{O}\left(\sqrt{\log\frac{K^2N}{\delta}}\alpha\right) - \mathcal{O}\left(\alpha^2\right)\right)\|\boldsymbol{w}_j\|^2 + \mathcal{O}(\epsilon).$$

When $c_{kj} \geq \frac{2}{3}$, we have

$$c_{kj}^p - \sum_{l\neq k}|c_{lj}|^p \geq c_{kj}^p - (1-c_{kj}^2)^{\frac{p}{2}} > 0, \tag{H.54}$$

then for sufficiently small $\alpha$ and $\epsilon$, we have $\frac{d}{dt}\|\boldsymbol{w}_j\|^2 \geq 0$. Then during $t_{\text{aux}}^{(k)} \leq t \leq t_1$, we have

$$\frac{d}{dt}\sum_{j \in \mathcal{N}_k}\|\boldsymbol{w}_j\|^2 \geq 0. \tag{H.55}$$

The proof is finished. $\qquad\qquad\square$

**Lemma 18.** *15[Restated] Given some $0 < \Delta < \frac{1}{4}$, if for some $z(t)$, the following holds*

$$\frac{d}{dt}z \geq (1-z-\Delta)z, \; z(0) = z_0, \; z(T) = z_1, \tag{H.56}$$

*for some $0 < z_0 \leq \frac{1}{4}$, and $z_0 \leq z_1 < 1 - \Delta$. Then the travel time $T$ for $z(t)$ to go from $z_0$ to $z_1$ satisfies:*

$$T \leq 2 \left( \log \frac{1}{1 - z_1 - \Delta} + \log \frac{1}{z_0} \right) . \tag{H.57}$$

*Proof.* We have

$$\int_{z_0}^{z_1} \frac{1}{(1 - z - \Delta)z} dz \geq \int_0^T dt , \tag{H.58}$$

thus

$$T \leq \frac{1}{1 - \Delta} \left( \log \frac{1 - z_0 - \Delta}{1 - z_1 - \Delta} + \log \frac{z_1}{z_0} \right) \leq 2 \left( \log \frac{1}{1 - z_1 - \Delta} + \log \frac{1}{z_0} \right) . \tag{H.59}$$

$\square$

**Lemma 16** (Restated). *Condition on good event $\mathcal{E}_{good}$, we have*

$$\sum_{j \in \mathcal{N}_c} \| \boldsymbol{w}_j(t) \|^2 = \tilde{o}(\alpha^2) , \ \forall t \leq T^* . \tag{H.60}$$

*Proof.* We deal with neurons with $\text{sign}(v_j) = +1$, the other case has a similar proof. If $j \in \mathcal{N}_c$, it means $\boldsymbol{w}_{j0}$ is initialized into the void region with $c_{kj}(0) < 0$ and $|c_{kj}(t)| = \Theta(1)$, for $1 \leq k \leq K_1$. Therefore, the inner product between $\boldsymbol{w}_j(0)$ and a data point $x_i$ from the $k$-th cluster is always negative, and this holds continuously as long as $c_{kj}(t) < 0$ and $|c_{kj}(t)| = \Theta(1)$.

We will show that

1. Until $t \leq t^*$, we still have $c_{kj}(t) < 0$ and $|c_{kj}(t)| = \Theta(1)$, thus none of the data in positive clusters activates $\boldsymbol{w}_j$.

2. Then $c_{kj}(t) < 0$ and $|c_{kj}(t)| = \Theta(1)$ suggests that, $\sum_{j \in \mathcal{N}_c} \| \boldsymbol{w}_j \|^2$ has an at most $\mathcal{O}(\alpha^2)$ growth rate. And during $[t^*, T^*]$, with a slightly different argument, $\sum_{j \in \mathcal{N}_c} \| \boldsymbol{w}_j \|^2$ still has an at most $\mathcal{O}(\alpha^2)$ growth rate, thus continually stays at $\tilde{o}(\alpha^2)$.

The a more formal proof requires proof by contradiction, with previous lemmas we have proved, but the provided argument should easily be translated into a proof by contradiction.

**First step**: Given a $j \in \mathcal{N}_c \cup \mathcal{N}_+$ and $1 \leq k \leq K_1$, we have during $t \leq t^*$,

$$\frac{d}{dt} c_{kj} = -\frac{1}{N} \sum_{i: \langle \boldsymbol{x}_i, \boldsymbol{w}_j \rangle > 0} \nabla_{\hat{y}} \ell_i \, p \left( \left\langle \boldsymbol{x}_i, \frac{\boldsymbol{w}_j}{\| \boldsymbol{w}_j \|} \right\rangle \right)^{p-1} \left( \langle \boldsymbol{\mu}_k, \boldsymbol{x}_i \rangle - \left\langle \boldsymbol{x}_i, \frac{\boldsymbol{w}_j}{\| \boldsymbol{w}_j \|} \right\rangle c_{kj} \right)$$

$$= -\frac{1}{N} \sum_{K_1 + 1 \leq l \leq K} \sum_{i \in \mathcal{I}_l: \langle \boldsymbol{x}_i, \boldsymbol{w}_j \rangle > 0} \nabla_{\hat{y}} \ell_i \, p \left( \left\langle \boldsymbol{x}_i, \frac{\boldsymbol{w}_j}{\| \boldsymbol{w}_j \|} \right\rangle \right)^{p-1} \left( \underbrace{\langle \boldsymbol{\mu}_k, \boldsymbol{x}_i \rangle}_{= \mathcal{O}(\frac{\alpha}{\sqrt{D}})} - \left\langle \boldsymbol{x}_i, \frac{\boldsymbol{w}_j}{\| \boldsymbol{w}_j \|} \right\rangle c_{kj} \right)$$

$$= -\frac{1}{N} \sum_{K_1 + 1 \leq l \leq K} \sum_{i \in \mathcal{I}_l: \langle \boldsymbol{x}_i, \boldsymbol{w}_j \rangle > 0} \nabla_{\hat{y}} \ell_i \, p \left( \left\langle \boldsymbol{x}_i, \frac{\boldsymbol{w}_j}{\| \boldsymbol{w}_j \|} \right\rangle \right)^p \underbrace{c_{kj}}_{<0} + \mathcal{O} \left( \frac{\alpha}{\sqrt{D}} \right) .$$

Since $\nabla_{\hat{y}} \ell_i$ is either $< 0$ (during alignment phase) or $= \mathcal{O}(\alpha^2)$ (after norm growth). Then we always have $\frac{d}{dt} c_{kj} = \mathcal{O}(\alpha^2)$. Therefore, $\forall t \leq t^*$

$$c_{kj}(t) \leq c_{kj}(0) + t \cdot \mathcal{O}(\alpha^2) \leq c_{kj}(0) + t^* \mathcal{O}(\alpha^2) = c_{kj}(0) + \mathcal{O} \left( \alpha^2 \log \frac{1}{\alpha} \right) , \tag{H.61}$$

thus, we still have $c_{kj}(t) < 0$ and $|c_{kj}(t)| = \Theta(1)$.

**Second step**: During $[0, t^*]$, since none of the data in positive clusters activates $\boldsymbol{w}_j$, we have

$$
\frac{d}{dt} \|\boldsymbol{w}_j\|^2 = -2 \frac{1}{N} \left( \sum_{i:\langle \boldsymbol{x}_i, \boldsymbol{w}_j \rangle > 0} \nabla_{\hat{y}} \ell_i \left( \left\langle \boldsymbol{x}_i, \frac{\boldsymbol{w}_j}{\|\boldsymbol{w}_j\|} \right\rangle \right)^p \right) \|\boldsymbol{w}_j\|^2
$$

$$
= -2 \frac{1}{N} \left( \sum_{K_1 + 1 \le l \le K} \sum_{i \in \mathcal{I}_l: \langle \boldsymbol{x}_i, \boldsymbol{w}_j \rangle > 0} \nabla_{\hat{y}} \ell_i \left( \left\langle \boldsymbol{x}_i, \frac{\boldsymbol{w}_j}{\|\boldsymbol{w}_j\|} \right\rangle \right)^p \right) \|\boldsymbol{w}_j\|^2 .
$$

Since $\nabla_{\hat{y}} \ell_i$ is either $< 0$ (during alignment phase) or $= \mathcal{O}(\alpha^2)$ (after norm growth). We have $\frac{d}{dt} \|\boldsymbol{w}_j\|^2 = \mathcal{O}(\alpha^2) \cdot \|\boldsymbol{w}_j\|^2$.

During $[t^*, T^*]$, we have $\nabla_{\hat{y}} \ell_i = \mathcal{O}(\alpha^2)$ for all $i$ (as the consequence of $\left| 1 - \sum_{j \in \mathcal{N}_k} \|\boldsymbol{w}_j\|^2 \right| = \mathcal{O}(\alpha^2)$ and Lemma 11). Therefore we still have $\frac{d}{dt} \|\boldsymbol{w}_j\|^2 = \mathcal{O}(\alpha^2) \cdot \|\boldsymbol{w}_j\|^2$.

Then we have $\forall t \le T^*$,

$$
\frac{d}{dt} \sum_{j \in \mathcal{N}_c} \|\boldsymbol{w}_j(t)\|^2 \le \mathcal{O}\left( \exp(\alpha^2 T^*) \right) \sum_{j \in \mathcal{N}_c} \|\boldsymbol{w}_j(0)\|^2 \le \mathcal{O}(1) \sum_{j \in \mathcal{N}_c} \|\boldsymbol{w}_j(0)\|^2 \le \mathcal{O}(\epsilon^2) = \tilde{o}(\alpha^2) . \tag{H.62}
$$

$\square$

**Lemma 17** (Restated). *If the neurons $\{\boldsymbol{w}_j\}_{j=1}^h$ satisfies the following for some $0 \le \delta \le 1$ and $\nu, \zeta > 0$:*

- $\max_k \max_{j \in \mathcal{N}_k} c_{kj}(t) \ge 1 - \delta$;

- $\left| 1 - \sum_{j \in \mathcal{N}_k} \|\boldsymbol{w}_j\|^2 \right| \le \nu$;

- $\sum_{j \in \mathcal{N}^c} \|\boldsymbol{w}_j\|^2 \le \zeta$,

*then $\sup_{\boldsymbol{x} \in \mathbb{S}^{D-1}} \left| f^{(p)}(\boldsymbol{x}; \boldsymbol{\theta}) - F^{(p)}(\boldsymbol{x}) \right| \le K(1 + \nu)(2^p - 1)2\delta + K\nu + \zeta$*

*Proof.*

$$
f^{(p)}(\boldsymbol{x}; \boldsymbol{\theta}) \tag{H.63}
$$

$$
= \sum_{j=1}^h v_j \frac{\sigma^p(\langle \boldsymbol{w}_j, \boldsymbol{x} \rangle)}{\|\boldsymbol{w}_j\|^{p-1}}
$$

$$
= \sum_{j=1}^h \text{sign}(v_j) \|\boldsymbol{w}_j\|^2 \frac{\sigma^p(\langle \boldsymbol{w}_j, \boldsymbol{x} \rangle)}{\|\boldsymbol{w}_j\|^p}
$$

$$
= \sum_{j=1}^h \text{sign}(v_j) \|\boldsymbol{w}_j\|^2 \sigma^p \left( \left\langle \frac{\boldsymbol{w}_j}{\|\boldsymbol{w}_j\|}, \boldsymbol{x} \right\rangle \right)
$$

$$
= \sum_{1 \le k \le K_1} \sum_{j \in \mathcal{N}_k} \|\boldsymbol{w}_j\|^2 \sigma^p \left( \left\langle \frac{\boldsymbol{w}_j}{\|\boldsymbol{w}_j\|}, \boldsymbol{x} \right\rangle \right) - \sum_{K_1 + 1 \le k \le K} \sum_{j \in \mathcal{N}_k} \|\boldsymbol{w}_j\|^2 \sigma^p \left( \left\langle \frac{\boldsymbol{w}_j}{\|\boldsymbol{w}_j\|}, \boldsymbol{x} \right\rangle \right)
$$

$$
+ \sum_{j \in \mathcal{N}^c} \text{sign}(v_j) \|\boldsymbol{w}_j\|^2 \sigma^p \left( \left\langle \frac{\boldsymbol{w}_j}{\|\boldsymbol{w}_j\|}, \boldsymbol{x} \right\rangle \right) . \tag{H.64}
$$

For the first term, we have $\forall \boldsymbol{x} \in \mathbb{S}^{D-1}$

$$
\left| \sum_{1 \le k \le K_1} \sum_{j \in \mathcal{N}_k} \|\boldsymbol{w}_j\|^2 \sigma^p \left( \left\langle \frac{\boldsymbol{w}_j}{\|\boldsymbol{w}_j\|}, \boldsymbol{x} \right\rangle \right) - \sum_{1 \le k \le K_1} \sigma^p(\langle \boldsymbol{\mu}_k, \boldsymbol{x} \rangle) \right|
$$

$$\leq \sum_{1\leq k\leq K_1} \left| \sum_{j\in\mathcal{N}_k} \|\boldsymbol{w}_j\|^2 \sigma^p\left(\left\langle \frac{\boldsymbol{w}_j}{\|\boldsymbol{w}_j\|} - \boldsymbol{\mu}_k + \boldsymbol{\mu}_k, \boldsymbol{x}\right\rangle\right) - \sigma^p(\langle\boldsymbol{\mu}_k,\boldsymbol{x}\rangle)\right|$$

$$\leq \sum_{1\leq k\leq K_1} \left| \sum_{j\in\mathcal{N}_k} \|\boldsymbol{w}_j\|^2 \sigma^p\left(\langle\boldsymbol{\mu}_k,\boldsymbol{x}\rangle + \left\|\frac{\boldsymbol{w}_j}{\|\boldsymbol{w}_j\|} - \boldsymbol{\mu}_k\right\|\right) - \sigma^p(\langle\boldsymbol{\mu}_k,\boldsymbol{x}\rangle)\right|$$

$$= \sum_{1\leq k\leq K_1} \left| \sum_{j\in\mathcal{N}_k} \|\boldsymbol{w}_j\|^2 \sigma^p\left(\langle\boldsymbol{\mu}_k,\boldsymbol{x}\rangle + 2(1-c_{kj})\right) - \sigma^p(\langle\boldsymbol{\mu}_k,\boldsymbol{x}\rangle)\right|$$

$$\leq \sum_{1\leq k\leq K_1} \left| \sum_{j\in\mathcal{N}_k} \|\boldsymbol{w}_j\|^2 \sigma^p\left(\langle\boldsymbol{\mu}_k,\boldsymbol{x}\rangle + 2(1-c_{kj})\right) - \sum_{j\in\mathcal{N}_k}\|\boldsymbol{w}_j\|^2\sigma^p(\langle\boldsymbol{\mu}_k,\boldsymbol{x}\rangle)\right|$$

$$+ \sum_{1\leq k\leq K_1} \left| \sum_{j\in\mathcal{N}_k} \|\boldsymbol{w}_j\|^2 \sigma^p(\langle\boldsymbol{\mu}_k,\boldsymbol{x}\rangle) - \sigma^p(\langle\boldsymbol{\mu}_k,\boldsymbol{x}\rangle)\right|$$

$$\leq \sum_{1\leq k\leq K_1} (1+\nu)\left|\sigma^p\left(\langle\boldsymbol{\mu}_k,\boldsymbol{x}\rangle + 2(1-c_{kj})\right) - \sigma^p(\langle\boldsymbol{\mu}_k,\boldsymbol{x}\rangle)\right| + \sum_{1\leq k\leq K_1} \nu\left|\sigma^p(\langle\boldsymbol{\mu}_k,\boldsymbol{x}\rangle)\right|$$

$$\leq \sum_{1\leq k\leq K_1} (1+\nu)\left|\sigma^p\left(\langle\boldsymbol{\mu}_k,\boldsymbol{x}\rangle + 2(1-c_{kj})\right) - \sigma^p(\langle\boldsymbol{\mu}_k,\boldsymbol{x}\rangle)\right| + K_1\nu$$

$$\leq K_1(1+\nu)(2^p-1)2\delta + K_1\nu,$$

where the last inequality is due to the following derivation (notice that ReLU $\sigma(z)$ is non-decreasing in $z$, and polynomial $z^p$ is non-decreasing for $z > 0$)

$$\left|\sigma^p\left(\langle\boldsymbol{\mu}_k,\boldsymbol{x}\rangle + 2(1-c_{kj})\right) - \sigma^p(\langle\boldsymbol{\mu}_k,\boldsymbol{x}\rangle)\right|$$
$$= \sigma^p\left(\langle\boldsymbol{\mu}_k,\boldsymbol{x}\rangle + 2(1-c_{kj})\right) - (\langle\boldsymbol{\mu}_k,\boldsymbol{x}\rangle)$$
$$\leq (1+2\delta)^p - 1 \leq (2^p-1)2\delta.$$

Similarly, for the second term, we have $\forall \boldsymbol{x}\in\mathbb{S}^{D-1}$

$$\left|\sum_{K_1+1\leq k\leq K}\sum_{j\in\mathcal{N}_k}\|\boldsymbol{w}_j\|^2\sigma^p\left(\left\langle\frac{\boldsymbol{w}_j}{\|\boldsymbol{w}_j\|},\boldsymbol{x}\right\rangle\right) - \sum_{K_1+1\leq k\leq K}\sigma^p(\langle\boldsymbol{\mu}_k,\boldsymbol{x}\rangle)\right| \leq K_2(1+\nu)(2^p-1)2\delta + K_2\nu.$$

Lastly, for the third term, we have

$$\left|\sum_{j\in\mathcal{N}^c}\mathrm{sign}(v_j)\|\boldsymbol{w}_j\|^2\sigma^p\left(\left\langle\frac{\boldsymbol{w}_j}{\|\boldsymbol{w}_j\|},\boldsymbol{x}\right\rangle\right)\right| \leq \sum_{j\in\mathcal{N}^c}\|\boldsymbol{w}_j\|^2 \leq \zeta.$$

Therefore, for any $\boldsymbol{x}\in\mathbb{S}^{D-1}$, we have

$$\left|f^{(p)}(\boldsymbol{x};\boldsymbol{\theta}) - F^{(p)}(\boldsymbol{x})\right| \leq \left|\sum_{1\leq k\leq K_1}\sum_{j\in\mathcal{N}_k}\|\boldsymbol{w}_j\|^2\sigma^p\left(\left\langle\frac{\boldsymbol{w}_j}{\|\boldsymbol{w}_j\|},\boldsymbol{x}\right\rangle\right) - \sum_{1\leq k\leq K_1}\sigma^p(\langle\boldsymbol{\mu}_k,\boldsymbol{x}\rangle)\right|$$

$$+ \left|\sum_{K_1+1\leq k\leq K}\sum_{j\in\mathcal{N}_k}\|\boldsymbol{w}_j\|^2\sigma^p\left(\left\langle\frac{\boldsymbol{w}_j}{\|\boldsymbol{w}_j\|},\boldsymbol{x}\right\rangle\right) - \sum_{K_1+1\leq k\leq K}\sigma^p(\langle\boldsymbol{\mu}_k,\boldsymbol{x}\rangle)\right|$$

$$+ \left|\sum_{j\in\mathcal{N}^c}\mathrm{sign}(v_j)\|\boldsymbol{w}_j\|^2\sigma^p\left(\left\langle\frac{\boldsymbol{w}_j}{\|\boldsymbol{w}_j\|},\boldsymbol{x}\right\rangle\right)\right|$$

$$\leq K(1+\nu)(2^p-1)2\delta + K\nu + \zeta.$$

$\square$

