# OpenReview forum: "Gradient Flow Provably Learns Robust Classifiers for Orthonormal GMMs"
_ICML.cc/2025/Conference — ICML 2025 poster_

### Official Review · Reviewer_AZei · 2025-03-12

**Overall Recommendation:** 4

**Summary:**

This paper investigates the problem of adversarial robustness in deep learning classifiers and provides a theoretical framework demonstrating that standard training methods, specifically gradient flow, can lead to a provably robust classifier under certain conditions. Unlike existing approaches that require adversarial training or explicit defense mechanisms, this work focuses on the natural robustness that emerges from the structure of the data.

This work focuses on a binary classification problem where data is drawn from an isotropic Gaussian mixture model with orthonormal cluster centers. The theoretical study first establish the limit of adversarial robustness by showing that no classifier can defend against an L2 attack of radius greater than $\sqrt{2}/2$, while also proving the existence of an optimal classifier that achieves this bound. This classifier follows a nearest-cluster rule, assigning labels based on the closest cluster center in feature space.

The core result of the work is the proof that a two-layer neural network with a polynomial ReLU (pReLU) activation function, trained using gradient flow, can successfully learn this optimal robust classifier without requiring adversarial training. This stands in contrast to standard ReLU networks, which fail to achieve robustness due to their inability to internally learn the multi-cluster structure of the data. This work  further provides a rigorous convergence analysis demonstrating that gradient flow on pReLU networks can find a classifier that is not only highly accurate on clean data but also robust to adversarial perturbations.

**Claims And Evidence:**

The claims made in the submission are supported by a combination of theoretical proofs and empirical observations. The core theoretical contributions, particularly the characterization of the maximum achievable adversarial robustness and the proof that gradient flow on a two-layer network with a polynomial ReLU activation can learn a robust classifier, are well-supported by rigorous mathematical derivations. This work establishes upper bounds on robustness and demonstrate that their proposed classifier achieves these bounds, providing convincing justification through formal theorems and proofs.

One of the key strengths of the paper is its detailed convergence analysis, which demonstrates that gradient flow on a properly parameterized network finds a nearly optimal robust classifier under the assumed data distribution. This theoretical backing is essential in supporting the claim that standard training, without adversarial examples, can yield robustness in certain structured settings. The results align with prior works that study adversarial robustness in relation to data geometry.

**Essential References Not Discussed:**

None

**Experimental Designs Or Analyses:**

I also checked the experiments presented in the paper.  The experiments primarily serve to illustrate key theoretical results rather than to provide a comprehensive empirical evaluation of the proposed approach. The authors use synthetic data generated from an Orthonormal Gaussian Mixture Model (GMM), which is a well-chosen setting to validate their theoretical analysis but does not necessarily reflect the complexities of real-world datasets.

The experimental design effectively demonstrates the failure of standard ReLU networks in learning robust classifiers and the corresponding success of pReLU networks under gradient flow. The figures provided show that specific activation functions play a crucial role in achieving robustness, which supports their claim that the architecture must be designed to exploit the structure of the data. However, the evaluation lacks a comparison with standard adversarial training techniques or alternative defense mechanisms, making it difficult to assess how well the proposed approach performs relative to existing state-of-the-art methods.

One major concern over the experiment design is the absence of evaluations on widely-used benchmark datasets such as CIFAR-10, MNIST, or ImageNet. While synthetic experiments are useful for validating the theoretical results, their practical significance remains unclear. The paper does not explore how well the proposed method generalizes to more complex, high-dimensional data distributions where the assumptions of the Orthonormal GMM may not hold.

In summary, I think the experimental design supports the theoretical findings within the specific scope of the problem but does not fully validate the broader applicability of the approach. A more extensive empirical evaluation, including real-world datasets and comparisons with adversarial training methods, would be necessary to establish its practical relevance.

**Methods And Evaluation Criteria:**

The proposed method in this paper is aligned with the theoretical analysis. The primary methodological contribution is the demonstration that gradient flow on a two-layer neural network with a polynomial ReLU (pReLU) activation function can provably learn a robust classifier under the specified conditions. This is a novel and interesting result, particularly in showing that standard training methods can achieve adversarial robustness **without explicit adversarial training**. The evaluation criterion used to assess the robustness of the learned classifier—the maximum L2 perturbation it can withstand while maintaining high accuracy—is a reasonable and widely accepted metric in adversarial robustness research. The paper provides formal derivations to establish the theoretical upper bound on robustness and proves that the proposed  method can achieve this bound empirically, which is a strong result.

**Other Comments Or Suggestions:**

I would suggest the authors conduct experiments on real-world datasets such as CIFAR-10, MNIST, or ImageNet, which may significantly enhance the paper’s practical relevance. In addition, this paper could also benefit from a comparative analysis with adversarial training methods, such as PGD adversarial training and randomized smoothing. Comparing with the robust training baseline can consolidate the adversarial robustness claim made in this work.

The paper shows that ReLU networks fail, yet pReLU networks succeeds in mitigating adversarial attacks. How about using other variants of ReLU, such as GELU, or leaky ReLU ? Could they yield similar robustness properties ?

I also suggest the authors discuss potential extensions beyond the Gaussian mixture model. The structured nature of the data model is useful for theoretical analysis, but adversarial robustness is often studied in much more complex, high-dimensional, and non-Gaussian settings. Exploring whether similar guarantees can be derived for broader classes of data distributions can make the findings more broadly applicable.

**Other Strengths And Weaknesses:**

The most significant weakness is the lack of empirical validation on real-world datasets. The theoretical results are developed within the framework of an Orthonormal Gaussian Mixture Model (GMM), which is a well-structured and mathematically convenient setting but does not necessarily reflect the complexity of natural datasets. Without experiments on standard benchmarks such as CIFAR-10, MNIST, or ImageNet, it remains unclear whether the proposed method would work effectively in practical scenarios where data distributions are more intricate and do not satisfy the strict assumptions of the orthonormal GMM.

The initialization requirements also present a potential weakness. The analysis relies on a non-degenerate initialization with a positive gap to ensure that neurons align correctly with the cluster structure of the data. However, this condition is not naturally satisfied by random initialization, meaning that additional interventions may be needed to ensure that gradient flow leads to a robust classifier. This assumption is acknowledged in the paper, but there is little discussion on how one might relax it or adapt standard training methods to meet these conditions in practice.

As well, the requirement that the data follows a high-dimensional mixture of orthonormal Gaussians is a strong assumption that may not hold in many real-world applications. This limits the generalizability of the results

**Questions For Authors:**

Please check the suggestions made.

**Relation To Broader Scientific Literature:**

The main contribution in this paper is closely related to the study of adversarial robustness, neural network training dynamics, and implicit regularization in deep learning.

**Theoretical Claims:**

I checked the key theoretical proofs provided in the submission and found that the high-level arguments and overall structure of the proofs are sound. The authors develop their results with careful attention to the necessary assumptions, such as the non-degenerate parameter initialization and the scaling conditions required for the convergence analysis of the gradient flow on the pReLU network. Their derivations for the maximum L2-robustness and the subsequent convergence guarantees are presented with clarity and logical consistency. While I do not verify every technical detail down to the last inequality, the main steps and techniques—such as the use of concentration inequalities and the analysis of activation dynamics—appear to be correct and align with established methods in the literature. Overall, I think the proofs convincingly support the theoretical claims in the paper.

---

> ### Author Rebuttal · Authors · 2025-03-31
>
> Thank you for the valuable comments and suggestions. Here are our responses to your questions:
>
> **Experiments on real datasets**: As we stated in Remark 2 (and we will expand it as per reviewer MTUy's suggestion), we focus on developing theoretical results in this paper, and the related experiments on real datasets (MNIST, CIFAR) have been presented in prior works. Nonetheless, we will add experiments on synthetic data that validate our Theorems (See our rebuttal *"Experiments: degenerate v.s. non-degenerate initialization"* to the reviewer hboM). Lastly, the comparison with adversarial training suggested by the reviewer is an interesting future research topic.
>
> **Other choices of activation functions**: If one uses GELU or Leaky-ReLU, the trained network has a similar level of robustness to those (ReLU, linear, Tanh) that are non-robust. We are happy to add their corresponding plot to Figure 2.
>
> **Beyond Guassian models**: We will add a concluding remark discussing future work. The future work will be along the lines of extending the current analysis to more complex data models, for example, union of low-dimensional subspaces.
>
> We hope our rebuttal addresses your questions. If you have additional questions/concerns, please feel free to post them during the discussion phase.

---

### Official Review · Reviewer_hboM · 2025-03-14

**Overall Recommendation:** 2

**Summary:**

A common concern in the design of deep learning systems is their susceptibility to imperceptible noise. The authors approach this problem from the angle of finding the maximum adversarial perturbation tolerated by a neural network, without needing adversarial training. It is clarified that this often conditions on the data geometry, since the proof of a robust classifier depends on a sufficiently large margin between each class-conditional cluster in the desired classification space. The desire is to provably find a robust classifier while maintaining clean accuracy.  The submission proposes theoretical proof that on specific data models, the maximum robustness any classifier can achieve is based on the separation between the class-conditioned probability mass. Although the data distribution follows specific structural patterns that enable learning a robust classifier, the ability to learn the robust classifier by gradient descent is not viable unless using a polynomial ReLU (pReLU) proposed in prior work. Based on a balanced gaussian mixture model, the authors first show that a nearly optimal classifier can be constructed from nearest-cluster rules. Unfortunately, gradient descent must derive classification rules over the trajectory of optimizing weights, which makes learning the robust classifier more difficult. With bounded network parameter initialization, the main result tries to prove the conjecture provided in prior work, which states that gradient descent on a pReLU network with $p > 2$ eventually converges to an adversarially robust network in the radius close to $\frac{\sqrt{2}}{2}$. This is said to hold for an arbitrary classifier $f$ within a specific distance measure (proposition 3). The result assumes: a balanced data sampling of the GMM (Assumption 1), and given the definition of non-degeneracy gap using a union of Voronoi regions and void regions, there is a limit on where individual neuron weights may initialize given the weight's angle to nearby cluster centers (Assumption 2).

## Post-rebuttal update

Thanks to the authors for a detailed response and new experiments, which will improve the next version. The main pain point remaining is the layout and organization of the main text, which is difficult to remedy in a single review cycle without a major revision, so I will keep my original score.

**Claims And Evidence:**

- The main proof is structured similar to prior work studying GF dynamics - the GF problem is split into an alignment phase and a convergence phase. The alignment of neuron weights can be bounded given the assumptions on initialization and non-degeneracy. The convergence can be checked based on monotonic growth of the neuron norm in Voronoi regions and small bounded amount in void regions. The proof seems reasonable and an iterative step over prior work.

**Essential References Not Discussed:**

N/A

**Experimental Designs Or Analyses:**

N/A

**Methods And Evaluation Criteria:**

The authors provide some small experiments using synthetic data in Appendix B, although they mainly relate to comparison of regular polynomial ReLU and pReLU.

**Other Comments Or Suggestions:**

N/A

**Other Strengths And Weaknesses:**

- The convergence proofs would be a useful extension for the conjectures proposed in prior work.
- The submission mainly suffers from structural and organization issues. The main theorem is not presented until page 6. The introduction runs until page 3 and contains  information that is not necessarily needed for introducing the problem domain (e.g., defining GMM, informal versions of proofs). Most of section 2 could likely be moved to an appendix since it relates to the idealized Bayes classifier with cluster center rules. The main text ends abruptly, but should ideally devote more space to synthetic experiments to validate assumptions, closing remarks, and future work.
- The main results are interesting but I was expecting to see some verification of assumptions (1 and 2) on small-scale synthetic data, e.g., demonstration of the effect from degenerate and non-degenerate initializations, or when initialization places neurons close to region borders. It would've been interesting to check the classifier training which obeys Assumption 1 but not Assumption 2 (non-degeneracy shape). It makes sense to need a specific placement of neuron weights relative to cluster centers for the convergence analysis, but it wasn't clear how necessary this is during practical implementation.

**Questions For Authors:**

N/A

**Relation To Broader Scientific Literature:**

- The proposed theorems may have implications for prior work which offered conjectures on the convergence properties of pReLU-based 2-layer neural networks.
- Future work may extend the results to different classification losses, since the current format assumes a specific loss (due to difficulty in characterizing neuron movement during training).

**Theoretical Claims:**

I did not see any major errors with the proofs.

---

> ### Author Rebuttal · Authors · 2025-03-31
>
> Thank you for the valuable comments and suggestions. Here are our responses to your questions and concerns:
>
> **Structure of this paper**: We have shared our view on the current structure of our paper in our rebuttal *"Organization/presentation of this paper"* to reviewer MTUy, and we will make revisions to our manuscript to further improve its presentation and clarity. Regarding your specific suggestions, we will add numerical experiments to the manuscript (please see below) and a concluding remark discussing future work. The future work will be along the lines of extending the current analysis to more complex data models, for example, union of low-dimensional subspaces, and possibly extending to the transfer learning setting (Please see our rebuttal *"Practical implication of our results"* to reviewer MTUy) for end-to-end robustness certificates, when provided some properties of the feature extractor.
>
> **Experiments: degenerate v.s. non-degenerate initialization**: Great suggestions. We have conducted numerical experiments on synthetic GMM data to illustrate how this (non)degeneracy of the initialization affects the convergence of GD. These experiments validate our Theorem 1, and also show that GD still converges under random initialization (which does not satisfy our Assumption 2) except with a slightly slower alignment between neurons and cluster centers, which agrees with our discussion in our remark "Limitation of our current result." We will add these experiments to the revised manuscript.
>
> We describe these numerical experiments in detail: we consider, as per the reviewer's suggestion, two types of initialization:
>  - *Random initialization*: the initialization follows Assumption 1, with entries of initialization shape $w_{j0},j=1,\cdots,h$ all i.i.d. samples from standard Gaussian. As we discussed, this does not satisfy our Assumption 2 in high-dimension scenarios.
>  - *Non-degenerate initialization*: We nudge the above random initialization shape toward cluster centers to increase its non-degeneracy gap. Specifically, for every $j$, we let $w_{j0}\leftarrow w_{j0}+\delta \cdot(\mu-w_{j0})$, where $\mu$ is randomly uniformly selected from one of the cluster centers $\mu_k,k=1,\cdots,K$. The resulting new initialization shape has a non-degeneracy gap of roughly $\delta$, thus satisfying Assumption 2. We also adjust the $|v_j|$ accordingly so the initialization satisfies Assumption 1.
>  - The initialization scale is $\epsilon=1e-7$
>
> We run GD with step size $0.2$ on a synthetic GMM dataset of size $n=5000$ with $D=1000, K_1=5, K_2=5,\alpha=0.1$, and keep track the following:
>  - *Alignment*: The alignment measures we are interested in are $\max_k\cos(\mu_k,w_j), j=1,\cdots,h$, and our Theorem 1 and its proof suggests that they converge to close to 1 after sufficient training time. For clarity, we report their mean value and the 1st quantile.
>  - *Loss*: The mean square loss.
>  - *Distantance to $F^{(p)}$*: The quantity $\sup_{x\in S^{D-1}}|f^{(p)}(x;\theta)-F^{(p)}(x)|$. We estimate this quantity by randomly sampling large batches of $x$ and running projected gradient ascent on this distance.
>
> We summarize the findings
>  1. GD converges and finds $F^{(p)}$ in both cases, thus verifies our Theorem 1;
>  2. The alignment between neurons and cluster centers is slower for random initialization than with the nudged initialization with $\delta=0.05$ non-degeneracy gap. This agrees with our discussion in "Limitation of our current result": having some non-degeneracy gap skips a "burn-in" phase for the neurons' directional dynamics,
>
> from the following tables:
>
> |GD, random init.|||||||||
> |---|---|---|---|---|---|---|---|---|
> | Iterations| t=0| t=50 | t=100  | t=150 | t=200 | t=300 | t=400|t=500|
> | Alignment (mean)| 0.05 | 0.06 | 0.14  | 0.29 | 0.47 | 0.69 |0.80 | 0.80|
> | Alignment (1st quantile)|0.04| 0.06 | 0.04  | 0.04 | 0.05 | 0.14 |1.00 | 1.00|
> | Loss | 1.00| 1.00 | 1.00  | 1.00 | 1.00 | 9.99e-1 | 4.97e-1|1.30e-4|
> | Distance to $F^{(p)}$| 5.33e-1| 5.20e-1 | 5.25e-1  | 5.14e-1 | 5.21e-1 | 5.21e-1 | 5.05e-1|3.89e-5|
>
> |GD, non-degenergate init. $\delta=0.05$|||||||||
> |---|---|---|---|---|---|---|---|---|
> | Iterations| t=0| t=50 | t=100  | t=150 | t=200 | t=300 | t=400|t=500|
> | Alignment (mean)| 0.07 | 0.12 | 0.34  | 0.58 | 0.77 | 0.92 |0.95 | 0.95|
> | Alignment (1st quantile)|0.05| 0.05 | 0.08  | 0.14 | 0.48 | 1.00 |1.00 | 1.00|
> | Loss | 1.00| 1.00 | 1.00  | 1.00 | 1.00 | 9.92e-1 | 1.94e-1|9.15e-5|
> | Distance to $F^{(p)}$| 5.33e-1| 5.20e-1 | 5.25e-1  | 5.14e-1 | 5.21e-1 | 5.18e-1 | 4.80e-1|1.57e-5|
>
> Note: 1) for the alignment, $1.00$ is a rounded value; 2) by the end of the training, "GD, random initialization" has a lower mean alignment because it has more neurons in the void regions that do not receive GD update during training.
>
> We hope our rebuttal addresses your concerns regarding the organization of our paper. If you have additional questions/concerns, please feel free to post them during the discussion phase.

---

### Official Review · Reviewer_MTUy · 2025-03-15

**Overall Recommendation:** 2

**Summary:**

This paper presents new theoretical findings regarding the feasibility of achieving robustness without adversarial training. Specifically, the paper focuses on a specific data model: a mixture of Gaussian distributions whose cluster centers (i.e., mean vectors of each Gaussian distribution) are orthonormal vectors. The paper first proves that $\ell_2$-robustness cannot be achieved for a particular attack radius. However, for smaller attack radii, the adversary’s attack success probability can be lower-bounded for the Bayes optimal classifier, implying the robustness of the Bayes optimal classifier for the given data model. Then, the paper presents its theoretical achievement that, for the given data model, it is feasible to train a pReLU network without adversarial training with a proper assumption on the parameter initialization.

## Update after rebuttal
I appreciate the authors' rebuttal. However, I don't think that the rebuttal properly addressed my concern on paper organization and I can also see that I'm not the only reviewer who concerned about the paper writing as a major issue of the paper. I'll keep my initial rating of 2.

**Claims And Evidence:**

All the claims contain proofs in appendices.

**Essential References Not Discussed:**

The paper cited the needed references well.

**Experimental Designs Or Analyses:**

This paper presents purely theoretical findings and does not contain any experiments.

**Methods And Evaluation Criteria:**

This paper neither proposes a new method nor evaluates it.

**Other Comments Or Suggestions:**

1. Please organize the paper's contents to have a better structure as an academic paper. For example, you can present the summary of the theoretical findings first, then a separate proof sketch section, and then another section containing additional content, such as a comparison to prior work and limitations. Also, please add a conclusion section that summarizes the paper's achievements.
2. Please be more selective about what you present in the main body of the paper. For example, while Propositions 1 and 2 are interesting theoretical statements, I don’t think they have more significant value than the results in Section 3. Moving those Propositions to the Appendix and discussing the results in Section 3 more would be better.
3. Be clear about the message the paper wants to deliver to the ML community. The paper proves some interesting theoretical statements, but the reason why the findings are significant is unclear.

**Other Strengths And Weaknesses:**

### Strengths
1. To the best of my knowledge, the presented theoretical results are novel.
2. We still lack a theoretical understanding of adversarial training, and this paper adds some new findings to the theory.

### Weaknesses
1. The paper is not well-organized, which makes it harder to understand.
2. The theoretical findings are interesting, and the paper contains many findings. However, the practical implications of those findings are unclear. In Remark 2, the paper referred to other papers to find such practical implications of the problem. (Even this should be discussed separately in a section discussing related works.) However, those papers do not explain the message the reader should get from **this paper**.

**Questions For Authors:**

1. Regarding the impact of this paper, Remark 2 refers to other papers about the practical implications of the problem setting. Please briefly explain the practical implications to me. (In my opinion, this should be explicitly written in the paper.)
2. What do the paper's findings imply related to the practical value of the problem? Specifically, how can those findings benefit the ML community?

**Relation To Broader Scientific Literature:**

This paper contains an interesting finding that adversarial robustness is achievable without adversarial training for a specific data model with some assumptions. However, the impact of the findings is unclear to me.

**Theoretical Claims:**

I read most of the main body, including the proof sketch. The proof idea in the sketch makes sense.

---

> ### Author Rebuttal · Authors · 2025-03-31
>
> Thank you for the valuable comments and suggestions. Here are our responses to your questions and concerns:
>
> **Organization/presentation of this paper**: Although this is a concern raised by two reviewers (MTUy and hboM), we do not think there is any major issue with the organization of our manuscript. In fact, Reviewer KSsi finds the paper "extremely well-written". It is our view that the disagreement comes from the fact that in the main sections after the introduction, we put more emphasis on the interpretation of our assumptions and results (and Reviewer KSsi thinks they are "interesting", "very nice, and enlightening"), and relatively less emphasis on certain aspects (message to the ML community, practical implications). We thank reviewers MTUy and hboM for pointing out those issues. We address them in the rebuttal, and they can be easily incorporated into our current manuscript.
>
> **What message should one get from this paper?** As a theoretical paper, we believe our messages are clearly stated.
> - The paper title summarized our theoretical contributions;
> - Our message appeared in the abstract: "This paper shows that for certain data distributions, one can learn a provably robust classifier using standard learning methods and without adding a defense mechanism";
> - We outlined our messages in the introduction section in detail: (0) We aim to answer whether standard training methods can find a robust classifier without adversarial examples. We show that for GMM data, (1) data concentration ensures the existence of robust classifiers, and class separation determines the maximum achievable robustness; However, (2) GD on shallow networks can not find a robust classifier unless the activation is carefully chosen, and we show the GF probably finds a robust classifier with appropriate choices; (3) These results highlight that finding robust classifier by standard training is possible but requires a joint understanding of the data structure and the training dynamics of GD algorithms.
>
> Based on the reviewers' comments, we feel their concern arises because, after the introduction, the main body of the paper emphasizes the technical presentation of our results. To resolve this issue, we will remark on how our results reflect these messages in the main sections.
>
> **Practical implication of our results** We will expand Remark 2 to address the practical implications of our results. We explain them here: In the case of transfer learning, one is given a pretrained feature extractor and aims to train a shallow network classifier. Our theorem provides guidance in obtaining robust shallow network classifiers with standard training for tasks with coarse labels. By tasks with coarse labels, we mean that each class includes many sub-classes, but the label only reveals the class membership. For example, when classifying images of cats and dogs, the dog (cat) class may include different breeds. Notably, many feature extractors, like CLIP, are trained on a massive dataset that contains subclass information and thus can "distinguish" subclasses. That is, the extracted feature from different subclasses has nearly an orthogonal GMM structure (verified for CLIP embeddings of CIFAR in Li et al. 2024). Therefore, classifying these extracted features only with coarse class labels is close to the learning problem we entertained in this paper, and our theorem sheds light on how to train the shallow network with appropriate activation for robustly classifying these features. We note that here, we are addressing the robustness of the classifier against perturbations in the extracted feature; how robust the end-to-end model is also depends on the property of the feature extractor.
>
> **How our findings benefit the ML Community**: Our contributions are:
> - (High-level) We show that for certain data distributions, one can learn a provably robust classifier using standard learning methods and without adding a defense mechanism.
> - (Theoretical/Technical) Our main theorem introduces the analysis of GF on shallow networks with new activation functions, and reviewer KSsi thinks "the analysis methods developed here may serve as the foundation for future work to analyze more complicated (and more practical) settings."
> - (Practical) We have expanded our remarks on the practical implications of our results in this rebuttal and will add them to the revised manuscript.
>
> **Other additions**: We will add a conclusion section based on the responses above. Moreover, we will add numerical experiments to validate our theoretical results and remarks on future work (Please see our rebuttal to reviewer hboM).
>
> We hope our rebuttal addresses your concerns regarding the organization of our paper. If you have additional questions/concerns, please post them during the discussion.
>
> Reference:
>
> Li, B., Pan, Z., Lyu, K., and Li, J. Feature averaging: An implicit bias of gradient descent leading to non-robustness in neural networks. arXiv preprint arXiv:2410.10322, 2024.

---

### Official Review · Reviewer_KSsi · 2025-03-18

**Overall Recommendation:** 4

**Summary:**

This paper analyzes the gradient flow dynamics of training a pReLU neural network. It is shown for data coming from Gaussian mixture models with orthonormal cluster centers that, under some technical initialization conditions, the dynamics converge to a particular pReLU model that acts similar to a nearest-cluster rule, with a known (and high) level of robustness.

**Claims And Evidence:**

The claims are all theoretical, and are well-supported by clear and rigorous mathematical analyses.

**Essential References Not Discussed:**

N/A

**Experimental Designs Or Analyses:**

There are no experiments.

**Methods And Evaluation Criteria:**

N/A

**Other Comments Or Suggestions:**

1. Line 98 and Figure 1: Why would those two Gaussian clusters concentrate on D-1 dimensional affine subspaces rather than on spheres?
2. Line 119, Column 2: It seems odd to restrict your attack model to a sphere of radius $r$ rather than a ball of radius $r$, which is the more common method for modeling additive adversarial attacks. That is, why are you using $\min_{\|d\| = 1}$ rather than $\min_{\|d\|\le 1}$? With your model, is it not possible to potentially find some large radius $r$ for which the model is robust, even though there may exists some smaller radius $r' < r$ at which an attack results in misclassification?
3. The interpretation of the Bayes optimal classifier as a nearest cluster rule (with some error term) is very interesting.
4. Notation in Proposition 3: By $x\in S^{D-1}$, do you mean the $D-1$ dimensional unit sphere? If so, it would be good to explicitly define this notation for the reader.
5. Notation: Typically set minus is denoted with a backslash ("\setminus" in LaTeX), not a forward slash.
6. Typo in Assumption 2: "The initialization has s"... there seems to be an extra floating letter "s".
7. Overall, I find the geometric descriptions of the rather technical initialization conditions in Section 3.2.1 to be very nice, and enlightening.
8. On the takeaway message of Theorem 1, and discussion "Nearly optimal robust classifier via GF": Does your analysis guarantee that GF will not suddenly diverge away from the robust baseline $F^{(p)}$ once the flow time increases beyond $T^*$? It seems to me like this might be a possible limitation. If this is the case, how would one know when to stop the gradient flow process to "capture" the right set of parameters giving maximal robustness?
9. Your analysis in Theorem 1 seems to be saying that GF will result in $f^{(p)}$ approaching $F^{(p)}$, and then claiming robustness of $f^{(p)}$ from that of $F^{(p)}$. If $F^{(p)}$ is such a desirable baseline for robust prediction, why wouldn't one simply use $F^{(p)}$ for prediction in the first place? What is one gaining by learning $f^{(p)}$?
10. Footnote 3: This footnote should be placed after the punctuation, to avoid looking like a mathematical "cubed" symbol.

**Other Strengths And Weaknesses:**

The paper is extremely well-written. The content is highly technical, but the authors do a good job at shedding light on what the technical conditions mean intuitively (for instance, through geometric descriptions).

The problem setup is definitely impractical in terms of the assumptions being made, as the authors themselves have acknowledged. However, in my opinion, this does not necessarily detract from the contributions and possible impact of the paper; the paper's goal is to conduct a theoretical analysis of a gradient flow that is in general intractable to characterize for complicated data distributions and initialization schemes. So, appropriate assumptions are employed and clearly stated in order to reduce the analysis down to a tractable one, enabling the authors to attain their goal (and to prove a previously open conjecture). Such simplified analyses can still offer intuitive insights into what we may expect for gradient-based learning in the cases where some of the assumptions are relaxed. Furthermore, the analysis methods developed here may serve as the foundation for future work to analyze more complicated (and more practical) settings.

That said, I do have a few questions and concerns that I would like to see addressed. See below.

**Questions For Authors:**

See above "Other Comments Or Suggestions."

**Relation To Broader Scientific Literature:**

This paper is somewhat niche, with impacts primarily in the area of mathematical analysis of neural network training.

**Theoretical Claims:**

The mathematical analyses appear to be rigorous and correct.

---

> ### Author Rebuttal · Authors · 2025-03-31
>
> Thank you for the valuable comments and suggestions, and for the encouraging words acknowledging the strengths of our manuscript. Here are our responses to your questions and concerns:
>
> **Concentration of Gaussians**: The claim that a Gaussian $\mathcal{N}(\mu,\frac{\alpha^2}{D}I)$ concentrates around a $(D-1)$-dimensional affine subspace is derived from the fact that (1) as the reviewer pointed out, it concentrates around a sphere, and at the same time, (2) one can cover most masses of this high-dimensional sphere by any set $S$ as long as $S$ is a Minkowski sum of a $\mathcal{O}(1/\sqrt{D})$-radius ball and a $(D-1)$-dimensional affine subspace that contains the Gaussian mean $\mu$ (one should think $S$ being the inflated version of the $(D-1)$-dimensional affine subspace with thickness $\mathcal{O}(1/\sqrt{D})$); This geometric property of high-dimensional sphere is discussed in the section 2.2 of the book from (Hopcroft and Kannan). Therefore, most masses of this Gaussian $\mathcal{N}(\mu,\frac{\alpha^2}{D}I)$ is within $\mathcal{O}(1/\sqrt{D})$ eculidean distance to the chosen affine subspace, from which we say in line 101, "the class-conditioned probability masses concentrate around two (D−1)-dimensional affine subspaces". We hope this clarifies and are happy to add this explanation to the revised manuscript.
>
> **Attack radius**: The $\|d\|=1$ in line 119 is a typo. Thank you for pointing it out. We have $\|d\|\leq 1$ in all other places.
>
> **Stop time in Theorem 1**: This is a technical limitation of our analysis. In Appendix A, line 592, we acknowledged this limitation in the remark "Analysis until finite time $T^*$", and we referred to some previous work on local convergence of GF, for example (Chatterjee, 2022), which can potentially be utilized to show that the weights stay around the neighborhood around where they are at time $T^*$ for the rest of the GF because the loss is already close to its global minimum (thus no sudden divergence).
>
> **Why not use $F^{(p)}$ directly**: Our aim is to theoretically understand how a neural network can learn a robust classifier by standard training. Although $F^{(p)}$ can be easily constructed, it is unknown whether a network can learn $F^{(p)}$, and our theorem highlights that provable learning $F^{(p)}$ with networks requires some specifical choice of activations.
>
> We also thank the reviewer for pointing out typos and notation issues in our manuscript. We will fix them in the revision.
>
> We hope our rebuttal addresses your concerns. If you have additional questions/concerns, please feel free to post them during the discussion phase.
>
> Reference:
>
> Hopcroft, J. and Kannan, R. Computer Science Theory for the Information Age.
>
> Chatterjee, S. Convergence of gradient descent for deep neural networks. arXiv preprint arXiv:2203.16462, 2022.

---

> > ### Comment · Reviewer_KSsi · 2025-04-03
> >
> > Thank you for your thorough responses and clarifications. I encourage you to incorporate your explanations for the concentration, the stop time, and the motivation for studying the training dynamics (and not necessarily construction of a new or better model), in your revisions. Overall, I find this to be a very nice theoretical paper. I increase my score to 4 (accept).

---

### Decision · Program_Chairs · 2025-05-01

**Decision:**

Accept (poster)

**Comment:**

The paper proves that, with suitable choice of activation (pReLU), one can learn robust classifiers even with standard gradient descent. The analysis is under the framework of a Gaussian mixture model, and the dynamics for the pReLU activation is quite non-trivial. All reviewers agreed that the theoretical results are solid and interesting, while some reviewers raised concerns about the writing style. The authors are encouraged to include the simulation results in the revised version.